# RESOLVNET: A GRAPH CONVOLUTIONAL NETWORK WITH MULTI-SCALE CONSISTENCY

## ABSTRACT

It is by now a well known fact in the graph learning community that the presence of bottlenecks severely limits the ability of graph neural networks to propagate information over long distances. What so far has not been appreciated is that, counter-intuitively, also the presence of strongly connected sub-graphs may severely restrict information flow in common architectures. Motivated by this observation, we introduce the concept of multi-scale consistency. At the node level this concept refers to the retention of a connected propagation graph even if connectivity varies over a given graph. At the graph-level, a multi-scale consistent graph neural network assigns similar feature vectors to distinct graphs describing the same object at different resolutions. As we show, both properties are not satisfied by poular graph neural network architectures. To remedy these shortcomings, we introduce ResolvNet, a flexible graph neural network based on the mathematical concept of resolvents. We rigorously establish its multi-scale consistency theoretically and verify it in extensive experiments on both synthetic and real-world data: Here networks based on this ResolvNet architecture prove expressive, stable and transferable.

## 1 INTRODUCTION

Learning on graphs has developed into a rich and complex field, providing spectacular results on problems as varied as protein design (Pablo Gainza, 2023), traffic forecasting (Li et al., 2018b), particle physics (Shlomi et al., 2021), recommender systems (Gao et al., 2023) and traditional tasks such as node- and graph classification (Wu et al., 2021; Xiao et al., 2022) and recently also in the self-supervised setting (Hou et al., 2022; 2023). Despite their successes, graph neural networks (GNNs) are still plagued by fundamental issues: Perhaps best known is the phenomenon of over-smoothing, capturing the fact that node-features generated by common GNN architectures become less informative as network depth increases (Li et al., 2018a; Oono & Suzuki, 2020). From the perspective of information flow however deeper networks would be preferable, as a $K$ layer message passing network (Gilmer et al., 2017), may only facilitate information exchange between nodes that are at most $K$-edges apart – a phenomenon commonly referred to as under-reaching (Alon & Yahav, 2021; Topping et al., 2021).

However, even if information *is* reachable within $K$ edges, the structure of the graph might not be conducive to communicating it between distant nodes: If bottlenecks are present in the graph at hand, information from an exponentially growing receptive field needs to be squashed into fixed-size vectors to pass through the bottleneck. This oversquashing-phenomenon (Alon & Yahav, 2021; Topping et al., 2021) prevents common architectures from propagating messages between distant nodes without information loss in the presence of bottlenecks.

What has so far not been appreciated within the graph learning community is that – somewhat counter-intuitively – also the presence of strongly connected subgraphs severely restricts the information flow within popular graph neural network architectures; as we establish in this work. Motivated by this observation, we consider the setting of multi-scale graphs and introduce, define and study the corresponding problem of multi-scale consistency for graph neural networks:

Multi-scale graphs are graphs whose edges are distributed on (at least) two scales: One large scale indicating strong connections within certain (connected) clusters, and one regular scale indicating a weaker, regular connectivity outside these subgraphs. The lack of multi-scale consistency of common

architectures then arises as two sides of the same coin: At the node level, prominent GNNs are unable to consistently integrate multiple connectivity scales into their propagation schemes: They essentially only propagate information along edges corresponding to the largest scale. At the graph level, current methods are not stable to variations in resolution scale: Two graphs describing the same underlying object at different resolutions are assigned vastly different feature vectors.

**Contributions:** We introduce the concept of multi-scale consistency for GNNs and study its two defining characteristics at the node- and graph levels. We establish that common GNN architectures suffer from a lack of multi-scale consistency and – to remedy this shortcoming – propose the **ResolvNet** architecture. This method is able to consistently integrate multiple connectivity scales occurring within graphs. At the node level, this manifests as ResolvNet – in contrast to common architectures – not being limited to propagating information via a severely disconnected effective propagation scheme, when multiple scales are present within a given graph. At the graph-level, this leads to ResolvNet provably and numerically verifiably assigning similar feature vectors to graphs describing the same underlying object at varying resolution scales; a property which – to the best of our knowledge – no other graph neural network has demonstrated.

## 2 MULTI-SCALE GRAPHS AND MULTI-SCALE CONSISTENCY

### 2.1 MULTI-SCALE GRAPHS

We are interested in graphs with edges distributed on (at least) two scales: A large scale indicating strong connections within certain clusters, and a regular scale indicating a weaker, regular connectivity outside these subgraphs. Before giving a precise definition, we consider two instructive examples:

**Example I. Large Weights:** A two-scale geometry as outlined above, might e.g. arise within weighted graphs discretizing underlying continuous spaces: Here, edge weights are typically determined by the inverse discretization length ($w_{ij} \sim 1/d_{ij}$), which might vary over the graph (Post, 2012; Post & Simmer, 2021). Strongly connected sub-graphs would then correspond to clusters of nodes that are spatially closely co-located. Alternatively, such different scales can occur in social networks; e.g. if edge-weights are set to number of exchanged messages. Nodes representing (groups of) close friends would then typically be connected by stronger edges than nodes encoding mere acquaintances, which would typically have exchanged fewer messages.

Given such a weighted graph, we partitions its weighted adjacency matrix $W = W_{\text{reg.}} + W_{\text{high}}$ into a disjoint sum over a part $W_{\text{reg.}}$ containing only regular edge-weights and part $W_{\text{high}}$ containing only large edge-weights. This decomposition induces two graph structures on the common node set $\mathcal{G}$: We set $G_{\text{reg.}} := (\mathcal{G}, W_{\text{reg.}})$ and $G_{\text{high}} := (\mathcal{G}, W_{\text{high}})$ (c.f. also Fig. 1).

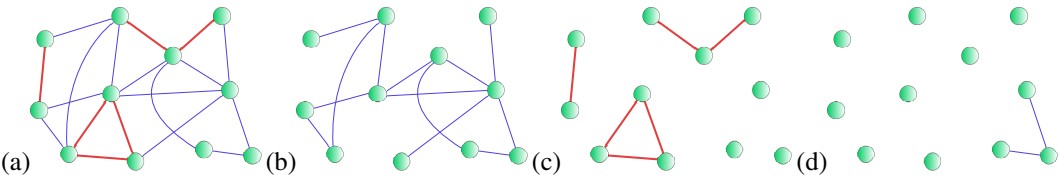

Figure 1: (a) Graph $G$ with $\mathcal{E}_{\text{reg.}}$ (blue) & $\mathcal{E}_{\text{high}}$ (red); (b) $G_{\text{reg.}}$; (c) $G_{\text{high}}$; (d) $G_{\text{excl.-reg.}}$

In preparation for our discussion in Section 2.2, we also define the graph $G_{\text{excl.-reg.}}$ whose edges consists of those elements $(i, j) \in \mathcal{G} \times \mathcal{G}$ that do not have a neighbouring edge in $G_{\text{high}}$; i.e. those edges $(i, j) \in \mathcal{E} \subsetneq \mathcal{G} \times \mathcal{G}$ so that for any $k \in \mathcal{G}$ we have $(W_{\text{high}})_{ik}, (W_{\text{high}})_{kj} = 0$ (c.f. Fig. 1 (d)).

**Example 2. Many Connections:** Beyond weighted edges, disparate connectivities may also arise in unweightd graphs with binary adjacency matrices: In a social network where edge weights encode a binary friendship status for example, there might still exist closely knit communities within which every user is friends with every other, while connections between such friend-groups may be sparser.

Here we may again split the adjacency matrix $W = W_{\text{reg.}} + W_{\text{high}}$ into a disjoint sum over a part $W_{\text{reg.}}$ encoding regular connectivity outside of tight friend groups and a summand $W_{\text{high}}$ encoding closely knit communities into dense matrix blocks. Fig. 2 depicts the corresponding graph structures.

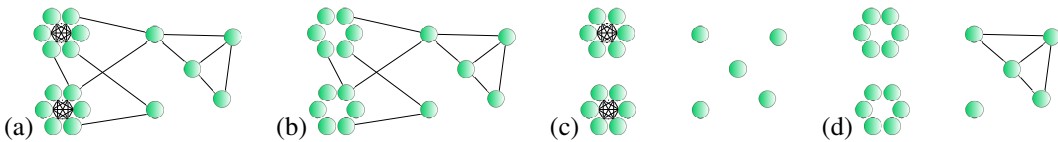

Figure 2: (a) Graph $G$; (b) $G_{\text{reg.}}$; (c) $G_{\text{high}}$; (d) $G_{\text{excl.-reg.}}$.

**Exact Definition:** To unify both examples above into a common framework, we make use of tools from spectral graph theory; namely the spectral properties of the **Graph Laplacian**:

Given a graph $G$ on $N$ nodes, with weighted adjacency matrix $W$, diagonal degree matrix $D$ and node weights $\{\mu_i\}_{i=1}^N$ collected into the (diagonal) node-weight matrix $M = \text{diag}(\{\mu_i\})$, the (un-normalized) graph Laplacian $\Delta$ associated to the graph $G$ is defined as

$$\Delta = M^{-1}(D - W).$$

It is a well known fact in spectral graph theory, that much information about the connectivity of the graph $G$ is encoded into the first (i.e. smallest) non-zero eigenvalue $\lambda_1(\Delta)$ of this graph Laplacian $\Delta$ (Brouwer & Haemers, 2012; Chung, 1997). For an unweighted graph $G$ on $N$ nodes, this eigenvalue $\lambda_1(\Delta)$ is for example maximised if every node is connected to all other nodes (i.e. $G$ is an $N$-clique); in which case we have $\lambda_1(\Delta) = N$. For weighted graphs, it is clear that scaling all weights by a (large) constant $c$ exactly also scales this eigenvalue as $\lambda_1(\Delta) \mapsto c \cdot \lambda_1(\Delta)$. Thus the eigenvalue $\lambda_1(\Delta)$ is indeed a good proxy for measuring the strength of communities within a given graph $G$.

In order to formalize the concept of multi-scale graphs containing strongly connected subgraphs, we thus make the following definition:

**Definition 2.1.** A Graph is called multi-scale if its weight-matrix $W$ admits a *disjoint* decomposition

$$W = W_{\text{reg.}} + W_{\text{high}} \quad \text{with} \quad \lambda_1(\Delta_{\text{high}}) > \lambda_{\max}(\Delta_{\text{reg.}}).$$

Note that this decomposition of $W$ also implies $\Delta = \Delta_{\text{reg.}} + \Delta_{\text{high}}$ for the respective Laplacians. Note also that the graph-structure determined by $G_{\text{high}}$ need not be completely connected for $\lambda_1(\Delta_{\text{high}})$ to be large (c.f. Fig.s 1 and 2 (c)): If there are multiple disconnected communities, $\lambda_1(\Delta_{\text{high}})$ is given as the minimal *non-zero* eigenvalue of $\Delta_{\text{high}}$ restricted to these individual components of $G_{\text{high}}$. The largest eigenvalue $\lambda_{\max}(\Delta_{\text{reg.}})$ of $\Delta_{\text{reg.}}$ can be interpreted as measuring the "maximal connectivity" within the graph structure $G_{\text{reg.}}$: By means of Gershgorin's circle theorem (Bárány & Solymosi, 2017), we may bound it as $\lambda_{\max}(\Delta_{\text{reg.}}) \leqslant 2 \cdot d_{\text{reg.,max}}$, with $d_{\text{reg.,max}}$ the maximal node-degree occuring in the graph $G_{\text{reg.}}$. Hence $\lambda_{\max}(\Delta_{\text{reg.}})$ is small, if the connectivity within $G_{\text{reg.}}$ is sparse.

## 2.2 MULTI-SCALE CONSISTENCY

We are now especially interested in the setting where the scales occuring in a given graph $G$ are well separated (i.e. $\lambda_1(\Delta_{\text{high}}) \gg \lambda_{\max}(\Delta_{\text{reg.}})$). Below, we describe how graph neural networks should ideally consistently incorporate such differing scales and detail how current architectures fail to do so. As the influence of multiple scales within graphs manifests differently depending on whether node-level- or graph-level tasks are considered, we will discuss these settings separately.

### 2.2.1 NODE LEVEL CONSISTENCY: RETENTION OF CONNECTED PROPAGATION GRAPHS

The fundamental purpose of graph neural networks is that of generating node embeddings not only dependent on local node-features, but also those of surrounding nodes. Even in the presence of multiple scales in a graph $G$, it is thus very much desirable that information is propagated between all nodes connected via the edges of $G$ – and not, say, only along the dominant scale (i.e. via $G_{\text{high}}$).

This is however not the case for popular graph neural network architectures: Consider for example the graph convolutional network GCN (Kipf & Welling, 2017): Here, feature matrices $X$ are updated via the update rule $X \mapsto \hat{A} \cdot X$, with the off-diagonal elements of $\hat{A}$ given as $\hat{A}_{ij} = W_{ij}/\sqrt{\hat{d}_i \cdot \hat{d}_j}$. Hence the relative importance $\hat{A}_{ij}$ of a message between a node $i$ of large (renormalised) degree $\hat{d}_i \gg 1$ and a node $j$ that is less strongly connected (e.g. $\hat{d}_j = \mathcal{O}(1)$) is severely discounted.

In the presence of multiple scales as in Section 2.1, this thus leads to messages essentially only being propagated over a disconnected effective propagation graph that is determined by the effective behaviour of $\hat{A}$ in the presence of multiple scales. As we show in Appendix A using the decompositions $W = W_{\text{reg.}} + W_{\text{high}}$, the matrix $\hat{A}$ can in this setting effectively be approximated as:

$$\hat{A} \approx \left( D_{\text{high}}^{-\frac{1}{2}} W_{\text{high}} D_{\text{high}}^{-\frac{1}{2}} + D_{\text{reg.}}^{-\frac{1}{2}} \tilde{W}_{\text{excl.-reg.}} D_{\text{reg.}}^{-\frac{1}{2}} \right)$$

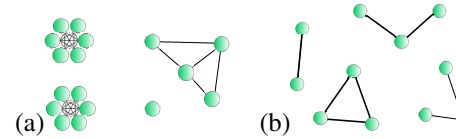

Thus information is essentially only propagated within the connected components of $G_{\text{high}}$ and via edges in $G_{\text{excl.-reg.}}$ (detached from edges in $G_{\text{high}}$).

Figure 3: Effective propagation graphs for original graphs in Fig. 2 (a) and Fig. 1 (a)

Appendix A further details that this reduction to propagating information only along a disconnected effective graph in the presence of multiple scales generically persists for popular methods (such as e.g. attention based methods (Velickovic et al., 2018) or spectral methods (Defferrard et al., 2016)).

Propagating only over severely disconnected effective graphs as in Fig. 3 is clearly detrimental:

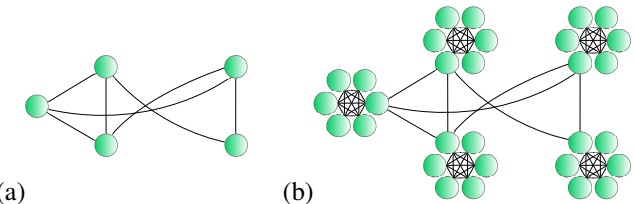

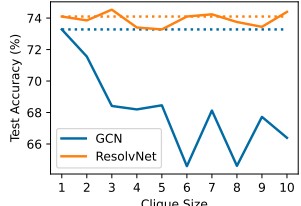

Figure 4: Individual nodes (a) replaced by 6-cliques (b)

Figure 5: Classification Accuracy

As is evident from GCN's performance in Fig.5, duplicating individual nodes of a popular graph dataset into fully connected $k$-cliques as in Fig. 4 leads to a significant decrease in node-classification accuracy, as propagation between cliques becomes increasingly difficult with growing clique-size $k$. Details are provided in the Experimental-Section 5. In principle however, duplicating nodes does not increase the complexity of the classification task at hand: Nodes and corresponding labels are only duplicated in the train-, val.- and test-sets. What *is* changing however, is the geometry underlying the problem; turning from a one-scale- into a two-scale setting with increasingly separated scales.

In Section 3 below, we introduce ResolvNet, which is able to consistently integrate multiple scales within a given graph into its propagation scheme. As a result (c.f. Fig. 5) its classification accuracy is not affected by an increasing clique-size $k$ (i.e. an increasing imbalance in the underlying geometry).

### 2.2.2 GRAPH LEVEL CONSISTENCY: TRANSFERABILITY BETWEEN DIFFERENT RESOLUTIONS

At the graph level, we desire that graph-level feature vectors $\Psi(G)$ generated by a network $\Psi$ for graphs $G$ are stable to changes in resolution scales: More precisely, if two graphs $G$ and $\underline{G}$ describe the same underlying object, space or phenomenon at different resolution scales, the generated feature vectors should be close, as they encode *the same* object in the latent space. Ideally, we would have a Lipschitz continuity relation that allows to bound the difference in generated feature vectors $\|\Phi(G) - \Phi(\underline{G})\|$ in terms of a judiciously chosen distance $d(G, \underline{G})$ between the graphs as

$$\|\Psi(G) - \Psi(\underline{G})\| \lesssim d(G, \underline{G}). \tag{1}$$

Note that a relation such as (1) also allows to make statements about *different* graphs $G, \tilde{G}$ describing an underlying object at *the same* resolution scale: If both such graphs are close to *the same* coarse grained description $\underline{G}$, the triangle inequality yields $\|\Psi(G) - \Psi(\tilde{G})\| \lesssim (d(G, \underline{G}) + d(\tilde{G}, \underline{G})) \ll 1$.

To make precise what we mean by the coarse grained description $\underline{G}$, we revisit the example of graphs discretising an underlying continuous space, with edge weights corresponding to inverse discretization length ($w_{ij} \sim 1/d_{ij}$). Coarse-graining – or equivalently lowering the resolution scale – then corresponds to merging multiple spatially co-located nodes in the original graph $G$ into single aggregate nodes in $\underline{G}$. As distance scales inversely with edge-weight, this means that we are precisely collapsing the strongly connected clusters within $G_{\text{high}}$ into single nodes. Mathematically, we then make this definition of the (lower resolution) coarse-grained graph $\underline{G}$ exact as follows:

**Definition 2.2.** Denote by $\mathcal{G}$ the set of connected components in $G_{\text{high}}$. We give this set a graph structure $\underline{G}$ as follows: Let $R$ and $P$ be elements of $\mathcal{G}$ (i.e. connected components in $G_{\text{high}}$). We define the real number $\underline{W}_{RP}$ as $\underline{W}_{RP} = \sum_{r \in R} \sum_{p \in P} W_{rp}$, with $r$ and $p$ nodes in the original graph $G$. We define the set of edges $\underline{\mathcal{E}}$ on $\underline{G}$ as $\underline{\mathcal{E}} = \{(R, P) \in \mathcal{G} \times \mathcal{G} : \underline{W}_{RP} > 0\}$ and assign $\underline{W}_{RP}$ as weight to such edges. Node weights of nodes in $\underline{G}$ are defined similarly by aggregating weights of all nodes $r$ contained in the connected component $R$ of $G_{\text{high}}$ as $\underline{\mu}_R = \sum_{r \in R} \mu_r$.

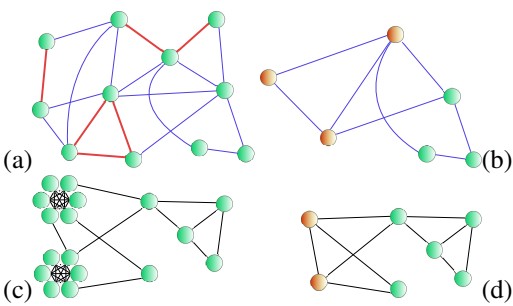

Figure 6: Original $G$ (a,c) and coarsified $\underline{G}$ (b,d)

This definition is of course also applicable to Example 2 of Section 2.1. Collapsing corresponding strongly connected component in a social network might then e.g. be interpreted as moving from interactions between individual users to considering interactions between (tightly-knit) communities.

While there have been theoretical investigations into this issue of **transferability** of graph neural networks between *distinct graphs* describing the *same system* (Levie et al., 2019; Ruiz et al., 2020; Maskey et al., 2021; Koke, 2023), the construction of an actual network with such properties – especially outside the asymptotic realm of very large graphs – has – to the best of our knowledge – so far not been successful. In Theorem 4.2 and Section 5 below, we show however that the ResolvNet architecture introduced in Section 3 below indeed provably and numerically verifiably satisfies (1), and is thus robust to variations in fine-print articulations of graphs describing the same object.

## 3 RESOLVNET

We now design a network – termed ResolvNet – that can consistently incorporate multiple scales within a given graph into its propagation scheme. At the node level, we clearly want to avoid disconnected effective propagation schemes as discussed in Section 2.2.1 in settings with well-separated connectivity scales. At the graph level – following the discussion of Section 2.2.2 – we want to ensure that graphs $G$ containing strongly connected clusters and graphs $\underline{G}$ where these clusters are collapsed into single nodes are assigned similar feature vectors.

We can ensure both properties at the same time, if we manage to design a network whose propagation scheme when deployed on a multi-scale graph $G$ is effectively described by propagating over a coarse grained version $\underline{G}$ if the connectivity within the strongly connected clusters $G_{\text{high}}$ of $G$ is very large:

- At the node level, this avoids effectively propagating over disconnected limit graphs as in Section 2.2.1. Instead, information within strongly connected clusters is approximately homogenized and message passing is then performed on a (much better connected) coarse-grained version $\underline{G}$ of the original graph $G$ (c.f. Fig. 6).

- At the graph level, this means that the stronger the connectivity within the strongly connected clusters is, the more the employed propagation on $G$ is like that on its coarse grained version $\underline{G}$. As we will see below, this can then be used to ensure the continuity property (1).

### 3.1 THE RESOVENT OPERATOR

As we have seen in Section 2.2.1 (and as is further discussed in Appendix A), standard message passing schemes are unable to generate networks having our desired multi-scale consistency properties.

A convenient multi-scale description of graphs is instead provided by the graph Laplacian $\Delta$ (c.f. Section 2.1), as this operator encodes information about coarse geometry of a graph $G$ into small eigenvalues, while fine-print articulations of graphs correspond to large eigenvalues. (Brouwer & Haemers, 2012; Chung, 1997). We are thus motivated to make use of this operator in our propagation scheme for ResolvNet.

In the setting of Example I of Section 2.1, letting the weights within $G_{\text{high}}$ go to infinity (i.e. increasing the connectivity within the strongly connected clusters) however implies $\|\Delta\| \to \infty$ for the norm of the Laplacian on $G$. Hence we *can not* implement propagation simply as $X \mapsto \Delta \cdot X$: This would

not reproduce the corresponding propagation scheme on $\underline{G}$ as we increase the connectivity within $G_{\text{high}}$: The Laplacian on $G$ does not converge to the Laplacian on $\underline{G}$ in the usual sense (it instead diverges $\|\Delta\| \to \infty$).

In order to capture convergence between operators with such (potentially) diverging norms, mathematicians have developed other – more refined – concepts: Instead of distances between original operators, one considers distances between **resolvents** of such operators (Teschl, 2014) :

**Definition 3.1.** The resolvent of an operator $\Delta$ is defined as $R_z(\Delta) := (\Delta - z \cdot Id)^{-1}$, with $Id$ the identity mapping. Such resolvents are defined whenever $z$ is not an eigenvalue of $\Delta$.

For Laplacians, taking $z < 0$ hence ensures $R_z(\Delta)$ is defined. Using this concept, we now rigorously establish convergence (in the resolvent sense) of the Laplacian $\Delta$ on $G$ to the (coarse grained) Laplacian $\underline{\Delta}$ on $\underline{G}$ as the connectivity within $G_{\text{high}}$ is increased. To rigorously do so, we need to be able to translate signals between the original graph $G$ and its coarse-grained version $\underline{G}$:

**Definition 3.2.** Let $x$ be a scalar graph signal. Denote by $\mathbb{1}_R$ the vector that has 1 as entries on nodes $r$ belonging to the connected (in $G_{\text{high}}$) component $R$ and has entry zero for all nodes not in $R$. We define the down-projection operator $J^{\downarrow}$ component-wise via evaluating at node $R$ in $\underline{\mathcal{G}}$ as $(J^{\downarrow}x)_R = \langle \mathbb{1}_R, x \rangle / \underline{\mu}_R$. This is then extended to feature *matrices* $\{X\}$ via linearity. The interpolation operator $J^{\uparrow}$ is defined as $J^{\uparrow}u = \sum_R u_R \cdot \mathbb{1}_R$; where $u_R$ is a scalar value (the component entry of $u$ at $R \in \underline{\mathcal{G}}$) and the sum is taken over all connected components of $G_{\text{high}}$.

With these preparations, we can rigorously establish that the *resolvent* of $\Delta$ approaches that of $\underline{\Delta}$:

**Theorem 3.3.** We have $R_z(\Delta) \to J^{\uparrow} R_z(\underline{\Delta}) J^{\downarrow}$ as connectivity within $G_{\text{high}}$ increases. Explicitly:

$$\left\| R_z(\Delta) - J^{\uparrow} R_z(\underline{\Delta}) J^{\downarrow} \right\| = \mathcal{O}\left( \frac{\lambda_{\max}(\Delta_{\text{reg.}})}{\lambda_1(\Delta_{\text{high}})} \right)$$

The fairly involved proof of Theorem 3.3 is contained in Appendix B and builds on previous work: We extend preliminary results in Koke (2023) by establishing *omni-directional* transferability (c.f. Theorem 4.1 below) and go beyond the toy-example of expanding a *single* node into a fixed and connected sub-graph with pre-defined edge-weights.

The basic idea behind ResolvNet is then to (essentially) implement message passing as $X \mapsto R_z(\Delta) \cdot X$. Building on Theorem 3.3, Section 4 below then makes precise how this rigorously enforces multiscale-consistency as introuced in Section 2.2 in the corresponding ResolvNet architecture.

## 3.2 The ResolvNet Architecture

Building on Section 3.1, we now design filters for which feature propagation essentially proceeds along the coarsified graph of Definition 2.2 as opposed to the disconnected effective graphs of Section 2.2.1, if multiple – well separated – edge-weight scales are present.

To this end, we note that Theorem 3.3 states for $\lambda_1(\Delta_{\text{high}}) \gg \lambda_{\max}(\Delta_{\text{reg.}})$ (i.e. well separated scales), that applying $R_z(\Delta)$ to a signal $x$ is essentially the same as first projecting $x$ to $\underline{G}$ via $J^{\downarrow}$, then applying $R_z(\underline{\Delta})$ there and finally lifting back to $G$ with $J^{\uparrow}$. Theorem B.4 In Appendix B establishes that this behaviour also persists for powers of resolvents; i.e. we also have $R_z^k(\Delta) \approx J^{\uparrow} R_z^k(\underline{\Delta}) J^{\downarrow}$.

**Resolvent filters:**   This motivates us to choose our learnable filters as polynomials in resolvents

$$f_{z,\theta}(\Delta) := \sum_{k=a}^{K} \theta_i \left[ (\Delta - zId)^{-1} \right]^k \tag{2}$$

with learnable parameters $\{\theta_k\}_{k=a}^{K}$. Thus our method can be interpreted as a spectral method (Defferrard et al., 2016), with learned functions $f_{z,\theta}(\lambda) = \sum_{k=a}^{K} \theta_k (\lambda - z)^{-k}$ applied to the operator $\Delta$ determining our convolutional filters. The parameter $a$, which determines the starting index of the sum in (2), may either be set to $a = 0$ (Type-0) or $a = 1$ (Type-I). As we show in Theorem 4.1 below, this choice will determine transferability properties of our models based on such filters.

Irrespectively, both Type-0 and Type-I filters are able to learn a wide array of functions, as the following theorem (proved in Appendix C) shows:

**Theorem 3.4.** Fix $\epsilon > 0$ and $z < 0$. For arbitrary functions $g, h : [0, \infty] \to \mathbb{R}$ with $\lim_{\lambda \to \infty} g(\lambda) =$ const. and $\lim_{\lambda \to \infty} h(\lambda) = 0$, there are filters $f_{z,\theta}^0, f_{z,\theta}^I$ of Type-0 and Type-I respectively such that $\|f_{z,\theta}^0 - g\|_\infty, \|f_{z,\theta}^I - h\|_\infty < \epsilon$.

**The ResolvNet Layer:** Collecting resolvent filters into a convolutional architecture, the layer wise update rule is then given as follows: Given a feature matrix $X^\ell \in \mathbb{R}^{N \times F_\ell}$ in layer $\ell$, with column vectors $\{X_j^\ell\}_{j=1}^{F_\ell}$, the feature vector $X_i^{\ell+1}$ in layer $\ell + 1$ is then calculated as $X_i^{\ell+1} =$ ReLu $\left( \sum_{j=1}^{F_\ell} f_{z,\theta_{ij}^{\ell+1}}(\Delta) \cdot X_j^\ell + b_i^{\ell+1} \right)$ with a learnable bias vector $b_i^{\ell+1}$. Collecting biases into a matrix $B^{\ell+1} \in \mathbb{R}^{N \times F_{\ell+1}}$, we can efficiently implement this using matrix-multiplications as

$$X^{\ell+1} = \text{ReLu} \left( \sum_{k=a}^K (\Delta - \omega Id)^{-k} \cdot X^\ell \cdot W_k^{\ell+1} + B^{\ell+1} \right)$$

with weight matrices $\{W_k^{\ell+1}\}$ in $\mathbb{R}^{F_\ell \times F_{\ell+1}}$. Biases are implemented as $b_i = \beta_i \cdot \mathbb{1}_G$, with $\mathbb{1}_G$ the vector of all ones on $G$ and $\beta_i \in \mathbb{R}$ learnable. This is done to ensure that the effective propagation on $\underline{G}$ (if well seperated scales are present in $G$) is not disturbed by non-transferable bias terms on the level of entire networks. This can be traced back to the fact that $J^\downarrow \mathbb{1}_G = \mathbb{1}_{\underline{G}}$ and $J^\uparrow \mathbb{1}_{\underline{G}} = \mathbb{1}_G$. A precise discussion of this matter is contained in Appendix D.

**Graph level feature aggregation:** As we will also consider the prediction of *graph-level* properties in our experimental Section 5 below, we need to sensibly aggregate node-level features into graph-level features on *node-weighted* graphs: As opposed to standard aggregation schemes (c.f. e.g. Xu et al. (2019)), we define an aggregation scheme $\Psi$ that takes into account node weights and maps a feature matrix $X \in \mathbb{R}^{N \times F}$ to a graph-level feature vector $\Psi(X) \in \mathbb{R}^F$ via $\Psi(X)_j = \sum_{i=1}^N |X_{ij}| \cdot \mu_i$.

# 4 MULTI-SCALE CONSISTENCY AND STABILITY GUARANTEES

**Node Level:** We now establish rigorously that instead of propagating along disconnected effective graphs (c.f. Fig. 3), ResolvNet instead propagates node features along the coarse-grained graphs of Fig. 6 if multiple separated scales are present:

**Theorem 4.1.** Let $\Phi$ and $\underline{\Phi}$ be the maps associated to ResolvNets with the same learned weight matrices and biases but deployed on graphs $G$ and $\underline{G}$ as defined in Section 3. We have

$$\|\Phi(J^\uparrow \underline{X}) - J^\uparrow \underline{\Phi}(\underline{X})\|_2 \leqslant (C_1(\mathscr{W}) \cdot \|\underline{X}\|_2 + C_2(\mathscr{W}, \mathscr{B})) \cdot \|R_z(\Delta) - J^\uparrow R_z(\underline{\Delta}) J^\downarrow\|$$

if the network is based on Type-0 resolvent filters (c.f. Section 3). Additionally, we have

$$\|\Phi(X) - J^\uparrow \underline{\Phi}(J^\downarrow X)\|_2 \leqslant (C_1(\mathscr{W}) \cdot \|X\|_2 + C_2(\mathscr{W}, \mathscr{B})) \cdot \|R_z(\Delta) - J^\uparrow R_z(\underline{\Delta}) J^\downarrow\|$$

if only Type-I filters are used in the network. Here $C_1(\mathscr{W})$ and $C_2(\mathscr{W}, \mathscr{B})$ are constants that depend polynomially on singular values of learned weight matrices $\mathscr{W}$ and biases $\mathscr{B}$.

The proof – as well as additional results – may be found in Appendix E. Note that Theorem 3.3 implies that both equations tends to zero for increasing scale separation $\lambda_1(\Delta_{\text{high}}) \gg \lambda_{\max}(\Delta_{\text{reg.}})$.

The difference between utilizing Type-0 and Type-I resolvent filters, already alluded to in the preceding Section 3, now can be understood as follows: Networks based on Type-0 filters effectively propagate signals *lifted* from the coarse grained graph $\underline{G}$ to the original graph $G$ along $\underline{G}$ if $\lambda_1(\Delta_{\text{high}}) \gg \lambda_{\max}(\Delta_{\text{reg.}})$. In contrast – in the same setting – networks based on Type-I resolvent filters effectively first *project* any input signal on $G$ to $\underline{G}$, propagate there and then lift back to $G$.

**Graph Level:** Beyond a single graph, we also establish graph-level multi-scale consistency: As discussed in Section 2.2.2, if two graphs describe the same underlying object (at different resolution scales) corresponding feature vectors should be similar. This is captured by our next result:

**Theorem 4.2.** Denote by $\Psi$ the aggregation method introduced in Section 3. With $\mu(G) = \sum_{i=1}^N \mu_i$ the total weight of the graph $G$, we have in the setting of Theorem 4.1 with Type-I filters, that

$$\|\Psi(\Phi(X)) - \Psi(\underline{\Phi}(J^\downarrow X))\|_2 \leqslant \sqrt{\mu(G)} (C_1(\mathscr{W}) \|X\|_2 + C_2(\mathscr{W}, \mathscr{B})) \|R_z(\Delta) - J^\uparrow R_z(\underline{\Delta}) J^\downarrow\|.$$

This result thus indeed establishes the desired continuity relation (1), with the distance metric $d(G, \underline{G})$ provided by the similarity $\|R_z(\Delta) - J^\uparrow R_z(\underline{\Delta}) J^\downarrow\|$ of the resolvents of the two graphs.

## 5 EXPERIMENTS

**Node Classification:**  To establish that the proposed **ResolvNet** architecture **not only performs well** in multi-scale settings, we conduct node classification experiments on multiple *un-weighted* real world datasets, ranging in edge-homophily $h$ from $h = 0.11$ (very heterophilic), to $h = 0.81$ (very homophilic). Baselines constitute an ample set of established and recent methods: Spectral approaches, are represented by ChebNet (Defferrard et al., 2016), GCN (Kipf & Welling, 2017), BernNet (He et al., 2021), ARMA (Bianchi et al., 2019), ChebNetII (He et al., 2022) and MagNet (Zhang et al., 2021). Spatial methods are given by GAT (Velickovic et al., 2018), SAGE (Hamilton et al., 2017) and GIN (Xu et al., 2019). We also consider PPNP (Gasteiger et al., 2019b), GCNII (Chen et al., 2020) and NSD (Bodnar et al., 2022). Details on datasets, experimental setup and hyperparameters are provided in Appendix F.

Table 1: Average Accuracies [%] with uncertainties encoding the 95 % confidence Level. Top three models are coloured-coded as **First**, **Second**, **Third**.

| | **MS. Acad.** | **Cora** | **Pubmed** | **Citeseer** | **Cornell** | **Actor** | **Squirrel** | **Texas** |
| $h$ | 0.81 | 0.81 | 0.80 | 0.74 | 0.30 | 0.22 | 0.22 | 0.11 |
|---|---|---|---|---|---|---|---|---|
| SAGE | $91.75_{\pm0.09}$ | $80.68_{\pm0.30}$ | $74.42_{\pm0.42}$ | $72.68_{\pm0.32}$ | $86.01_{\pm0.72}$ | $28.88_{\pm0.32}$ | $25.99_{\pm0.28}$ | $88.92_{\pm0.73}$ |
| GIN | $72.93_{\pm1.94}$ | $74.12_{\pm1.21}$ | $74.59_{\pm0.45}$ | $68.11_{\pm0.69}$ | $65.58_{\pm1.23}$ | $23.69_{\pm0.28}$ | $24.91_{\pm0.58}$ | $72.64_{\pm1.19}$ |
| GAT | $89.49_{\pm0.15}$ | $80.12_{\pm0.33}$ | $77.12_{\pm0.41}$ | $73.20_{\pm0.37}$ | $74.39_{\pm0.93}$ | $24.55_{\pm0.28}$ | $27.22_{\pm0.31}$ | $75.31_{\pm1.09}$ |
| NSD | $90.78_{\pm0.13}$ | $70.34_{\pm0.47}$ | $69.74_{\pm0.50}$ | $64.39_{\pm0.50}$ | $87.78_{\pm0.65}$ | $27.62_{\pm0.39}$ | $24.96_{\pm0.27}$ | $91.64_{\pm0.62}$ |
| PPNP | $91.22_{\pm0.13}$ | $83.77_{\pm0.27}$ | $78.42_{\pm0.31}$ | $73.25_{\pm0.37}$ | $71.93_{\pm0.84}$ | $25.93_{\pm0.35}$ | $23.69_{\pm0.43}$ | $70.73_{\pm1.27}$ |
| ChebNet | $91.62_{\pm0.10}$ | $78.70_{\pm0.37}$ | $73.63_{\pm0.43}$ | $72.10_{\pm0.43}$ | $85.99_{\pm0.10}$ | $29.51_{\pm0.31}$ | $25.68_{\pm0.28}$ | $91.01_{\pm0.59}$ |
| GCN | $90.81_{\pm0.10}$ | $81.49_{\pm0.36}$ | $76.60_{\pm0.44}$ | $71.34_{\pm0.45}$ | $73.35_{\pm0.88}$ | $24.60_{\pm0.28}$ | $30.40_{\pm0.40}$ | $76.16_{\pm1.12}$ |
| MagNet | $87.23_{\pm0.16}$ | $76.50_{\pm0.42}$ | $68.23_{\pm0.44}$ | $70.92_{\pm0.49}$ | $87.15_{\pm0.66}$ | $30.50_{\pm0.32}$ | $23.54_{\pm0.32}$ | $90.84_{\pm0.54}$ |
| ARMA | $88.97_{\pm0.18}$ | $81.24_{\pm0.24}$ | $76.28_{\pm0.35}$ | $70.64_{\pm0.45}$ | $83.68_{\pm0.80}$ | $24.40_{\pm0.45}$ | $22.72_{\pm0.42}$ | $87.41_{\pm0.73}$ |
| BernNet | $91.37_{\pm0.14}$ | $83.26_{\pm0.24}$ | $77.24_{\pm0.37}$ | $73.11_{\pm0.34}$ | $87.14_{\pm0.57}$ | $28.90_{\pm0.45}$ | $22.86_{\pm0.32}$ | $89.81_{\pm0.68}$ |
| ChebNetII | $92.45_{\pm0.14}$ | $80.56_{\pm0.34}$ | $77.79_{\pm0.41}$ | $72.72_{\pm0.45}$ | $85.24_{\pm0.92}$ | $22.83_{\pm0.28}$ | $22.83_{\pm0.28}$ | $86.14_{\pm0.84}$ |
| GCNII | $88.37_{\pm0.16}$ | $84.41_{\pm0.26}$ | $78.45_{\pm0.31}$ | $73.12_{\pm0.35}$ | $66.32_{\pm1.35}$ | $28.82_{\pm0.80}$ | $27.24_{\pm0.26}$ | $85.77_{\pm0.76}$ |
| ResolvNet | $92.73_{\pm0.08}$ | $84.16_{\pm0.26}$ | $79.29_{\pm0.36}$ | $75.03_{\pm0.29}$ | $84.92_{\pm1.43}$ | $29.06_{\pm0.32}$ | $26.51_{\pm0.23}$ | $87.73_{\pm0.89}$ |

As is evident from Table 1, ResolvNet performs better than all baselines on all but one homophilc dataset; on which the gap to the best performing model is negligible. This performance can be traced back to the inductive bias ResolvNet is equipped with by design: It might be summarized as "Nodes that are strongly connected should be assigned similar feature vectors" (c.f. Theorem 4.1) . This inductive bias – necessary to achieve a consistent incorporation of multiple scales – is of course counterproductive in the presence of heterophily; here nodes that are connected by edges generically have *differing* labels and should thus be assigned different feature vectors. However the ResolvNet architecture also performs well on most heterophilic graphs: It e.g. out-performs NSD – a recent state of the art method spefically developed for heterophily – on two such graphs.

**Node Classification for increasingly separated scales:** To test ResolvNet's ability to consistently incorporate multiple scales in the unweighted setting against a representative baseline, we duplicated individual nodes on the Citeseer dataset (Sen et al., 2008) $k$-times to form (fully connected) $k$-cliques; keeping the train-val-test partition constant. GCN and ResolvNet were then trained on the same ($k$-fold expanded) train-set and asked to classify nodes on the ($k$-fold expanded) test-partition. As discussed in Section 1 (c.f. Fig.5) GCN's performance decreased significantly, while ResolvNet's accuracy stayed essentially constant; showcasing its ability to consistently incorporate multiple scales.

**Regression on real-world multi-scale graphs:**  In order to showcase the properties of our newly developed method on real world data admitting a two-scale behaviour, we evaluate on the task of molecular property prediction. While ResolvNet is not designed for this setting, this task still allows to fairly compare its expressivity and stability properties against other non-specialized graph neural networks (Hu et al., 2020). Our dataset (QM7; Rupp et al. (2012)) contains descriptions of 7165 organic molecules; each containing both hydrogen and heavy atoms. A molecule is represented by its Coulomb matrix, whose off-diagonal elements $C_{ij} = Z_i Z_j / |\vec{x}_i - \vec{x}_j|$ correspond to the Coulomb repulsion between atoms $i$ and $j$. We treat $C$ as a weighted adjacency matrix. Prediction target is the molecular atomization energy, which – crucially – depends on long range interaction within molecules (Zhang et al., 2022). However, with edge-weights $C_{ij}$ scaling as inverse distance, long range propagation of information is scale-suppressed in the graph determined by $C$, when compared to the much larger weights between closer atoms. We choose Type-I filters in ResolvNet, set node weights as atomic charge ($\mu_i = Z_i$) and use one-hot encodings of atomic charges $Z_i$ as node features.

As is evident from Table 2, our method produces significantly lower mean-absolute-errors (MAEs) than baselines of Table 1 deployable on weighted graphs. We attribute this to the fact that our model allows for long range information propagation within each molecule, as propagation along corresponding edges is suppressed for baselines but not for our model (c.f. Section 2.2.1). Appendix contains additional experiments on QM9 (Ramakrishnan et al., 2014); finding similar performance for (long-range dependent) energy targets.

Table 2: QM7-MAE

| **QM7** ($\downarrow$) | MAE [$kcal/mol$] |
| --- | --- |
| BernNet | $113.57_{+62.90}$ |
| GCN | $61.32_{+1.62}$ |
| ChebNet | $59.57_{+1.58}$ |
| ARMA | $59.39_{+1.79}$ |
| ResolvNet | $\mathbf{16.52}_{+0.67}$ |

**Stability to varying the resolution-scale:** To numerically verify the Stability-Theorem 4.2 – which guarantees similar graph-level feature vectors for graphs describing the same underlying object at different resolution scales – we conduct additional experiments: We modify (all) molecular graphs of QM7 by deflecting hydrogen atoms (H) out of their equilibrium positions towards the respective nearest heavy atom. This introduces a two-scale setting precisely as discussed in section 2: Edge weights between heavy atoms remain the same, while Coulomb repulsions between H-atoms and respective nearest heavy atom increasingly diverge. Given an original molecular graph $G$ with node weights $\mu_i = Z_i$, the corresponding coarse-grained graph $\underline{G}$ corresponds to a description where heavy atoms and surrounding H-atoms are aggregated into single super-nodes. Node-features of aggregated nodes are now no longer one-hot encoded charges, but normalized bag-of-word vectors whose individual entries encode how much of the total charge of a given super-node is contributed by individual atom-types. Appendix F provides additional details and examples.

In this setting, we now compare features generated for coarsified graphs $\{\underline{G}\}$, with feature generated for graphs $\{G\}$ where hydrogen atoms have been deflected but have not yet completely arrived at the positions of nearest heavy atoms. Feature vectors are generated with the previously trained networks of Table 2. A corresponding plot is presented in Figure 7. Features generated by ResolvNet converge as the larger scale increases (i.e. the distance between hydrogen and heavy atoms decreases). This result numerically verifies the scale-invariance Theorem 4.2. As reference, we also plot the norm differences corresponding to baselines, which do not decay. We might thus conclude that these models – as opposed to ResolvNet – are scale- and resolution sensitive when generating graph level features. For BernNet we observe a divergence behaviour, which we attribute to numerical instabilities.

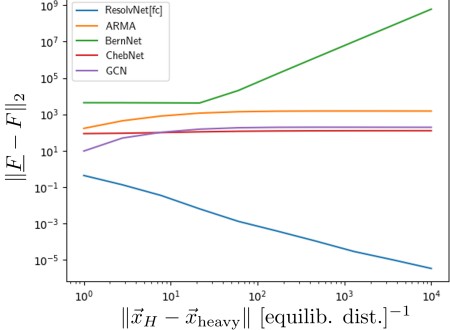

Figure 7: Feature-vector-difference for collapsed ($\underline{F}$) and deformed ($F$) graphs.

As a final experiment, we treat the coarse-grained molecular graphs $\{\underline{G}\}$ as a model for data obtained from a resolution-limited observation process, that is unable to resolve positions of hydrogen and only provides information about how many H-atoms are bound to a given heavy atom. Given models trained on higher resolution data, atomization energies for such observed molecules are now to be predicted. Table 3 contains corresponding results. While the performance of baselines decreases significantly if the resolution scale is varied during inference, the prediction accuracy of ResolvNet remains high; even slightly increasing. While ResolvNet out-performed baselines by a factor of three on same-resolution-scale data (c.f.Table 2), its lead increases to a factor of 10 and higher in the multi-scale setting.

Table 3: QM7$_{\text{coarse}}$-MAE

| **QM7** ($\downarrow$) | MAE [$kcal/mol$] |
| --- | --- |
| BernNet | $580.67_{+99.27}$ |
| GCN | $124.53_{+34.58}$ |
| ChebNet | $645.14_{+34.59}$ |
| ARMA | $248.96_{+15.56}$ |
| ResolvNet | $\mathbf{16.23}_{+2.74}$ |

**Additional Experiments:** We present additional conceptional results in Appendices G and H.

## 6 CONCLUSION

This work introduced the concept of multi-scale consistency: At the node level this refers to the retention of a propagation scheme not solely determined by the largest given connectivity scale. At the graph-level it mandates that distinct graphs describing the same object at different resolutions should be assigned similar feature vectors. Common GNN architectures were shown to not be multi-scale consistent, while the newly introduced ResolvNet architecture was theoretically and experimentally established to have this property. Deployed on real world data, ResolvNet proved expressive and stable; out-performing baselines significantly on many tasks in- and outside the multi-scale setting.

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

## A EFFECTIVE PROPAGATION SCHEMES

For definiteness, we here discuss limit-propagation schemes in the setting where **edge-weights** are large. The discussion for high-connectivity in the Sense of Example II of Section 2.1 proceeds in complete analogy.

In this section, we then take up again the setting of Section 2. We reformulate this setting here in a slightly modified language, that is more adapted to discussing effective propagation schemes of standard architectures:

We partition edges on a weighted graph $G$, into two disjoint sets $\mathcal{E} = \mathcal{E}_{\text{reg.}} \dot\cup \mathcal{E}_{\text{high}}$, where the set of edges with large weights is given by:

$$\mathcal{E}_{\text{high}} := \{(i,j) \in \mathcal{E} : w_{ij} \geq S_{\text{high}}\}$$

and the set with small weights is given by:

$$\mathcal{E}_{\text{reg.}} := \{(i,j) \in \mathcal{E} : w_{ij} \leq S_{\text{reg.}}\}$$

for weight scales $S_{\text{high}} > S_{\text{reg.}} > 0$. Without loss of generality, assume $S_{\text{reg.}}$ to be as low as possible (i.e. $S_{\text{reg.}} = \max_{(i,j) \in \mathcal{E}_{\text{reg.}}} w_{ij}$) and $S_{\text{high}}$ to be as high as possible (i.e. $S_{\text{large}} = \min_{(i,j) \in \mathcal{E}_{\text{high}}}$) and no weights in between the scales.

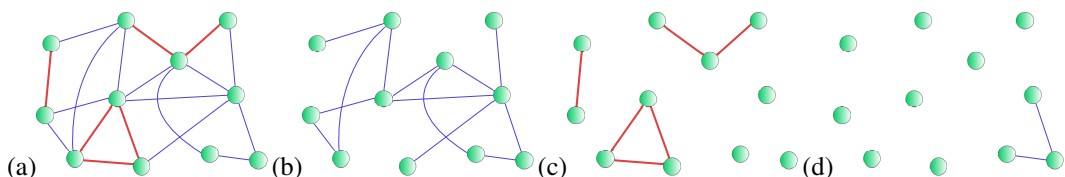

Figure 8: (a) Graph $G$ with $\mathcal{E}_{\text{reg.}}$ (blue) & $\mathcal{E}_{\text{high}}$ (red); (b) $G_{\text{reg.}}$; (c) $G_{\text{high}}$; (d) $G_{\text{reg., exclusive}}$

This decomposition induces two graph structures corresponding to the disjoint edge sets on the node set $\mathcal{G}$: We set $G_{\text{reg.}} := (\mathcal{G}, \mathcal{E}_{\text{reg.}})$ and $G_{\text{high}} := (\mathcal{G}, \mathcal{E}_{\text{high}})$ c.f. Fig. 8.
We also introduce the set of edges $\mathcal{E}_{\text{reg., exclusive}} := \{(i,j) \in \mathcal{E}_{\text{reg.}} | \forall k \in \mathcal{G} : (i,k) \notin \mathcal{E}_{\text{high}} \& (k,j) \notin \mathcal{E}_{\text{high}}\}$ connecting nodes that do not have an incident edge in $\mathcal{E}_{\text{high}}$. A corresponding example-graph $G_{\text{reg., exclusive}}$ is depicted in Fig. 8 (d).

We are now interested in the behaviour of graph convolution schemes if the scales are well separated:

$$S_{\text{high}} \gg S_{\text{reg.}}$$

### A.1 SPECTRAL CONVOLUTIONAL FILTERS

We first discuss resulting limit-propagation schemes for spectral convolutional networks. Such networks implement convolutional filters as a mapping

$$x \longmapsto g_\theta(T)x$$

for a node feature $x$, a learnable function $g_\theta$ and a graph shift operator $T$.

#### A.1.1 NEED FOR NORMALIZATION

The graph shift operator $T$ facilitating the graph convolutions needs to be normalized for established spectral graph convolutional architectures:

For Bianchi et al. (2019), this e.g. arises as a necessity for convergence of the proposed implementation scheme for the rational filters introduced there (c.f. eq. (10) in Bianchi et al. (2019)).

The work Defferrard et al. (2016) needs its graph shift operator to be normalized, as it approximates generic filters via a Chebyshev expansion. As argued in Defferrard et al. (2016), such Chebyshev polynomials form an orthogonal basis for the space $L^2([-1, 1], dx/\sqrt{1 - x^2})$. Hence, the spectrum of the operator $T$ to which the (approximated and learned) function $g_\theta$ is applied needs to be contained in the interval $[-1, 1]$.

In Kipf & Welling (2017), it has been noted that for the architecture proposed there, choosing $T$ to have eigenvalues in the range $[0, 2]$ (as opposed to the normalized ranges $[0, 1]$ or $[-1, 1]$) has the potential to lead to vanishing- or exploding gradients as well as numerical instabilities. To alleviate this, Kipf & Welling (2017) introduces a "renormalization trick" (c.f. Section 2.2. of Kipf & Welling (2017) to produce a normalized graph shift operator on which the network is then based.

We can understand the relationship between normalization of graph shift operator $T$ and the stability of corresponding convolutional filters explicitly: Assume that we have

$$\|T\| \gg 1.$$

This might e.g. happen when basing networks on the un-normalized graph Laplacian $\Delta$ or the weight-matrix $W$ if edge weights are potentially large (such as in the setting $S_{\text{high}} \gg S_{\text{reg.}}$ that we are considering).

By the spectral mapping theorem (see e.g. Teschl (2014)), we have

$$\sigma\left(g_\theta(T)\right) = \{g_\theta(\lambda) : \lambda \in \sigma(T)\}, \tag{3}$$

with $\sigma(T)$ denoting the spectrum (i.e. the set of eigenvalues) of $T$. For the largest (in absolute value) eigenvalue $\lambda_{\max}$ of $T$, we have

$$|\lambda_{\max}| = \|T\|. \tag{4}$$

Since learned functions are either implemented directly as a polynomial (as e.g. in Defferrard et al. (2016); He et al. (2021)) or approximated as a Neumann type power iteration (as e.g. in Bianchi et al. (2019); Gasteiger et al. (2019a)) which can be thought of as a polynomial, we have

$$\lim_{\lambda \to \pm\infty} |g_\theta(\lambda)| = \infty.$$

Thus in view of (3) and (4) we have for $\|T\|$ sufficiently large, that

$$\|g_\theta(T)\| = |g_\theta(\pm\|T\|)|$$

with the sign $\pm$ determined by $\lambda_{\max} \gtrless 0$. Since non-constant polynomials behave at least linearly for large inputs, there is a constant $C > 0$ such that

$$C \cdot \|T\| \leq \|g_\theta(T)\|$$

for all sufficiently large $\|T\|$. We thus have the estimate

$$\|x\| \cdot C \cdot \|T\| \leq \|g_\theta(T)x\|$$

for at least one input signal $x$ (more precisely all $x$ in the eigen-space corresponding to the largest (in absolute value) eigenvalue $\lambda_{\max}$). Thus if $T$ is not normalized (i.e. $\|T\|$ is not sufficiently bounded), the norm of (hidden) features might increase drastically when moving from one (hidden) layer to the next. This behaviour persists for all input signals $x$ have components in eigenspaces corresponding to large (in absolute value) eigenvalues of $T$.

### A.1.2 Spectral Normalizations

As discussed in the previous Section A.1.1, instabilities arising from non-normalized graph shift operators can be traced back to the problem of such operators having large eigenvalues. It was thus – among other considerations – suggested in Defferrard et al. (2016) to base convolutional filters on the spectrally normalized graph shift operator

$$T = \frac{1}{\lambda_{\max}(\Delta)}\Delta,$$

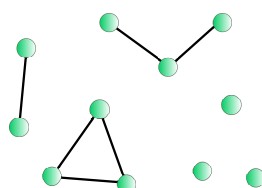

Figure 9: Limit graph corresponding to Fig 8 for spectral normalization

with $\Delta$ the un-normalized graph Laplacian. In the setting $S_{\text{high}} \gg S_{\text{reg.}}$ we are considering, this leads to an effective feature propagation along $G_{\text{high}}$ (c.f. also Fig. 9) only, as Theorem A.1 below establishes:

**Theorem A.1.** With

$$T = \frac{1}{\lambda_{\max}(\Delta)} \Delta,$$

and the scale decomposition as introduced in Section 2, we have that

$$\left\| T - \frac{1}{\lambda_{\max}(\Delta_{\text{high}})} \Delta_{\text{high}} \right\| = \mathcal{O}\left( \frac{S_{\text{reg.}}}{S_{\text{high}}} \right) \tag{5}$$

for $S_{\text{high}} \gg S_{\text{reg.}}$.

*Proof.* For convenience in notation, let us write

$$T_{\text{high}} = \frac{1}{\lambda_{\max}(\Delta_{\text{high}})} \Delta_{\text{high}}$$

and similarly

$$T_{\text{reg.}} = \frac{1}{\lambda_{\max}(\Delta_{\text{reg.}})} \Delta_{\text{reg.}}.$$

In section 2, we already noted that

$$\Delta = \Delta_{\text{high}} + \Delta_{\text{reg.}},$$

which we may rewrite as

$$\Delta = \lambda_{\max}(\Delta_{\text{high}}) \cdot \left( T_{\text{high}} + \frac{\lambda_{\max}(\Delta_{\text{reg.}})}{\lambda_{\max}(\Delta_{\text{high}})} \cdot T_{\text{reg.}} \right). \tag{6}$$

Let us consider the equivalent expression

$$\frac{1}{\lambda_{\max}(\Delta_{\text{high}})} \cdot \Delta = T_{\text{high}} + \frac{\lambda_{\max}(\Delta_{\text{reg.}})}{\lambda_{\max}(\Delta_{\text{high}})} \cdot T_{\text{reg.}}. \tag{7}$$

We next note that

$$\lambda_{\max}\left( \frac{1}{\lambda_{\max}(\Delta_{\text{high}})} \cdot \Delta \right) = \frac{\lambda_{\max}(\Delta)}{\lambda_{\max}(\Delta_{\text{high}})}. \tag{8}$$

and

$$\lambda_{\max}(T_{\text{high}}) = 1$$

since the operation of taking eigenvalues of operators is multiplicative in the sense of

$$\lambda_{\max}(|a| \cdot T) = |a| \cdot \lambda_{\max}(T)$$

for non-negative $|a| \geqslant 0$.

Since the right-hand-side of (7) constitutes an analytic perturbation of $T_{\text{high}}$, we may apply analytic perturbation theory (c.f. e.g. Kato (1976) for an extensive discussion) to this problem. With this (together with $\|T_{\text{high}}\| = 1$) we find

$$\lambda_{\max}\left( \frac{1}{\lambda_{\max}(\Delta_{\text{high}})} \cdot \Delta \right) = 1 + \mathcal{O}\left( \frac{\lambda_{\max}(\Delta_{\text{reg.}})}{\lambda_{\max}(\Delta_{\text{high}})} \right). \tag{9}$$

Using (8) and the fact that

$$\frac{\lambda_{\max}(\Delta_{\text{reg.}})}{\lambda_{\max}(\Delta_{\text{high}})} \propto \frac{S_{\text{reg.}}}{S_{\text{high}}}, \tag{10}$$

we thus have

$$\frac{\lambda_{\max}(\Delta)}{\lambda_{\max}(\Delta_{\text{high}})} = 1 + \mathcal{O}\left( \frac{S_{\text{reg.}}}{S_{\text{high}}} \right).$$

Since for small $\epsilon$, we also have

$$\frac{1}{1 + \epsilon} = 1 + \mathcal{O}(\epsilon),$$

the relation (10) also implies

$$\frac{\lambda_{\max}(\Delta_{\text{high}})}{\lambda_{\max}(\Delta)} = 1 + \mathcal{O}\left( \frac{S_{\text{reg.}}}{S_{\text{high}}} \right).$$

Multiplying (6) with $1/\lambda_{\max}(\Delta)$ yields

$$T = \frac{\lambda_{\max}(\Delta_{\text{high}})}{\lambda_{\max}(\Delta)} \cdot \left( T_{\text{high}} + \frac{\lambda_{\max}(\Delta_{\text{reg.}})}{\lambda_{\max}(\Delta_{\text{high}})} \cdot T_{\text{reg.}} \right). \tag{11}$$

Since $\|T_{\text{high}}\|, \|T_{\text{reg.}}\| = 1$ and

$$\frac{\lambda_{\max}(\Delta_{\text{reg.}})}{\lambda_{\max}(\Delta_{\text{high}})} \propto \frac{S_{\text{reg.}}}{S_{\text{high}}} < 1$$

for sufficiently large $S_{\text{high}}$, relation (11) implies

$$\left\| T - \frac{1}{\lambda_{\max}(\Delta_{\text{high}})} \Delta_{\text{high}} \right\| = \mathcal{O}\left( \frac{S_{\text{reg.}}}{S_{\text{high}}} \right)$$

as desired.

Note that we might in principle also make use of Lemma A.2 below, to provide quantitative bounds: Lemma A.2 states that

$$|\lambda_k(A) - \lambda_k(B)| \leqslant \|A - B\|$$

for self-adjoint operators $A$ and $B$ and their respective $k^{\text{th}}$ eigenvalues ordered by magnitude. On a graph with $N$ nodes, we clearly have $\lambda_{\max} = \lambda_N$ for eigenvalues of (rescaled) graph Laplacians, since all such eigenvalues are non-negative. This implies for the difference $|1 - \lambda_{\max}(\Delta)/\lambda_{\max}(\Delta_{\text{high}})|$ arising in (9) that explicitly

$$\left| 1 - \frac{\lambda_{\max}(\Delta)}{\lambda_{\max}(\Delta_{\text{high}})} \right| \leqslant \frac{\lambda_{\max}(\Delta_{\text{reg.}})}{\lambda_{\max}(\Delta_{\text{high}})}.$$

This in turn can then be used to provide a quantitative bound in (5). Since we are only interested in the qualitative behaviour for $S_{\text{high}} \gg S_{\text{reg.}}$, we shall however not pursue this further.

$\square$

It remains to state and establish Lemma A.2 referenced at the end of the proof of Theorem A.1:

**Lemma A.2.** Let $A$ and $B$ be two hermitian $n \times n$ dimensional matrices. Denote by $\{\lambda_k(M)\}_{k=1}^n$ the eigenvalues of a hermitian matrix in increasing order.
With this we have:

$$|\lambda_k(A) - \lambda_k(B)| \leqslant \|A - B\|.$$

*Proof.* After the redefinition $B \mapsto (-B)$, what we need to prove is

$$|\lambda_i(A + B) - \lambda_i(A)| \leqslant \|B\|$$

for Hermitian $A, B$. Since we have

$$\lambda_i(A) - \lambda_i(A + B) = \lambda_i((A + B) + (-B)) - \lambda_i(A + B)$$

and $\| - B\| = \|B\|$ it follows that it suffices to prove

$$\lambda_i(A + B) - \lambda_i(A) \leqslant \|B\|$$

for arbitrary hermitian $A, B$.

We note that the Courant-Fischer $\min - \max$ theorem tells us that if $A$ is an $n \times n$ Hermitian matrix, we have

$$\lambda_i(M) = \sup_{\dim(V)=i} \inf_{v \in V, \|v\|=1} v^* M v.$$

With this we find

$$
\begin{aligned}
\lambda_i(A+B) - \lambda_i(A) &= \sup_{\dim(V)=i} \inf_{v\in V, ||v||=1} v^*(A+B)v - \sup_{\dim(V)=i} \inf_{v\in V, ||v||=1} v^* A v \\
&\leqslant \sup_{\dim(V)=i} \inf_{v\in V, ||v||=1} v^* A v + \sup_{\dim(V)=i} \inf_{v\in V, ||v||=1} v^* B v \\
&\quad - \sup_{\dim(V)=i} \inf_{v\in V, ||v||=1} v^* A v \\
&= \sup_{\dim(V)=i} \inf_{v\in V, ||v||=1} v^* B v \\
&= \sup_{\dim(V)=i} \inf_{v\in V, ||v||=1} v^* B v \\
&\leqslant \max_{1\leqslant k\leqslant n} \{|\lambda_k(B)|\} \\
&= ||B||.
\end{aligned}
$$

$\square$

### A.1.3 Symmetric Normalizations

Most common spectral graph convolutional networks (such as e.g. He et al. (2021); Bianchi et al. (2019); Defferrard et al. (2016)) base the learnable filters that they propose on the symmetrically normalized graph Laplacian

$$
\mathscr{L} = Id - D^{-\frac{1}{2}} W D^{-\frac{1}{2}}.
$$

In the setting $S_{\text{high}} \gg S_{\text{reg.}}$ we are considering, this leads to an effective feature propagation along edges in $\mathcal{E}_{\text{high}}$ and $\mathcal{E}_{\text{low, exclusive}}$ (c.f. also Fig. 10) only, as Theorem A.3 below establishes:

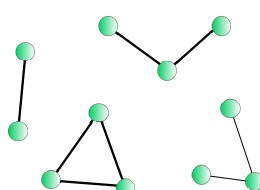

Figure 10: Limit graph corresponding to Fig 8 for symmetric normalization

**Theorem A.3.** With

$$
T = Id - D^{-\frac{1}{2}} W D^{-\frac{1}{2}},
$$

and the scale decomposition as introduced in Section 2, we have that

$$
\left\| T - \left( Id - D_{\text{high}}^{-\frac{1}{2}} W_{\text{high}} D_{\text{high}}^{-\frac{1}{2}} - D_{\text{reg.}}^{-\frac{1}{2}} W_{\text{low, exclusive}} D_{\text{reg.}}^{-\frac{1}{2}} \right) \right\| = \mathcal{O}\left( \sqrt{\frac{S_{\text{reg.}}}{S_{\text{high}}}} \right) \tag{12}
$$

for $S_{\text{high}} \gg S_{\text{reg.}}$.

*Proof.* We first note that instead of (12), we may equivalently establish

$$
\left\| D^{-\frac{1}{2}} W D^{-\frac{1}{2}} - \left( D_{\text{high}}^{-\frac{1}{2}} W_{\text{high}} D_{\text{high}}^{-\frac{1}{2}} + D_{\text{reg.}}^{-\frac{1}{2}} W_{\text{low, exclusive}} D_{\text{reg.}}^{-\frac{1}{2}} \right) \right\| = \mathcal{O}\left( \sqrt{\frac{S_{\text{reg.}}}{S_{\text{high}}}} \right).
$$

In Section 2, we already noted that

$$
W = W_{\text{high}} + W_{\text{reg.}}.
$$

With this, we may write

$$
D^{-\frac{1}{2}} W D^{-\frac{1}{2}} = D^{-\frac{1}{2}} W_{\text{high}} D^{-\frac{1}{2}} + D^{-\frac{1}{2}} W_{\text{reg.}} D^{-\frac{1}{2}}. \tag{13}
$$

Let us first examine the term $D^{-\frac{1}{2}} W_{\text{high}} D^{-\frac{1}{2}}$. We note for the corresponding matrix entries that

$$
\left( D^{-\frac{1}{2}} W_{\text{high}} D^{-\frac{1}{2}} \right)_{ij} = \frac{1}{\sqrt{d_i}} \cdot (W_{\text{high}})_{ij} \cdot \frac{1}{\sqrt{d_j}}
$$

Let us use the notation

$$
d_i^{\text{high}} = \sum_{j=1}^{N} (W_{\text{high}})_{ij}, \quad d_i^{\text{reg.}} = \sum_{j=1}^{N} (W_{\text{reg.}})_{ij} \text{ and } d_i^{\text{low,exclusive}} = \sum_{j=1}^{N} (W_{\text{low,exclusive}})_{ij}.
$$

We then find

$$\frac{1}{\sqrt{d_i}} = \frac{1}{\sqrt{d_i^{\text{high}}}} \cdot \frac{1}{\sqrt{1 + \frac{d_i^{\text{reg.}}}{d_i^{\text{high}}}}}$$

Using the Taylor expansion

$$\frac{1}{\sqrt{1 + \epsilon}} = 1 - \frac{1}{2}\epsilon + \mathcal{O}(\epsilon^2),$$

we thus have

$$\left(D^{-\frac{1}{2}} W_{\text{high}} D^{-\frac{1}{2}}\right)_{ij} = \frac{1}{\sqrt{d_i^{\text{high}}}} \cdot (W_{\text{high}})_{ij} \cdot \frac{1}{\sqrt{d_j^{\text{high}}}} + \mathcal{O}\left(\frac{d_i^{\text{reg.}}}{d_i^{\text{high}}}\right).$$

Since we have

$$\frac{d_i^{\text{reg.}}}{d_i^{\text{high}}} \propto \frac{S_{\text{reg.}}}{S_{\text{high}}},$$

this yields

$$D^{-\frac{1}{2}} W_{\text{high}} D^{-\frac{1}{2}} = D_{\text{high}}^{-\frac{1}{2}} W_{\text{high}} D_{\text{high}}^{-\frac{1}{2}} + \mathcal{O}\left(\frac{S_{\text{reg.}}}{S_{\text{high}}}\right).$$

Thus let us turn towards the second summand on the right-hand-side of (13). We have

$$\left(D^{-\frac{1}{2}} W_{\text{reg.}} D^{-\frac{1}{2}}\right)_{ij} = \frac{1}{\sqrt{d_i}} \cdot (W_{\text{reg.}})_{ij} \cdot \frac{1}{\sqrt{d_j}}.$$

Suppose that either $i$ or $j$ is not in $G_{\text{low, exclusive}}$. Without loss of generality (since the matrix under consideration is symmetric), assume $i \notin G_{\text{low, exclusive}}$, but $(W_{\text{reg.}})_{ij} \neq 0$. We may again write

$$\frac{1}{\sqrt{d_j}} = \frac{1}{\sqrt{d_j^{\text{high}}}} \cdot \frac{1}{\sqrt{1 + \frac{d_i^{\text{reg.}}}{d_i^{\text{high}}}}}.$$

Since

$$\frac{1}{\sqrt{1 + \frac{d_i^{\text{reg.}}}{d_i^{\text{high}}}}} \leqslant 1,$$

we have

$$\left|\left(D^{-\frac{1}{2}} W_{\text{reg.}} D^{-\frac{1}{2}}\right)_{ij}\right| \leqslant \left|\frac{1}{\sqrt{d_i}} \cdot (W_{\text{reg.}})_{ij}\right| \cdot \frac{1}{\sqrt{d_j^{\text{high}}}} = \mathcal{O}\left(\sqrt{\frac{S_{\text{reg.}}}{S_{\text{high}}}}\right).$$

If instead we have $i, j \in G_{\text{low, exclusive}}$, then clearly

$$\left(D^{-\frac{1}{2}} W_{\text{reg.}} D^{-\frac{1}{2}}\right)_{ij} = \left(D_{\text{reg.}}^{-\frac{1}{2}} W_{\text{low,exclusive}} D_{\text{reg.}}^{-\frac{1}{2}}\right)_{ij}.$$

Thus in total we have established

$$D^{-\frac{1}{2}} W D^{-\frac{1}{2}} = \left(D_{\text{high}}^{-\frac{1}{2}} W_{\text{high}} D_{\text{high}}^{-\frac{1}{2}} + D_{\text{reg.}}^{-\frac{1}{2}} W_{\text{low, exclusive}} D_{\text{reg.}}^{-\frac{1}{2}}\right) + \mathcal{O}\left(\frac{S_{\text{reg.}}}{S_{\text{high}}}\right)$$

which was to be established.

$\square$

Apart from networks that make use of the symmetrically normalized graph Laplacian $\mathscr{L}$, some methods, such as most notably Kipf & Welling (2017), instead base their filters on the operator

$$T = \tilde{D}^{-\frac{1}{2}} \tilde{W} \tilde{D}^{-\frac{1}{2}},$$

with

$$\tilde{W} = (W + Id)$$

and

$$\tilde{D} = D + Id.$$

In analogy to Theorem A.3, we here establish the limit propagation scheme determined by such operators:

**Theorem A.4.** With
$$T = \tilde{D}^{-\frac{1}{2}}\tilde{W}\tilde{D}^{-\frac{1}{2}},$$
where $\tilde{W} = (W + Id)$ and $\tilde{D} = D + Id$ as well as the scale decompositionof Section 2, we have that

$$\left\| T - \left( D_{\text{high}}^{-\frac{1}{2}} W_{\text{high}} D_{\text{high}}^{-\frac{1}{2}} + D_{\text{reg.}}^{-\frac{1}{2}} \tilde{W}_{\text{low, exclusive}} D_{\text{reg.}}^{-\frac{1}{2}} \right) \right\| = \mathcal{O}\left( \sqrt{\frac{S_{\text{reg.}} + 1}{S_{\text{high}}}} \right)$$

for $S_{\text{high}} \gg S_{\text{reg.}}$. Here $\tilde{W}_{\text{low, exclusive}}$ is given as

$$\tilde{W}_{\text{low, exclusive}} := W_{\text{low, exclusive}} + \text{diag}\left( \mathbb{1}_{G_{\text{low, exclusive}}} \right)$$

and $\mathbb{1}_{G_{\text{low, exclusive}}}$ denotes the vector whose entries are one for nodes in $G_{\text{low, exclusive}}$ and zero for all other nodes.

The difference to the result of Theorem A.3 is thus that applicability of the limit propagation scheme of Fig. 10 for the GCN Kipf & Welling (2017) is not only contingent upon $S_{\text{high}} \gg S_{\text{reg.}}$ but also $S_{\text{high}} \gg 1$.

*Proof.* To establish this – as in the proof of Theorem A.3 – we first decompose $T$:

$$\tilde{D}^{-\frac{1}{2}}\tilde{W}\tilde{D}^{-\frac{1}{2}} = \tilde{D}^{-\frac{1}{2}}W_{\text{high}}\tilde{D}^{-\frac{1}{2}} + \tilde{D}^{-\frac{1}{2}}W_{\text{reg.}}\tilde{D}^{-\frac{1}{2}} + \tilde{D}^{-\frac{1}{2}}Id\tilde{D}^{-\frac{1}{2}} \qquad (14)$$
$$= \tilde{D}^{-\frac{1}{2}}W_{\text{high}}\tilde{D}^{-\frac{1}{2}} + \tilde{D}^{-\frac{1}{2}}W_{\text{reg.}}\tilde{D}^{-\frac{1}{2}} + \tilde{D}^{-1}$$

For the first term, we note

$$\left( \tilde{D}^{-\frac{1}{2}}W_{\text{high}}\tilde{D}^{-\frac{1}{2}} \right)_{ij} = \frac{1}{\sqrt{d_i + 1}} \cdot (W_{\text{high}})_{ij} \cdot \frac{1}{\sqrt{d_j + 1}}.$$

We then find

$$\frac{1}{\sqrt{d_i + 1}} = \frac{1}{\sqrt{d_i^{\text{high}}}} \cdot \frac{1}{\sqrt{1 + \frac{d_i^{\text{reg.}} + 1}{d_i^{\text{high}}}}}.$$

Analogously to the proof of Theorem A.3, this yields

$$\left( \tilde{D}^{-\frac{1}{2}}W_{\text{high}}\tilde{D}^{-\frac{1}{2}} \right)_{ij} = \frac{1}{\sqrt{d_i^{\text{high}}}} \cdot (W_{\text{high}})_{ij} \cdot \frac{1}{\sqrt{d_j^{\text{high}}}} + \mathcal{O}\left( \frac{1 + d_i^{\text{reg.}}}{d_i^{\text{high}}} \right).$$

This implies

$$\tilde{D}^{-\frac{1}{2}}W_{\text{high}}\tilde{D}^{-\frac{1}{2}} = D_{\text{high}}^{-\frac{1}{2}}W_{\text{high}}D_{\text{high}}^{-\frac{1}{2}} + \mathcal{O}\left( \frac{S_{\text{reg.}} + 1}{S_{\text{high}}} \right).$$

Next we turn to the second summand in (14):

$$\left( \tilde{D}^{-\frac{1}{2}}W_{\text{reg.}}\tilde{D}^{-\frac{1}{2}} \right)_{ij} = \frac{1}{\sqrt{d_i + 1}} \cdot (W_{\text{reg.}})_{ij} \cdot \frac{1}{\sqrt{d_j + 1}}.$$

Suppose that either $i$ or $j$ is not in $G_{\text{low, exclusive}}$. Without loss of generality (since the matrix under consideration is symmetric), assume $i \notin G_{\text{low, exclusive}}$, but $(W_{\text{reg.}})_{ij} \neq 0$. We may again write

$$\frac{1}{\sqrt{d_j + 1}} = \frac{1}{\sqrt{d_j^{\text{high}}}} \cdot \frac{1}{\sqrt{1 + \frac{d_i^{\text{reg.}} + 1}{d_i^{\text{high}}}}}.$$

Since

$$\frac{1}{\sqrt{1 + \frac{d_i^{\text{reg.}} + 1}{d_i^{\text{high}}}}} \leqslant 1,$$

we have

$$
\left| \left( D^{-\frac{1}{2}} W_{\text{reg.}} D^{-\frac{1}{2}} \right)_{ij} \right| \leqslant \left| \frac{1}{\sqrt{1 + d_i}} \cdot (W_{\text{reg.}})_{ij} \right| \cdot \frac{1}{\sqrt{d_j^{\text{high}}}}
$$

$$
\leqslant \left| \frac{1}{\sqrt{d_i^{\text{reg.}}}} \cdot (W_{\text{reg.}})_{ij} \right| \cdot \frac{1}{\sqrt{d_j^{\text{high}}}}
$$

$$
= \mathcal{O}\left( \sqrt{\frac{S_{\text{reg.}}}{S_{\text{high}}}} \right).
$$

If instead we have $i, j \in G_{\text{low, exclusive}}$, then clearly

$$
\left( \tilde{D}^{-\frac{1}{2}} W_{\text{reg.}} \tilde{D}^{-\frac{1}{2}} \right)_{ij} = \left( \tilde{D}_{\text{reg.}}^{-\frac{1}{2}} W_{\text{low,exclusive}} \tilde{D}_{\text{reg.}}^{-\frac{1}{2}} \right)_{ij}.
$$

Finally we note for the third term on the right-hand-side of (14) that

$$
\frac{1}{d_i} \leqslant \frac{1}{d_i^{\text{high}}} = \mathcal{O}\left( \frac{1}{S_{\text{high}}} \right)
$$

if $i \notin G_{\text{low, exclusive}}$.

In total we thus have found

$$
\tilde{D}^{-\frac{1}{2}} \tilde{W} \tilde{D}^{-\frac{1}{2}} = \left( D_{\text{high}}^{-\frac{1}{2}} W_{\text{high}} D_{\text{high}}^{-\frac{1}{2}} + D_{\text{reg.}}^{-\frac{1}{2}} \tilde{W}_{\text{low, exclusive}} D_{\text{reg.}}^{-\frac{1}{2}} \right) + \mathcal{O}\left( \sqrt{\frac{S_{\text{reg.}} + 1}{S_{\text{high}}}} \right);
$$

which was to be proved. $\qquad\square$

## A.2 SPATIAL CONVOLUTIONAL FILTERS

Apart from spectral methods, there of course also exist methods that purely operate in the spatial domain of the graph. Such methods most often fall into the paradigm of message passing neural networks (MPNNs) Gilmer et al. (2017); Fey & Lenssen (2019): With $X_i^\ell \in \mathbb{R}^F$ denoting the features of node $i$ in layer $\ell$ and $w_{ij}$ denoting edge features, a message passing neural network may be described by the update rule (c.f. Gilmer et al. (2017))

$$
X_i^{\ell+1} = \gamma \left( X_i^\ell, \coprod_{j \in \mathcal{N}(i)} \phi \left( X_i^\ell, X_j^\ell, w_{ij} \right) \right). \tag{15}
$$

Here $\mathcal{N}(i)$ denotes the neighbourhood of node $i$, $\coprod$ denotes a differentiable and permutation invariant function (typically "sum", "mean" or "max") while $\gamma$ and $\phi$ denote differentiable functions such as multi-layer-perceptrons (MLPs) which might not be the same in each layer. Fey & Lenssen (2019).

Before we discuss corresponding limit-propagation schemes, we first establish that MPNNs are not able to reproduce the limit propagation scheme of Section 3 and are thus not stable to scale transitions and topological perturbations as discussed in Theorem 4.2 and Section 2.2.2.

### A.2.1 SCALE-SENSITIVITY OF MESSAGE PASSING NEURAL NETWORKS

As we established in Theorem 4.1 and Theorem 4.2 (c.f. also the corresponding proofs in Appendix D and Appendix E respectively), the stability to scale-variations (such as coarse-graining) of ResolvNets arises from the reliance on *resolvents* and the limit propagation scheme that they establish if separated weight-scales are present (c.f. Appendix B below).

Here we establish that message passing networks (as defined in (15) above) are unable to emulate this limit propagation scheme. Hence such architectures are also not stable to scale-changing topological perturbations such as coarse-graining procedures.

To this end, we consider a simple, fully connected graph $G$ on three nodes labeled 1, 2 and 3 (c.f. Fig. 11). We assume all node-weights to be equal to one ($\mu_i = 1$ for $i = 1, 2, 3$) and edge weights

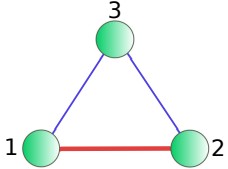

$$w_{13}, w_{23} \leqslant S_{\text{reg.}}$$

as well as

$$w_{12} = S_{\text{high}}.$$

Figure 11: Three node Graph $G$ with on large weight $w_{12} \gg 1$.

We now assume $S_{\text{high}} \gg S_{\text{reg.}}$.

Given states $\{X_1^\ell, X_2^\ell, X_3^\ell\}$ in layer $\ell$, the limit propagation scheme introduced in Section 3 would require the updated feature vector of node 3 to be given by

$$X_{3,\text{desired}}^{\ell+1} := \gamma \left( X_3^\ell, \phi \left( X_3^\ell, \frac{X_1^\ell + X_2^\ell}{2}, (w_{31} + w_{32}) \right) \right)$$

However, the actual updated feature at node 3 is given as (c.f. (15)):

$$X_{3,\text{actual}}^{\ell+1} := \gamma \left( X_3^\ell, \phi \left( X_3^\ell, X_1^\ell, w_{31} \right) \coprod \phi \left( X_3^\ell, X_2^\ell, w_{32} \right) \right) \tag{16}$$

Since there is no dependence on $S_{\text{high}}$ in equation (16) – which defines $X_{3,\text{actual}}^{\ell+1}$ – the desired propagation scheme can not arise, unless it is paradoxically already present at all scales $S_{\text{high}}$. If it is present at all scales, there is however only propagation along edges in $\underline{G}$, even if $S_{\text{high}} \approx S_{\text{reg.}}$, which would imply that the message passing network would not respect the graph structure of $G$. Hence $X_{3,\text{actual}}^{\ell+1} \not\to X_{3,\text{desired}}^{\ell+1}$ does not converge as $S_{\text{high}}$ increases.

### A.2.2 LIMIT PROPAGATION SCHEMES

The number of possible choices of message functions $\phi$, aggregation functions $\coprod$ and update functions $\gamma$ is clearly endless. Here we shall exemplarily discuss limit propagation schemes for two popular architectures: We first discuss the most general case where the message function $\phi$ is given as a learnable perceptron. Subsequently we assume that node features are updated with an attention-type mechanism.

**Generic message functions:** We first consider the possibility that the message function $\phi$ in (16) is implemented via an MLP using ReLU-activations: Assuming (for simplicity in notation) a one-hidden-layer MLP mapping features $X_i^\ell \in \mathbb{R}^{F_\ell}$ to features $X_i^{\ell+1} \in \mathbb{R}^{F_{\ell+1}}$ we have

$$\phi(X_i^\ell, X_j^\ell, w_{ij}) = \text{ReLU} \left( W_1^\ell \cdot X_i^\ell + W_2^\ell \cdot X_2^\ell + W_3^\ell \cdot w_{ij} + B^\ell \right)$$

with bias term $B^{\ell+1} \in \mathbb{R}^{F_{\ell+1}}$ and weight matrices $W_1^{\ell+1}, W_2^{\ell+1} \in \mathbb{R}^{F_{\ell+1} \times F_\ell}$ and $W_3^\ell \in \mathbb{R}^{F_{\ell+1}}$.

We will assume that the weight-vecor $W_3^{\ell+1}$ has no-nonzero entries. This is not a severe limitation experimentally and in fact generically justified: The complementary event of at-least one entry of $W_3$ being assigned precisely zero during training has probability weight zero (assuming an absolutely continuous probability distribtuion according to which weights are learned).

Let us now assume that the edge $(ij)$ belongs to $\mathcal{E}_{\text{high}}$ and the corresponding weight $w_{ij}$ is large ($w_{ij} \gg 1$). The behaviour of entries $\phi(X_i^\ell, X_j^\ell, w_{ij})_a$ of the message $\phi(X_i^\ell, X_j^\ell, w_{ij}) \in \mathbb{R}^{F_{\ell+1}}$ is then determined by the sign of the corresponding entry $\left( W_3^\ell \right)_a$ of the weight vector $W_3^\ell \in \mathbb{R}^{F_{\ell+1}}$:

If we have $\left( W_3^\ell \right)_a < 0$, then $\phi(X_i^\ell, X_j^\ell, w_{ij})_a$ approaches zero for larger edge-weights $w_{ij}$:

$$\lim_{w_{ij} \to \infty} \phi(X_i^\ell, X_j^\ell, w_{ij})_a = 0 \tag{17}$$

If we have $\left( W_3^\ell \right)_a > 0$, then $\phi(X_i^\ell, X_j^\ell, w_{ij})_a$ increasingly diverges for larger edge-weights $w_{ij}$:

$$\lim_{w_{ij} \to \infty} \phi(X_i^\ell, X_j^\ell, w_{ij})_a = \infty \tag{18}$$

For either choice of aggregation function $\coprod$ in (15) among "max", "sum" or "mean" the behaviour in (18) leads to unstable networks if the update function $\gamma$ is also given as an MLP with ReLU

activations. Apart from instabilities, we also make the following observation: If $S_{\text{high}} \gg S_{\text{reg.}}$, then by (18) and continuity of $\phi$ we can conclude that components $\phi(X_i^\ell, X_j^\ell, w_{ij})_a$ of messages propagated along $\mathcal{E}_{\text{high}}$ for which $(W_3^\ell)_a > 0$ dominate over messages propagated along edges in $\mathcal{E}_{\text{reg.}}$. By (17), the former clearly also dominate over components $\phi(X_i^\ell, X_j^\ell, w_{ij})_a$ of messages propagated along $\mathcal{E}_{\text{high}}$ for which $(W_3^\ell)_a < 0$. This behaviour is irrespective of whether "max", "sum" or "mean" aggregations are employed. Hence the limit propagation scheme essentially only takes into account message channels $\phi(X_i^\ell, X_j^\ell, w_{ij})_a$ for which $(ij) \in \mathcal{E}_{\text{high}}$ and $(W_3^\ell)_a > 0$.

Similar considerations apply, if non-linearities are chosen as leaky ReLU. If instead of ReLU activations a sigmoid-nonlinearity $\sigma$ like $\tanh$ is employed, messages propagated along $\mathcal{E}_{\text{large}}$ become increasingly uninformative, since they are progressively more independent of features $X_i^\ell$ and weights $w_{ij}$. Indeed, for sigmoid activations, the limits (17) and (18) are given as follows:

If we have $(W_3^\ell)_a < 0$, then we have for larger edge-weights $w_{ij}$ that

$$\lim_{w_{ij} \to \infty} \phi(X_i^\ell, X_j^\ell, w_{ij})_a = \lim_{y \to -\infty} \sigma(y).$$

If we have $(W_3^\ell)_a > 0$, then

$$\lim_{w_{ij} \to \infty} \phi(X_i^\ell, X_j^\ell, w_{ij})_a = \lim_{y \to \infty} \sigma(y).$$

In both cases, the messages $\phi(X_i^\ell, X_j^\ell, w_{ij})$ propagated along $\mathcal{E}_{\text{large}}$ become increasingly constant as the scale $S_{\text{high}}$ increases.

**Attention based messages:** Apart from general learnable message functions as above, we here also discuss an approach where edge weights are re-learned in an attention based manner. For this we modify the method Velickovic et al. (2018) to include edge weights. The resulting propagation scheme – with a single attention head for simplicity and a non-linearity $\rho$ – is given as

$$X_i^{\ell+1} = \rho \left( \sum_{j \in \mathcal{N}(i)} \alpha_{ij}(W X_j^{\ell+1}) \right).$$

Here we have $W \in \mathbb{R}^{F_{\ell+1} \times F_\ell}$ and

$$\alpha_{ij} = \frac{\exp\left( \text{LeakyRelu}\left( \vec{a}^\top \left[ W X_i^\ell \parallel W X_j^\ell \parallel w_{ij} \right] \right) \right)}{\sum\limits_{k \in \mathcal{N}(i)} \exp\left( \text{LeakyRelu}\left( \vec{a}^\top \left[ W X_i^\ell \parallel W X_k^\ell \parallel w_{ik} \right] \right) \right)}, \tag{19}$$

with $\parallel$ denoting concatenation. The weight vector $\vec{a} \in \mathbb{R}^{2F_{\ell+1}+1}$ is assumed to have a non zero entry in its last component. Otherwise, this attention mechanism would correspond to the one proposed in Velickovic et al. (2018), which does not take into account edge weights. Let us denote this entry of $\vec{a}$ ()determining attention on the weight $w_{ij}$) by $a_w$.

If $a_w < 0$, we have for $(i, j) \in \mathcal{E}_{\text{high}}$ that

$$\exp\left( \text{LeakyRelu}\left( \vec{a}^\top \left[ W X_i^\ell \parallel W X_j^\ell \parallel w_{ij} \right] \right) \right) \longrightarrow 0$$

as the weight $w_{ij}$ increases. Thus propagation along edges in $\mathcal{E}_{\text{high}}$ is essentially suppressed in this case.

If $a_w > 0$, we have for $(i, j) \in \mathcal{E}_{\text{high}}$ that

$$\exp\left( \text{LeakyRelu}\left( \vec{a}^\top \left[ W X_i^\ell \parallel W X_j^\ell \parallel w_{ij} \right] \right) \right) \longrightarrow \infty$$

as the weight $w_{ij}$ increases. Thus for edges $(i, j) \in \mathcal{E}_{\text{reg.}}$ (i.e. those that are *not* in $\mathcal{E}_{\text{high}}$), we have

$$\alpha_{ij} \to 0,$$

since the denominator in (19) diverges. Hence in this case, propagation along $\mathcal{E}_{\text{reg.}}$ is essentially suppressed and features are effectively only propagated along $\mathcal{E}_{\text{high}}$.

## B    PROOF OF THEOREM 3.3

In this section, we prove Theorem 3.3. For convenience, we first restate the result – together with the definitions leading up to it – again:

**Definition B.1.** Denote by $\mathcal{G}$ the set of connected components in $G_{\text{high}}$. We give this set a graph structure as follows: Let $R$ and $P$ be elements of $\underline{\mathcal{G}}$ (i.e. connected components in $G_{\text{high}}$). We define the real number

$$\underline{W}_{RP} = \sum_{r \in R} \sum_{p \in P} W_{rp},$$

with $r$ and $p$ nodes in the original graph $G$. We define the set of edges $\underline{\mathcal{E}}$ on $\underline{G}$ as

$$\underline{\mathcal{E}} = \{(R, P) \in \underline{\mathcal{G}} \times \underline{\mathcal{G}} : \underline{W}_{RP} > 0\}$$

and assign $\underline{W}_{RP}$ as weight to such edges. Node weights of limit nodes are defined similarly as aggregated weights of all nodes $r$ (in $G$) contained in the component $R$ as

$$\underline{\mu}_R = \sum_{r \in R} \mu_r.$$

In order to translate signals between the original graph $G$ and the limit description $\underline{G}$, we need translation operators mapping signals from one graph to the other:

**Definition B.2.** Denote by $\mathbb{1}_R$ the vector that has $1$ as entries on nodes $r$ belonging to the connected (in $G_{\text{hign}}$) component $R$ and has entry zero for all nodes not in $R$. We define the down-projection operator $J^{\downarrow}$ component-wise via evaluating at node $R$ in $\underline{\mathcal{G}}$ as

$$(J^{\downarrow}x)_R = \langle \mathbb{1}_R, x \rangle / \underline{\mu}_R.$$

The upsampling operator $J^{\uparrow}$ is defined as

$$J^{\uparrow}u = \sum_R u_R \cdot \mathbb{1}_R; \tag{20}$$

where $u_R$ is a scalar value (the component entry of $u$ at $R \in \underline{\mathcal{G}}$) and the sum is taken over all connected components in $G_{\text{high}}$.

The result we then have to prove is the following:

**Theorem B.3.** We have

$$\left\| R_z(\Delta) - J^{\uparrow} R_z(\underline{\Delta}) J^{\downarrow} \right\| = \mathcal{O}\left( \frac{\|\Delta_{\text{reg.}}\|}{\lambda_1(\Delta_{\text{high}})} \right)$$

holds; with $\lambda_1(\Delta_{\text{high}})$ denoting the first non-zero eigenvalue of $\Delta_{\text{high}}$.

Note that this then indeed proves Theorem 3.3, since we have

$$\lambda_{\max}(\Delta_{\text{reg.}}) = \|\Delta_{\text{reg.}}\|.$$

*Proof.* We will split the proof of this result into multiple steps. For $z < 0$ Let us denote by

$$R_z(\Delta) = (\Delta - zId)^{-1},$$
$$R_z(\Delta_{high}) = (\Delta_{high} - zId)^{-1}$$
$$R_z(\Delta_{reg.}) = (\Delta_{reg.} - zId)^{-1}$$

the resolvents correspodning to $\Delta$, $\Delta_{high}$ and $\Delta_{reg.}$ respectively.
Our first goal is establishing that we may write

$$R_z(\Delta) = [Id + R_z(\Delta_{high})\Delta_{reg.}]^{-1} \cdot R_z(\Delta_{high})$$

This will follow as a consequence of what is called the second resolvent formula Teschl (2014):

"Given self-adjoint operators $A$, $B$, we may write

$$R_z(A + B) - R_z(A) = -R_z(A)BR_z(A + B)."$$

In our case, this translates to

$$R_z(\Delta) - R_z(\Delta_{high}) = -R_z(\Delta_{high})\Delta_{\text{reg.}}R_z(\Delta)$$

or equivalently

$$[Id + R_z(\Delta_{high})\Delta_{\text{reg.}}]\, R_z(\Delta) = R_z(\Delta_{high}).$$

Multiplying with $[Id + R_z(\Delta_{high})\Delta_{\text{reg.}}]^{-1}$ from the left then yields

$$R_z(\Delta) = [Id + R_z(\Delta_{high})\Delta_{reg.}]^{-1} \cdot R_z(\Delta_{high})$$

as desired.

Hence we need to establish that $[Id + R_z(\Delta_{high})\Delta_{reg.}]$ is invertible for $z < 0$.

To establish a contradiction, assume it is not invertible. Then there is a signal $x$ such that

$$[Id + R_z(\Delta_{high})\Delta_{reg.}]\, x = 0.$$

Multiplying with $(\Delta_{\text{high}} - zId)$ from the left yields

$$(\Delta_{\text{high}} + \Delta_{\text{reg.}} - zId)x = 0$$

which is precisely to say that

$$(\Delta - zId)x = 0$$

But since $\Delta$ is a graph Laplacian, it only has non-negative eigenvalues. Hence we have reached our contradiction and established

$$R_z(\Delta) = [Id + R_z(\Delta_{high})\Delta_{reg.}]^{-1} R_z(\Delta_{high}).$$

Our next step is to establish that

$$R_z(\Delta_{high}) \to \frac{P_0^{\text{high}}}{-z},$$

where $P_0^{\text{high}}$ is the spectral projection onto the eigenspace corresponding to the lowest lying eigenvalue $\lambda_0(\Delta_{high}) = 0$ of $\Delta_{high}$. Indeed, by the spectral theorem for finite dimensional operators (c.f. e.g. Teschl (2014)), we may write

$$R_z(\Delta_{high}) \equiv (\Delta_{high} - zId)^{-1} = \sum_{\lambda \in \sigma(\Delta_{high})} \frac{1}{\lambda - z} \cdot P_\lambda^{high}.$$

Here $\sigma(\Delta_{high})$ denotes the spectrum (i.e. the collection of eigenvalues) of $\Delta_{high}$ and the $\{P_\lambda^{high}\}_{\lambda \in \sigma(\Delta_{high})}$ are the corresponding (orthogonal) eigenprojections onto the eigenspaces of the respective eigenvalues. Thus we find

$$\left\| R_z(\Delta_{high}) - \frac{P_0^{high}}{-z} \right\| = \left| \sum_{0 < \lambda \in \sigma(\Delta_{high})} \frac{1}{\lambda - z} \cdot P_\lambda^{high} \right\|;$$

where the sum on the right hand side now excludes the eigenvalue $\lambda = 0$.

Using orthonormality of the spectral projections, the fact that $z < 0$ and monotonicity of $1/(\cdot + |z|)$ we find

$$\left\| R_z(\Delta_{high}) - \frac{P_0^{high}}{-z} \right\| = \frac{1}{\lambda_1(\Delta_{high}) + |z|}.$$

Here $\lambda_1(\Delta_{high})$ is the firt non-zero eigenvalue of $(\Delta_{high})$.

Non-zero eigenvalues scale linearly with the weight scale since we have

$$\lambda(S \cdot \Delta) = S \cdot \lambda(\Delta)$$

for any graph Laplacian (in fact any matrix) $\Delta$ with eigenvalue $\lambda$. Thus we have

$$\left\| R_z(\Delta_{high}) - \frac{P_0^{high}}{-z} \right\| = \frac{1}{\lambda_1(\Delta_{high}) + |z|} \leqslant \frac{1}{\lambda_1(\Delta_{high})} \longrightarrow 0$$

as $\lambda_1(\Delta_{high}) \to \infty$.

Our next task is to use this result in order to bound the difference

$$I := \left\| \left[ Id + \frac{P_0^{high}}{-z} \Delta_{reg.} \right]^{-1} \frac{P_0^{high}}{-z} - [Id + R_z(\Delta_{high})\Delta_{reg.}]^{-1} R_z(\Delta_{high}) \right\|.$$

To this end we first note that the relation

$$[A + B - zId]^{-1} = [Id + R_z(A)B]^{-1} R_z(A)$$

provided to us by the second resolvent formula, implies

$$[Id + R_z(A)B]^{-1} = Id - B[A + B - zId]^{-1}.$$

Thus we have

$$\left\| [Id + R_z(\Delta_{high})\Delta_{reg.}]^{-1} \right\| \leqslant 1 + \|\Delta_{reg.}\| \cdot \|R_z(\Delta)\|$$

$$\leqslant 1 + \frac{\|\Delta_{reg.}\|}{|z|}.$$

With this, we have

$$\left\| \left[ Id + \frac{P_0^{high}}{-z} \Delta_{reg.} \right]^{-1} \cdot \frac{P_0^{high}}{-z} - R_z(\Delta) \right\|$$

$$= \left\| \left[ Id + \frac{P_0^{high}}{-z} \Delta_{reg.} \right]^{-1} \cdot \frac{P_0^{high}}{-z} - [Id + R_z(\Delta_{high})\Delta_{reg.}]^{-1} \cdot R_z(\Delta_{high}) \right\|$$

$$\leqslant \left\| \frac{P_0^{high}}{-z} \right\| \cdot \left\| \left[ Id + \frac{P_0^{high}}{-z} \Delta_{reg.} \right]^{-1} - [Id + R_z(\Delta_{high})\Delta_{reg.}]^{-1} \right\| + \left\| \frac{P_0^{high}}{-z} - R_z(\Delta_{high}) \right\| \cdot \left\| [Id + R_z(\Delta_{high})\Delta_{reg.}]^{-1} \right\|$$

$$\leqslant \frac{1}{|z|} \left\| \left[ Id + \frac{P_0^{high}}{-z} \Delta_{reg.} \right]^{-1} - [Id + R_z(\Delta_{high})\Delta_{reg.}]^{-1} \right\| + \left( 1 + \frac{\|\Delta_{reg.}\|}{|z|} \right) \cdot \frac{1}{\lambda_1(\Delta_{high})}.$$

Hence it remains to bound the left hand summand. For this we use the following fact (c.f. Horn & Johnson (2012), Section 5.8. "Condition numbers: inverses and linear systems"):

Given square matrices $A, B, C$ with $C = B - A$ and $\|A^{-1}C\| < 1$, we have

$$\|A^{-1} - B^{-1}\| \leqslant \frac{\|A^{-1}\| \cdot \|A^{-1}C\|}{1 - \|A^{-1}C\|}.$$

In our case, this yields (together with $\|P_0^{high}\| = 1$) that

$$\left\| \left[ Id + P_0^{high}/(-z) \cdot \Delta_{reg.} \right]^{-1} - [Id + R_z(\Delta_{high})\Delta_{reg.}]^{-1} \right\|$$

$$\leqslant \frac{(1 + \|\Delta_{reg.}\|/|z|)^2 \cdot \|\Delta_{reg.}\| \cdot \|\frac{P_0^{high}}{-z} - R_z(\Delta_{high})\|}{1 - (1 + \|\Delta_{reg.}\|/|z|) \cdot \|\Delta_{reg.}\| \cdot \|\frac{P_0^{high}}{-z} - R_z(\Delta_{high})\|}$$

For $S_{high}$ sufficiently large, we have

$$\| - P_0^{high}/z - R_z(\Delta_{high})\| \leqslant \frac{1}{2(1 + \|\Delta_{reg.}\|/|z|)}$$

so that we may estimate

$$\left\| \left[ Id + \Delta_{reg.} \frac{P_0^{high}}{-z} \right]^{-1} - [Id + \Delta_{reg.} R_z(\Delta_{high})]^{-1} \right\|$$

$$\leqslant 2 \cdot (1 + \|\Delta_{\text{reg.}}\|) \cdot \left\| \frac{P_0^{high}}{-z} - R_z(\Delta_{\text{high}}) \right\|$$

$$= 2 \frac{1 + \|\Delta_{\text{reg.}}\|/|z|}{\lambda_1(\Delta_{\text{high}})}$$

Thus we have now established

$$\left| \left[ Id + \frac{P_0^{high}}{-z} \Delta_{reg.} \right]^{-1} \cdot \frac{P_0^{high}}{-z} - R_z(\Delta) \right| = \mathcal{O}\left( \frac{\|\Delta_{\text{reg.}}\|}{\lambda_1(\Delta_{\text{high}})} \right).$$

Hence we are done with the proof, as soon as we can establish

$$\left[ -zId + P_0^{high} \Delta_{reg.} \right]^{-1} P_0^{high} = J^\uparrow R_z(\underline{\Delta}) J^\downarrow,$$

with $J^\uparrow, \underline{\Delta}, J^\downarrow$ as defined above. To this end, we first note that

$$J^\uparrow \cdot J^\downarrow = P_0^{high} \tag{21}$$

and

$$J^\downarrow \cdot J^\uparrow = Id_{\underline{G}}. \tag{22}$$

Indeed, the relation (21) follows from the fact that the eigenspace corresponding to the eignvalue zero is spanned by the vectors $\{\mathbb{1}_R\}_R$, with $\{R\}$ the connected components of $G_{\text{high}}$. Equation (22) follows from the fact that

$$\langle \mathbb{1}_R, \mathbb{1}_R \rangle = \underline{\mu}_R.$$

With this we have

$$\left[ Id + P_0^{high} \Delta_{reg.} \right]^{-1} P_0^{high} = \left[ Id + J^\uparrow J^\downarrow \Delta_{reg.} \right]^{-1} J^\uparrow J^\downarrow.$$

To proceed, set

$$\underline{x} := F^\downarrow x$$

and

$$\mathscr{X} = \left[ P_0^{high} \Delta_{reg.} - zId \right]^{-1} P_0^{high} x.$$

Then

$$\left[ P_0^{high} \Delta_{reg.} - zId \right] \mathscr{X} = P_0^{high} x$$

and hence $\mathscr{X} \in \text{Ran}(P_0^{high})$. Thus we have

$$J^\uparrow J^\downarrow (\Delta_{\text{reg.}} - zId) J^\uparrow J^\downarrow \mathscr{X} = J^\uparrow J^\downarrow x.$$

Multiplying with $J^\downarrow$ from the left yields

$$J^\downarrow (\Delta_{\text{reg.}} - zId) J^\uparrow J^\downarrow \mathscr{X} = J^\downarrow x.$$

Thus we have

$$(J^\downarrow \Delta_{\text{reg.}} J^\uparrow - zId) J^\uparrow J^\downarrow \mathscr{X} = J^\downarrow x.$$

This – in turn – implies

$$J^\uparrow J^\downarrow \mathscr{X} = \left[ J^\downarrow \Delta_{\text{reg.}} J^\uparrow - zId \right]^{-1} J^\downarrow x.$$

Using

$$P_0^{high} \mathscr{X} = \mathscr{X},$$

we then have

$$\mathscr{X} = J^\uparrow \left[ J^\downarrow \Delta_{\text{reg.}} J^\uparrow - zId \right]^{-1} J^\downarrow x.$$

We have thus concluded the proof if we can prove that $J^\downarrow \Delta_{\text{reg.}} J^\uparrow$ is the Laplacian corresponding to the graph $\underline{G}$ defined in Definition B.1. But this is a straightforward calculation. $\qquad \square$

As a corollary, we find

**Corollary B.4.** We have

$$R_z(\Delta)^k \to J^\uparrow R^k(\underline{\Delta}) J^\downarrow$$

*Proof.* This follows directly from the fact that

$$J^\downarrow J^\uparrow = Id_{\underline{G}}.$$

$\square$

## C  PROOF OF THEOREM 3.4

Here we prove Theorem 3.4, which we restate for convenience:

**Theorem C.1.** Fix $\epsilon > 0$ and $z < 0$. For arbitrary functions $g, h : [0, \infty] \to \mathbb{R}$ with $\lim_{\lambda \to \infty} g(\lambda) =$ const. and $\lim_{\lambda \to \infty} h(\lambda) = 0$, there are filters $f^0_{z,\theta}, f^I_{z,\theta}$ of Type-0 and Type-I respectively such that $\|f^0_{z,\theta} - g\|_\infty, \|f^I_{z,\theta} - h\|_\infty < \epsilon$.

*Proof.* The Stone-Weierstrass theorem (see e.g. Teschl (2014)) states that any sub-algebra of continuous functions that are constant at infinity is already dense (in the topoloogy of uniform convergence) if this sub-algebra separates points.

Thus – using the Stone-Weierstrass Theorem – all we have to prove to establish the claim is that for every pair of points $x, y \geqslant 0$ there is a function $f_\theta$ with

$$f_\theta(x) \neq f_\theta(y).$$

But this is clear since (for $z < 0$) the function

$$\frac{1}{\cdot - z} : [0, \infty) \longrightarrow \mathbb{R}$$

(which generates the algebra of functions we consider) is already everywhere defined and injective.

$\square$

## D  STABILITY THEORY

Here we provide stability results to input- and edge-weight- perturbations for our architecture. For convenience, we restate our layer-wise update rule here again:

Given a feature matrix $X^\ell \in \mathbb{R}^{N \times F_\ell}$ in layer $\ell$, with column vectors $\{X^\ell_j\}_{j=1}^{F_\ell}$, the feature vector $X^{\ell+1}_i$ in layer $\ell+1$ is calculated as $X^{\ell+1}_i = \text{ReLU}\left(\sum_{j=1}^{F_{\ell+1}} f_{z,\theta^{\ell+1}_{ij}}(\Delta) \cdot X^\ell_j + b^{\ell+1}_i\right)$ with a learnable bias vector $b^{\ell+1}_i$. Collecting biases into a matrix $B^{\ell+1} \in \mathbb{R}^{F_{\ell+1} \times N}$, we efficiently implement this using matrix-multiplications as

$$X^{\ell+1} = \text{ReLU}\left(\sum_{k=a}^{K} (T - \omega Id)^{-k} \cdot X^\ell \cdot W^{\ell+1}_k + B^{\ell+1}\right)$$

with weight matrices $\{W^{\ell+1}_k\}$ in $\mathbb{R}^{F_\ell \times F_{\ell+1}}$. Biases are implemented as $b_i = \beta_i \cdot \mathbb{1}_G$, with $\mathbb{1}_G$ the vector of all ones on $G$ and $\beta_i \in \mathbb{R}$ learnable.

Our first result main-body of the paper then concerns stability to perturbations of input signals:

**Theorem D.1.** Let $\Phi_L$ be the map associated to an $L$-layer deep ResolvNet. Denote the collection of weight matrices in layer $\ell$ by $\mathscr{W}^\ell := \{W_k\}_{K=a}^{K_\ell}$. We have

$$\|\Phi_L(X) - \Phi_L(Y)\|_2 \leqslant \|X - Y\|_2 \cdot \prod_{\ell=1}^{L} \|\mathscr{W}^\ell\|_z, \tag{23}$$

with

$$\|\mathscr{W}^\ell\|_z := \sum_{k=a}^{K} \frac{1}{|z|^k} \|W^\ell_k\|$$

aggregating singular values of weight matrices.

*Proof.* Let us denote (hidden) feature matrices in layer $\ell$ by $X^\ell$ (resp. $Y^\ell$).

We note the following:

$$\|X^L - Y^L\| = \left\| \text{ReLU}\left( \sum_{k=a}^{K} R_z^k(\Delta) X^{L-1} W_k^L + B^L \right) - \text{ReLU}\left( \sum_{k=a}^{K} R_z^k(\Delta) Y^{L-1} W_k^L + B^L \right) \right\|$$

$$\leqslant \left\| \left( \sum_{k=a}^{K} R_z^k(\Delta) X^{L-1} W_k^L + B^L \right) - \left( \sum_{k=a}^{K} R_z^k(\Delta) Y^{L-1} W_k^L + B^L \right) \right\|$$

$$\leqslant \left\| \sum_{k=a}^{K} R_z^k(\Delta) X^{L-1} W_k - \sum_{k=a}^{K} R_z^k(\Delta) Y^{L-1} W_k^L \right\|$$

$$\leqslant \sum_{k=a}^{K} \left\| R_z^k(\Delta) \right\| \cdot \left\| X^{L-1} - Y^{L-1} \right\| \cdot \left\| W_k^L \right\|$$

$$= \sum_{k=a}^{K} \frac{1}{|z|^k} \cdot \left\| X^{L-1} - Y^{L-1} \right\| \cdot \left\| W_k^L \right\|$$

$$\leqslant \left\| \mathscr{W}^L \right\|_z \cdot \left\| X^{L-1} - Y^{L-1} \right\|.$$

Iterating through the layers yields the desired inequality (23). $\qquad\square$

In preparation for our next result – Theorem D.5 below – we note the following:

**Lemma D.2.** Let $\Phi_L$ be the map associated to an $L$-layer deep ResolvNet. With weights and biases denoted as above, we have

$$\|\Phi_L(X)\| \leqslant \|B^L\| + \sum_{m=0}^{L} \left( \prod_{j=0}^{m} \|\mathscr{W}^{L-1-j}\|_z \right) \|B^{L-1-j}\| + \left( \prod_{\ell=1}^{L} \|\mathscr{W}^\ell\|_z \right) \cdot \|X\|_2 \qquad (24)$$

*Proof.* We have

$$\|X\|^L \leqslant \left\| \text{ReLU}\left( \sum_{k=a}^{K} R_z^k(\Delta) X^{L-1} W_k + B^L \right) \right\|$$

$$\leqslant \left\| \sum_{k=a}^{K} R_z^k(\Delta) X^{L-1} W_k^L + B^L \right\|$$

$$\leqslant \left\| \sum_{k=a}^{K} R_z^k(\Delta) X^{L-1} W_k^L \right\| + \left\| B^L \right\|$$

$$\leqslant \sum_{k=a}^{K} \|R_z^k(\Delta)\| \cdot \|X^{L-1}\| \cdot \|W_k^L\| + \|B^L\|$$

$$\leqslant \left( \sum_{k=a}^{K} \frac{\|W_k^L\|}{|z|^k} \right) \cdot \|X^{L-1}\| + \|B^L\|.$$

Iterating this through all layers, we obtain (24). $\qquad\square$

Before we can establish Theorem D.5 below, we need two additional (related) preliminary results:

**Lemma D.3.** Let us use the notation $\widetilde{R}_z := (\widetilde{\Delta} - zId)^{-1}$ and $R_z := (\Delta - zId)^{-1}$ for resulvents corresponding to two different Laplacians $\Delta$ and $\widetilde{\Delta}$. We have

$$\|R_z - \widetilde{R}_z\| \leqslant \frac{1}{|z|^3} \|\Delta - \widetilde{\Delta}\|$$

*Proof.* Let $T$ and $\widetilde{T}$ be (finite dimensional) operators. Choose $z$ so that it is neither an eigenvalue of $T$ nor $\widetilde{T}$.

To showcase the principles underlying the proof, let us use the notation

$$R_z(T) \equiv \frac{1}{T-z}.$$

We note the following

$$
\begin{aligned}
&\frac{1}{\widetilde{T}-z}(\widetilde{T}-T)\frac{1}{T-z} \\
=&\frac{1}{\widetilde{T}-z}\widetilde{T}\frac{1}{T-z}-\frac{1}{\widetilde{T}-z}T\frac{1}{T-z} \\
=&\left[\frac{1}{\widetilde{T}-z}(\widetilde{T}-z)+\frac{z}{\widetilde{T}-z}\right]\frac{1}{T-z}-\frac{1}{\widetilde{T}-z}\left[\frac{1}{T-z}(T-z)+\frac{z}{T-z}\right] \\
=&z\left(\frac{1}{T-z}-\frac{1}{\widetilde{T}-z}\right).
\end{aligned}
$$

Rearranging and using

$$\|R_z(\Delta)\| = \|R_z(\widetilde{(\Delta)})\| = \frac{1}{|z|}$$

together with the sub-multiplicativity of the operator-norm $\|\cdot\|$ yields the claim. $\qquad\square$

We also note the following estimate on differences of powers of resolvents:

**Lemma D.4.** Let $\widetilde{R}_z := (\widetilde{\Delta} - zId)^{-1}$ and $R_z := (\Delta - zId)^{-1}$. For any natural number $k$, we have

$$\|\widetilde{R}_z^k - R_z^k\| \leqslant \frac{k}{|z|^{k-1}}\|\widetilde{R}_z - R_z\|$$

*Proof.* We note that for arbitrary matrices $T, \widetilde{T}$, we have

$$
\begin{aligned}
\widetilde{T}^k - T^k &= \widetilde{T}^{k-1}(\widetilde{T}-T)+(\widetilde{T}^{k-1}-T^{k-1})T \\
&=\widetilde{T}^{k-1}(\widetilde{T}-T)+\widetilde{T}^{k-2}(\widetilde{T}-T)T+(\widetilde{T}^{k-2}-T^{k-2})T^2.
\end{aligned}
$$

Iterating this and using

$$\|R_z(\Delta)\| = \|R_z(\widetilde{\Delta})\| = \frac{1}{|z|}$$

for $z < 0$ then yields the claim. $\qquad\square$

Having established the preceding lemmata, we can now establish stability to perturbations of the edge weights:

**Theorem D.5.** Let $\Phi_L$ and $\widetilde{\Phi}_L$ be the maps associated to ResolvNets with the same network architecture, but based on Laplacians $\Delta$ and $\widetilde{\Delta}$ respectively. We have

$$\|\Phi_L(X) - \widetilde{\Phi}_L(X)\|_2 \leqslant (C_1(\mathscr{W}) \cdot \|X\|_2 + C_2(\mathscr{W}, \mathscr{B})) \cdot \|\Delta - \widetilde{\Delta}\|. \tag{25}$$

Here, the stability constants $C_1(\mathscr{W})$ and $C_2(\mathscr{W}, \mathscr{B})$ are polynomials in (the largest) singular values of weight matrices and weight matrices as well as bias matrices, respectively.

*Proof.* Denote by $X^\ell$ and $\widetilde{X}^\ell$ the (hidden) feature matrices generated in layer $\ell$ for networks based on Laplacians $\Delta$ and $\widetilde{\Delta}$ respectively: I.e. we have

$$X^\ell = \mathrm{ReLU}\left(\sum_{k=a}^{K} R_z^k(\Delta)X^{\ell-1}W_k + B^\ell\right)$$

and

$$\widetilde{X}^\ell = \mathrm{ReLU}\left(\sum_{k=a}^{K} R_z^k(\widetilde{\Delta})\widetilde{X}^{\ell-1}W_k + B^\ell\right).$$

Using the fact that $\mathrm{ReLU}(\cdot)$ is Lipschitz continuous with Lipschitz constant $D = 1$, we have

$$\|X^L - \widetilde{X}^L\|$$
$$= \left\|\mathrm{ReLU}\left(\sum_{k=a}^{K} R_z^k(\Delta)X^{L-1}W_k^L + B^L\right) - \mathrm{ReLU}\left(\sum_{k=a}^{K} R_z^k(\widetilde{\Delta})\widetilde{X}^{L-1}W_k^L + B^L\right)\right\|$$
$$\leqslant \left\|\left(\sum_{k=a}^{K} R_z^k(\Delta)X^{L-1}W_k^L + B^L\right) - \left(\sum_{k=a}^{K} R_z^k(\widetilde{\Delta})\widetilde{X}^{L-1}W_k^L + B^L\right)\right\|$$
$$\leqslant \left\|\sum_{k=a}^{K} R_z^k(\Delta)X^{L-1}W_k^L - \sum_{k=a}^{K} R_z^k(\widetilde{\Delta})\widetilde{X}^{L-1}W_k^L\right\|$$
$$\leqslant \left\|\sum_{k=a}^{K} (R_z^k(\Delta) - R_z^k(\widetilde{\Delta}))X^{L-1}W_k^L\right\| + \sum_{k=a}^{K} \|R_z(\widetilde{\Delta})\| \cdot \|\widetilde{X}^{L-1} - X^{L-1}\| \cdot \|W_k^L\|$$
$$\leqslant \left\|\sum_{k=a}^{K} (R_z^k(\Delta) - R_z^k(\widetilde{\Delta}))X^{L-1}W_k^L\right\| + \|\mathscr{W}^L\|_z \cdot \|\widetilde{X}^{L-1} - X^{L-1}\|$$
$$\leqslant \sum_{k=a}^{K} \left\|R_z^k(\Delta) - R_z^k(\widetilde{\Delta})\right\| \cdot \left\|X^{L-1}\right\| \cdot \left\|W_k^L\right\| + \|\mathscr{W}^L\|_z \cdot \|\widetilde{X}^{L-1} - X^{L-1}\|$$

Applying Lemma D.4 yields

$$\|X^L - \widetilde{X}^L\|$$
$$\leqslant \left(\sum_{k=a}^{K} \frac{k}{|z|^{k-1}} \left\|W_k^L\right\|\right) \cdot \left\|X^{L-1}\right\| \cdot \left\|R_z(\Delta) - R_z(\widetilde{\Delta})\right\| + \|\mathscr{W}^L\|_z \cdot \|\widetilde{X}^{L-1} - X^{L-1}\|.$$

Using Lemma D.3, we then have

$$\|X^L - \widetilde{X}^L\|$$
$$\leqslant \left(\sum_{k=a}^{K} \frac{k}{|z|^{k+2}} \left\|W_k^L\right\|\right) \cdot \left\|X^{L-1}\right\| \cdot \left\|\Delta - \widetilde{\Delta}\right\| + \|\mathscr{W}^L\|_z \cdot \|\widetilde{X}^{L-1} - X^{L-1}\|.$$

Lemma D.2 then yields

$$\|X^L - \widetilde{X}^L\|$$
$$\leqslant \left(\sum_{k=a}^{K} \frac{k}{|z|^{k+2}} \left\|W_k^L\right\|\right) \cdot$$
$$\cdot \left[\|B^L\| + \sum_{m=0}^{L} \left(\prod_{j=0}^{m} \|\mathscr{W}^{L-1-k}\|_z\right) \|B^{L-1-k}\| + \left(\prod_{\ell=1}^{L} \|\mathscr{W}^\ell\|_z\right) \cdot \|X\|_2\right] \cdot \|\widetilde{\Delta} - \Delta\|$$
$$+ \|\mathscr{W}^L\|_z \cdot \|\widetilde{X}^{L-1} - X^{L-1}\|.$$

Iterating this through the layers and collecting summands yields the desired relation (25). □

# E  STABILITY UNDER SCALE VARIATIONS

Here we provide details on the scale-invariance results discussed in Section 4.

In preparation, we will first need to prove a lemma relating powers of resolvents on the original graph $G$ and its limit-description $\underline{G}$:

**Lemma E.1.** Let $\underline{R}_z := (\underline{\Delta} - zId)^{-1}$ and $R_z := (\Delta - zId)^{-1}$. For any natural number $k$, we have

$$\|J^\uparrow \underline{R}_z^k J^\downarrow - R_z^k\| \leqslant \frac{k}{|z|^{k-1}} \|J^\uparrow \underline{R}_z J^\downarrow - R_z\|$$

The proof proceeds in analogy to that of Lemma D.4:

*Proof.* We note that for arbitrary matrices $T, \widetilde{T}$, we have

$$\widetilde{T}^k - T^k = \widetilde{T}^{k-1}(\widetilde{T} - T) + (\widetilde{T}^{k-1} - T^{k-1})T$$
$$= \widetilde{T}^{k-1}(\widetilde{T} - T) + \widetilde{T}^{k-2}(\widetilde{T} - T)T + (\widetilde{T}^{k-2} - T^{k-2})T^2.$$

Iterating this, using

$$\|R_z(\Delta)\| = \|J^\uparrow R_z(\underline{\Delta})J^\downarrow\| = \frac{1}{|z|}$$

for $z < 0$ together with $\|J^\uparrow\|, \|J^\downarrow\| \leqslant 1$ and

$$J^\uparrow \underline{R}_z^k J^\downarrow = \left(J^\uparrow \underline{R}_z J^\downarrow\right)^k$$

(which holds since $J^\downarrow J^\uparrow = Id_{\underline{G}}$) then yields the claim.

Note that the equation

$$\|J^\uparrow R_z(\underline{\Delta})J^\downarrow\| = \frac{1}{|z|}$$

holds, because we may write

$$\|J^\uparrow R_z(\underline{\Delta})J^\downarrow\| = \|\lim_{\lambda_1(\Delta_{\text{high}}) \to \infty} R_z(\Delta)\| = \lim_{\lambda_1(\Delta_{\text{high}}) \to \infty} \|R_z(\Delta)\| = \lim_{\lambda_1(\Delta_{\text{high}}) \to \infty} \frac{1}{|z|} = \frac{1}{|z|}.$$

$\square$

Hence let us now prove Stability-Theorem 4.1, which we restate here for convenience:

**Theorem E.2.** Let $\Phi_L$ and $\underline{\Phi}_L$ be the maps associated to ResolvNets with the same learned weight matrices and biases but deployed on graphs $G$ and $\underline{G}$ as defined in Section 2.2.2 . We have

$$\|\Phi_L(J^\uparrow \underline{X}) - J^\uparrow \underline{\Phi}_L(\underline{X})\|_2 \leqslant (C_1(\mathscr{W}) \cdot \|X\|_2 + C_2(\mathscr{W}, \mathscr{B})) \cdot \|R_z(\Delta) - J^\uparrow R_z(\underline{\Delta})J^\downarrow\| \quad (26)$$

if the network is based on Type-0 resolvent filters (c.f. Section 3). Additionally, we have

$$\|\Phi_L(X) - J^\uparrow \underline{\Phi}_L(J^\downarrow X)\|_2 \leqslant (C_1(\mathscr{W}) \cdot \|X\|_2 + C_2(\mathscr{W}, \mathscr{B})) \cdot \|R_z(\Delta) - J^\uparrow R_z(\underline{\Delta})J^\downarrow\| \quad (27)$$

if only Type-I filters are used in the network. Here $C_1(\mathscr{W})$ and $C_2(\mathscr{W}, \mathscr{B})$ are constants that depend polynomially on singular values of learned weight matrices $\mathscr{W}$ and biases $\mathscr{B}$.

*Proof.* Let us first prove (27). To this end, let us define

$$\underline{X} := J^\downarrow X.$$

Let us further use the notation $\underline{R}_z := (\underline{\Delta} - zId)^{-1}$ and $R_z := (\Delta - zId)^{-1}$.

Denote by $X^\ell$ and $\widetilde{X}^\ell$ the (hidden) feature matrices generated in layer $\ell$ for networks based on resolvents $R_z$ and $\underline{R}_z$ respectively: I.e. we have

$$X^\ell = \text{ReLU}\left(\sum_{k=a}^K R_z^k X^{\ell-1} W_k + B^\ell\right)$$

and

$$\widetilde{X}^\ell = \mathrm{ReLU}\left(\sum_{k=a}^K \underline{R}_z^k \widetilde{X}^{\ell-1} W_k + \underline{B}^\ell\right).$$

Here, since bias terms are proportional to constant vectors on the graphs, as detailed in Section 3, we have

$$J^\downarrow B = \underline{B}$$

and

$$J^\uparrow \underline{B} = B \tag{28}$$

for bias matrices $B$ and $\underline{B}$ in networks deployed on $G$ and $\underline{G}$ respectively.

We then have

$$\|\Phi_L(X) - J^\uparrow \underline{\Phi}_L(J^\downarrow X)\|$$
$$=\|X^L - J^\uparrow \widetilde{X}^L\|$$
$$= \left\|\mathrm{ReLU}\left(\sum_{k=a}^K R_z^k X^{L-1} W_k^L + B^L\right) - J^\uparrow \mathrm{ReLU}\left(\sum_{k=a}^K \underline{R}_z^k \widetilde{X}^{L-1} W_k^L + \underline{B}^L\right)\right\|$$
$$= \left\|\mathrm{ReLU}\left(\sum_{k=a}^K R_z^k X^{L-1} W_k^L + B^L\right) - \mathrm{ReLU}\left(\sum_{k=a}^K J^\uparrow \underline{R}_z^k \widetilde{X}^{L-1} W_k^L + B^L\right)\right\|.$$

Here we used the fact that since $\mathrm{ReLU}(\cdot)$ maps positive entries to positive entries and acts pointwise, it commutes with $J^\uparrow$. We also made use of (28).

Using the fact that $\mathrm{ReLU}(\cdot)$ is Lipschitz-continuous with Lipschitz constant $D = 1$, we can establish

$$\|\Phi_L(X) - J^\uparrow \underline{\Phi}_L(J^\downarrow X)\| \leqslant \left\|\sum_{k=a}^K R_z^k X^{L-1} W_k^L - \sum_{k=a}^K J^\uparrow \underline{R}_z^k \widetilde{X}^{L-1} W_k^L\right\|.$$

Using the fact that $J^\downarrow J^\uparrow = Id_{\underline{G}}$, we have

$$\|\Phi_L(X) - J^\uparrow \underline{\Phi}_L(J^\downarrow X)\| \leqslant \left\|\sum_{k=1}^K R_z^k X^{L-1} W_k^L - \sum_{k=1}^K (J^\uparrow \underline{R}_z^k J^\downarrow) J^\uparrow \widetilde{X}^{L-1} W_k^L\right\|.$$

From this, we find (using $\|J^\uparrow\|, \|J^\downarrow\| \leqslant 1$), that

$$\|X^L - J^\uparrow \widetilde{X}^L\|$$
$$\leqslant \left\|\sum_{k=0}^K R_z^k X^{L-1} W_k^L - \sum_{k=1}^K (J^\uparrow \underline{R}_z^k J^\downarrow) J^\uparrow \widetilde{X}^{L-1} W_k^L\right\|$$
$$\leqslant \left\|\sum_{k=1}^K (R_z^k - (J^\uparrow \underline{R}_z^k J^\downarrow)) X^{L-1} W_k^L\right\| + \sum_{k=1}^K \|J^\uparrow \underline{R}_z J^\downarrow\| \cdot \|J^\uparrow \widetilde{X}^{L-1} - X^{L-1}\| \cdot \|W_k^L\|$$
$$\leqslant \left\|\sum_{k=1}^K (R_z^k - (J^\uparrow \underline{R}_z^k J^\downarrow)) X^{L-1} W_k^L\right\| + \|\mathscr{W}^L\|_z \cdot \|J^\uparrow \widetilde{X}^{L-1} - X^{L-1}\|$$
$$\leqslant \sum_{k=1}^K \left\|R_z^k - (J^\uparrow \underline{R}_z^k J^\downarrow)\right\| \cdot \|X^{L-1}\| \cdot \|W_k^L\| + \|\mathscr{W}^L\|_z \cdot \|J^\uparrow \widetilde{X}^{L-1} - X^{L-1}\|$$

Applying Lemma E.1 yields

$$\|X^L - J^\uparrow \widetilde{X}^L\|$$
$$\leqslant \left(\sum_{k=1}^K \frac{k}{|z|^{k-1}} \|W_k^L\|\right) \cdot \|R_z - (J^\uparrow \underline{R}_z J^\downarrow)\| \cdot \|X^{L-1}\| + \|\mathscr{W}^L\|_z \cdot \|J^\uparrow \widetilde{X}^{L-1} - X^{L-1}\|.$$

Lemma then D.2 in Appendix D established that we have

$$\|X^L\| \leqslant \|B^L\| + \sum_{m=0}^{L} \left(\prod_{j=0}^{m} \|\mathscr{W}^{L-1-k}\|_z\right) \|B^{L-1-k}\| + \left(\prod_{\ell=1}^{L} \|\mathscr{W}^\ell\|_z\right) \cdot \|X\|. \qquad (29)$$

Hence the summand on the left-hand-side can be bounded in terms of a polynomial in singular values of bias- and weight matrices, as well as $\|X\|$ and most importantly the factor $\|R_z - (J^\uparrow \underline{R}_z J^\downarrow)\|$ which tends to zero.

For the summand on the right-hand-side, we can iterate the above procedure (aggregating terms like (29) multiplied by $\|R_z - (J^\uparrow \underline{R}_z J^\downarrow)\|$) until reaching the last layer $L = 1$. There we observe

$$\|X^1 - J^\uparrow \widetilde{X}^1\|$$

$$= \left\|\mathrm{ReLU}\left(\sum_{k=1}^{K} R_z^k X W_k^1 + B^1\right) - J^\uparrow \mathrm{ReLU}\left(\sum_{k=1}^{K} \underline{R}_z^k J^\downarrow X W_k^1 + \underline{B}^1\right)\right\|$$

$$\leqslant \left\|\sum_{k=1}^{K} R_z^k X W_k^1 - \sum_{k=1}^{K} J^\uparrow \underline{R}_z^k J^\downarrow X W_k^1\right\|$$

$$\leqslant \left\|\sum_{k=1}^{K} (R_z^k - J^\uparrow \underline{R}_z^k J^\downarrow) X W_k^1\right\|$$

$$\leqslant \left(\sum_{k=1}^{K} \frac{k}{|z|^{k-1}} \|W_k^1\|\right) \cdot \|R_z - (J^\uparrow \underline{R}_z J^\downarrow)\| \cdot \|X\|$$

The last step is only possible because we let the sums over powers of resolvents start at $a = 1$ as opposed to $a = 0$. In the latter case, there would have remained a term $\|X - J^\uparrow J^\downarrow X\|$, which would not decay as $\lambda_1(\Delta_{high}) \to \infty$.

Aggregating terms, we build up the polynomial stability constants of (27) layer by layer, and complete the proof.

The proof of (26) proceeds in complete analogy upon defining

$$X := J^\uparrow \underline{X}.$$

Note that starting with $\underline{X}$ on $\underline{G}$, implies that we have

$$J^\uparrow J^\downarrow X \equiv J^\uparrow J^\downarrow (J^\uparrow \underline{X}) = J^\uparrow \underline{X} \equiv X.$$

This avoids any complications arising from employing Type-0 filters in this setting.

$\square$

Next we transfer the previous result to the graph level setting:

**Theorem E.3.** Denote by $\Psi$ the aggregation method introduced in Section 3. With $\mu(G) = \sum_{i=1}^{N} \mu_i$ the total weight of the graph $G$, we have in the setting of Theorem 4.1 with Type-I filters, that

$$\|\Psi(\Phi_L(X)) - \Psi(\underline{\Phi}_L(J^\downarrow X))\|_2 \leqslant \sqrt{\mu(G)} \cdot (C_1(\mathscr{W}) \cdot \|X\|_2 + C_2(\mathscr{W}, \mathscr{B})) \cdot \|R_z(\Delta) - J^\uparrow R_z(\underline{\Delta}) J^\downarrow\|.$$

*Proof.* Let us first recall that our aggregation scheme $\Psi$ mapped a feature matrix $X \in \mathbb{R}^{N \times F}$ to a graph-level feature vector $\Psi(X) \in \mathbb{R}^F$ defined component-wise as

$$\Psi(X)_j = \sum_{i=1}^{N} |X_{ij}| \cdot \mu_i.$$

In light of Theorem E.2, we are done with the proof, once we have established that

$$\|\Psi(\Phi_L(X)) - \Psi(\underline{\Phi}_L(J^\downarrow X))\|_2 \leqslant \sqrt{\mu(G)} \cdot \|\Phi_L(X) - J^\uparrow \underline{\Phi}_L(J^\downarrow X)\|_2.$$

To this end, we first note that

$$\Psi(J^\uparrow \underline{X}) = \Psi(\underline{X}).$$

Indeed, this follows from the fact that given a connected component $R$ in $G_{\text{high}}$, the map $J^{\uparrow}$ assigns the same feature vector to each node $r \in R \subseteq G$ (c.f. (20)), together with the fact that

$$\underline{\mu}_R = \sum_{r \in R} \mu_r.$$

Thus we have

$$\| \Psi \left( \Phi_L(X) \right) - \Psi \left( \underline{\Phi}_L(J^{\downarrow}X) \right) \|_2 = \| \Psi \left( \Phi_L(X) \right) - \Psi \left( J^{\uparrow}\underline{\Phi}_L(J^{\downarrow}X) \right) \|_2.$$

Next let us simplify notation and write

$$A = \Phi_L(X)$$

and

$$B = J^{\uparrow}\underline{\Phi}_L(J^{\downarrow}X)$$

with $A, B \in \mathbb{R}^{N \times F}$. We note:

$$\| \Psi \left( \Phi_L(X) \right) - \Psi \left( J^{\uparrow}\underline{\Phi}_L(J^{\downarrow}X) \right) \|_2^2 = \sum_{j=1}^{F} \left( \sum_{i=1}^{N} (|A_{ij}| - |B_{ij}|) \cdot \mu_i \right)^2.$$

By means of the Cauchy-Schwarz inequality together with the inverse triangle-inequality, we have

$$\sum_{j=1}^{F} \left( \sum_{i=1}^{N} (|A_{ij}| - |B_{ij}|) \cdot \mu_i \right)^2 \leqslant \sum_{j=1}^{F} \left[ \left( \sum_{i=1}^{N} |A_{ij} - B_{ij}|^2 \cdot \mu_i \right) \cdot \left( \sum_{i=1}^{N} \mu_i \right) \right]$$

$$= \sum_{j=1}^{F} \left( \sum_{i=1}^{N} |A_{ij} - B_{ij}|^2 \cdot \mu_i \right) \cdot \mu(G).$$

Since we have

$$\| \Phi_L(X) - J^{\uparrow}\underline{\Phi}_L(J^{\downarrow}X) \|_2^2 = \sum_{j=1}^{F} \left( \sum_{i=1}^{N} |A_{ij} - B_{ij}|^2 \cdot \mu_i \right),$$

the claim is established. $\square$

## F  ADDITIONAL DETAILS ON EXPERIMENTS:

All experiments were performed on a single NVIDIA Quadro RTX 8000 graphics card.

### F.1  NODE CLASSIFICATION

**Datasets:**  We test our approach for the task of node-classification on eight different standard datasets across the entire homophily-spectrum. Among these, CITESEER Sen et al. (2008), CORA-ML McCallum et al. (2000) and PUBMED Namata et al. (2012) are citation graphs. Here each node represents a paper and edges correspond to citations. We also test on the MICROSOFT ACADEMIC graph Shchur et al. (2018) where an edge that is present corresponds to co-authorship. Bag-of-word representations act as node features. The WEBKB datasets CORNELL and TEXAS are datasets modeling links between websites at computer science departments of various universitiesPei et al. (2020). Node features are bag-of-words representation of the respective web pages. We also consider the actor co-occurence dataset ACTOR Tang et al. (2009) as well as the Wikipedia based dataset SQUIRREL Rozemberczki et al. (2021).

**Experimental setup**  We closely follow the experimental setup of Gasteiger et al. (2019b) on which our codebase builds: All models are trained for a fixed maximum (and unreachably high) number of $n = 10000$ epochs. Early stopping is performed when the validation performance has not improved for 100 epochs. Test-results for the parameter set achieving the highest validation-accuracy are then reported. Ties are broken by selecting the lowest loss (c.f. Velickovic et al. (2018); Gasteiger et al. (2019a)). Confidence intervals are calculated over multiple splits and random seeds at the $95\%$ confidence level via bootstrapping.

**Additional details on training and models:**   We train all models on a fixed learning rate of

$$\text{lr} = 0.1.$$

Global dropout probability $p$ of all models is optimized individually over

$$p \in \{0.3, 0.35, 0.4, 0.45, 0.5\}.$$

We use $\ell^2$ weight decay and optimize the weight decay parameter $\lambda$ for all models over

$$\lambda \in \{0.0001, 0.0005\}.$$

Where applicable (i.e. not for Gasteiger et al. (2019a); He et al. (2021)) we choose a two-layer deep convolutional architecture with the dimensions of hidden features optimized over

$$K_\ell \in \{32, 64, 128\}. \tag{30}$$

In addition to the hyperparemeters specified above, some baselines have additional hyperparameters, which we detail here: BernNet uses an additional in-layer dropout rate of dp_rate $= 0.5$ and for its filters a polynomial order of $K = 10$ as suggested in He et al. (2021). As suggested in Gasteiger et al. (2019a), the hyperparameter $\alpha$ of PPNP is set to $\alpha = 0.2$ on the MS_ACADEMIC dataset and to $\alpha = 0.1$ on other datasets. Hyperparameters depth $T$ and number of stacks $K$ of the ARMA convolutional layer Bianchi et al. (2019) are set to $T = 1$ and $K = 2$. ChebNet also uses $K = 2$ to avoid the known over-fitting issue Kipf & Welling (2017) for higher polynomial orders. For MagNet we use $K = 1$ as suggested in Zhang et al. (2021) and choose the parameter $q$ as given in Table 1 of Zhang et al. (2021) for the respective datasets. The graph attention network Velickovic et al. (2018) uses $8$ attention heads, as suggested in Velickovic et al. (2018).

For our ResolvNet model, we choose a depth of $L = 1$ with hidden feature dimension optimized over the values in (30) as for baselines. We empirically observed in the setting of *unweighted* graphs, that rescaling the Laplacian as

$$\Delta_{nf} := \frac{1}{c_{nf}}\Delta$$

with a normalizing factor $c_{nf}$ before calculating the resolvent

$$R_z(\Delta_{nf}) := (\Delta_{nf} - z \cdot Id)^{-1} \tag{31}$$

on which we base our ResolvNet architectures improved performance.

For our ResolvNet architecture, we express this normalizing factor in terms of the largest singular value $\|\Delta\|$ of the (non-normalized) graph Laplacian. It is then selected among

$$c_{nf}/\|\Delta\| \in \{0.001, 0.01, 0.1, 2\}.$$

The value $z$ in (31) is selected among

$$(-z) \in \{0.14, 0.15, 0.2, 0.25\}.$$

We base our ResolvNet architecture on Type-0 filters and choose the maximum resolvent-exponent $K$ as $K = 1$.

### F.2   GRAPH REGRESSION

**Datasets:**   The first dataset we consider is the **QM7** dataset, introduced in Blum & Reymond (2009); Rupp et al. (2012). This dataset contains descriptions of 7165 organic molecules, each with up to seven heavy atoms, with all non-hydrogen atoms being considered heavy. A molecule is represented by its Coulomb matrix $C^{\text{Clmb}}$, whose off-diagonal elements

$$C_{ij}^{\text{Clmb}} = \frac{Z_i Z_j}{|R_i - R_j|}$$

correspond to the Coulomb-repulsion between atoms $i$ and $j$. We discard diagonal entries of Coulomb matrices; which would encode a polynomial fit of atomic energies to nuclear charge Rupp et al. (2012).

For each atom in any given molecular graph, the individual Cartesian coordinates $R_i$ and the atomic charge $Z_i$ are also accessible individually. To each molecule an atomization energy - calculated via density functional theory - is associated. The objective is to predict this quantity. The performance metric is mean absolute error. Numerically, atomization energies are negative numbers in the range $-600$ to $-2200$. The associated unit is $[kcal/mol]$.

The second dataset we consider is the **QM9** dataset Ramakrishnan et al. (2014), which consists of roughly 130 000 molecules in equilibrium. Beyond atomization energy, there are in total 19 targets available on **QM9**. We provide a complete list of targets together with abbreviations in Table 4 below:

Table 4: Targets of QM9

| Symbol | Property | Unit |
|---|---|---|
| $U_0$ | Internal energy at $0K$ | $eV$ |
| $U$ | Internal energy at $298.15K$ | $eV$ |
| $H$ | Enthalpy at $298.15K$ | $eV$ |
| $G$ | Free energy at $298.15K$ | $eV$ |
| $U_0^{\text{ATOM}}$ | Atomization energy at $0K$ | $eV$ |
| $U^{\text{ATOM}}$ | Atomization energy at $298.15K$ | $eV$ |
| $H^{\text{ATOM}}$ | Atomization enthalpy at $298.15K$ | $eV$ |
| $G^{\text{ATOM}}$ | Atomization free energy at $298.15K$ | $eV$ |
| $c_v$ | Heat capacity at $298.15K$ | $\frac{\text{cal}}{\text{mol·K}}$ |
| $\mu$ | Dipole moment | $D$ |
| $\alpha$ | Isotropic polarizability | $\alpha_0^3$ |
| $\epsilon_{\text{HOMO}}$ | Highest occupied molecular orbital energy | $eV$ |
| $\epsilon_{\text{LUMO}}$ | Lowest unoccupied molecular orbital energy | $eV$ |
| $\Delta\epsilon$ | Gap between $\epsilon_{\text{HOMO}}$ and $\epsilon_{\text{LUMO}}$ | $eV$ |
| $\langle R^2 \rangle$ | Electronic spatial extent | $\alpha_0^2$ |
| ZPVE | Zero point vibrational energy | $eV$ |
| A | Rotational constant | $GHz$ |
| B | Rotational constant | $GHz$ |
| C | Rotational constant | $GHz$ |

Molecules in QM9 are not directly encoded via their Coulomb-matrices, as in QM7. However, positions and charges of individual molecules are available, from which the Coulomb matrix description is calculated for each molecule.

**Experimental Setup:** On both datasets, we randomly select 1500 molecules for testing and train on the remaining graphs. On QM7 we run experiments for 23 different random random seeds and report mean and standard deviation. Due to computational limitations we run experiments for 3 different random seeds on the larger QM9 dataset, and report mean and standard deviation.

**Additional details on training and models:** All considered convolutional layers are incorporated into a two layer deep and fully connected graph convolutional architecture. In each hidden layer, we set the width (i.e. the hidden feature dimension) to

$$F_1 = F_2 = 64.$$

For BernNet, we set the polynomial order to $K = 3$ to combat appearing numerical instabilities. ARMA is set to $K = 2$ and $T = 1$. ChebNet uses $K = 2$. For all baselines, the standard mean-aggregation scheme is employed after the graph-convolutional layers to generate graph level features. Finally, predictions are generated via an MLP.

For our model, we choose a two-layer deep instantiation of our ResolvNet architecture introduced in Section 3. We choose Type-I filters and set $z = -1$. Laplacians are *not* rescaled and resolvents are thus given as

$$R_{-1}(\Delta) = (\Delta + Id)^{-1}.$$

As aggregation, we employ the graph level feature aggregation scheme introduced at the end of Section 3 with node weights set to atomic charges of individual atoms. Predictions are then generated via a final MLP with the same specifications as the one used for baselines.

All models are trained independently on each respective target.

**Results:** Beyond the results already showcased in the main body of the paper, we here provide results for ResolvNet as well as baselines on all targets of Table 4. These results are collected in Table 5, Table 6 and Table 7 below.

As is evident from the tables, the ResolvNet architecture produces mean-absolute-errors comparable to those of baselines on $1/4$ of targets, while it performs significantly better on $3/4$ of targets.

The difference in performance is especially significant on the (extensive) energy targets of Table 5. In this Table, baselines are out-performed by factors varying between $4$ and $15$.

Table 6 contains three additional targets where MAEs produced by ResolvNet are lower by factors varying between roughly two and four, when compared to baselines.

Table 7 finally contains MAEs corresponding to predictions of rotational constants. Here our model yields a comparable error on one target and provides better results than baselines on two out of three targets.

Table 5: Energy prediction MAEs $[eV]$. Our Model is marked **R.N.** for **ResolvNet**.

| Property | $U_0$ | $U$ | $H$ | $G$ | $U_0^{\mathrm{ATOM}}$ | $U^{\mathrm{ATOM}}$ | $H^{\mathrm{ATOM}}$ | $G^{\mathrm{ATOM}}$ |
|---|---|---|---|---|---|---|---|---|
| BernNet | 370.42±38.91 | 382.64±36.52 | 398.32±46.00 | 362.69±24.84 | 3.112±0.285 | 3.096±0.249 | 3.046±0.277 | 2.919 ±0.375 |
| GCN | 381.41±0.42 | 376.41±7.10 | 368.01±16.77 | 380.65±6.67 | 2.766±0.081 | 2.828±0.091 | 2.803±0.077 | 2.575±0.084 |
| ChebNet | 345.74±12.30 | 346.39±19.11 | 398.32±22.48 | 350.22±12.32 | 2.665±0.040 | 2.672±0.056 | 2.745±0.104 | 2.477±0.036 |
| ARMA | 327.62±19.83 | 316.09±18.06 | 322.74±16.32 | 320.72±11.98 | 2.588±0.117 | 2.570±0.088 | 2.600±0.096 | 2.326±0.101 |
| R.N. | **21.72**±5.79 | **19.14**±7.19 | **31.18**±8.622 | **53.50**±4.58 | **0.605**±0.015 | **0.588**±0.024 | **0.593**±0.025 | **0.607**±0.041 |

Table 6: Various target prediction MAEs. Our Model is marked **R.N.** for **ResolvNet**.

| Property | $c_v \left[\frac{\mathrm{cal}}{\mathrm{mol \cdot K}}\right]$ | $\mu\,[D]$ | $\alpha\,[\alpha_0^3]$ | $\epsilon_{\mathrm{HOMO}}\,[eV]$ | $\epsilon_{\mathrm{LUMO}}\,[eV]$ | $\Delta\epsilon\,[eV]$ | $\langle R^2 \rangle\,[\alpha_0^2]$ | ZPVE $[eV]$ |
|---|---|---|---|---|---|---|---|---|
| BernNet | 2.610±0.986 | 0.948±0.042 | 3.519±0.288 | 0.376±0.028 | 0.649±0.092 | 0.841±0.085 | 157.982 ±34.804 | 0.237 ±0.032 |
| GCN | 1.521±0.038 | 0.936±0.003 | 3.114±0.112 | 0.301±0.009 | 0.523±0.018 | 0.566±0.016 | 130.461±5.445 | 0.185±0.004 |
| ChebNet | 1.455±0.053 | 0.881±0.007 | 3.049±0.092 | 0.234±0.005 | 0.433±0.018 | 0.515±0.010 | 132.695±2.218 | 0.180±0.005 |
| ARMA | 1.327±0.034 | 0.806±0.031 | 2.676±0.087 | **0.228**±0.010 | **0.333**±0.009 | **0.380**±0.007 | **93.760**±4.122 | 0.152±0.006 |
| R.N. | **0.747**±0.015 | **0.776**±0.018 | **1.308**±0.034 | 0.313±0.002 | 0.423±0.011 | 0.531±0.016 | 97.614±2.308 | **0.041**±0.008 |

Table 7: Rotational constants prediction MAEs. Our Model is marked **R.N.** for **ResolvNet**.

| Property | $A\,[GHz]$ | $B\,[GHz]$ | $C\,[GHz]$ |
|---|---|---|---|
| BernNet | 0.888±0.034 | 0.342±0.002 | 0.243±0.002 |
| GCN | 0.848±0.027 | 0.281±0.004 | 0.183±0.002 |
| ChebNet | 0.797±0.034 | 0.262±0.003 | 0.171±0.003 |
| ARMA | **0.715**±0.017 | 0.259±0.004 | 0.168±0.004 |
| R.N. | 0.783±0.802 | **0.249**±0.002 | **0.158**±0.001 |

### F.3 SCALE INVARIANCE

**Dataset:** Again, we make use of the QM7 dataset Rupp et al. (2012) and its Coulomb matrix description

$$C_{ij}^{\mathrm{Clmb}} = \frac{Z_i Z_j}{|R_i - R_j|} \tag{32}$$

of molecules.

**Details on collapsing procedure:** We modify (all) molecular graphs in QM7 by deflecting hydrogen atoms (H) out of their equilibrium positions towards the respective nearest heavy atom. This is possible since the QM7 dataset also contains the Cartesian coordinates of individual atoms.

This introduces a two-scale setting precisely as discussed in section 2: Edge weights between heavy atoms remain the same, while Coulomb repulsions between H-atoms and respective nearest heavy atom increasingly diverge; as is evident from (32).

Given an original molecular graph $G$ with node weights $\mu_i = Z_i$, the corresponding limit graph $\underline{G}$ corresponds to a coarse grained description, where heavy atoms and surrounding H-atoms are aggregated into single super-nodes in the sense of Section 2.2.2 .

Mathematically, $\underline{G}$ is obtained by removing all nodes corresponding to H-atoms from $G$, while adding the corresponding charges $Z_H = 1$ to the node-weights of the respective nearest heavy atom. Charges in (32) are modified similarly to generate the weight matrix $\underline{W}$.

On original molecular graphs, atomic charges are provided via one-hot encodings. For the graph of methane – consisting of one carbon atom with charge $Z_C = 6$ and four hydrogen atoms of charges $Z_H = 1$ – the corresponding node-feature-matrix is e.g. given as

$$X = \begin{pmatrix} 0 & 0 & \cdots & 0 & 1 & 0\cdots \\ 1 & 0 & \cdots & 0 & 0 & 0\cdots \\ 1 & 0 & \cdots & 0 & 0 & 0\cdots \\ 1 & 0 & \cdots & 0 & 0 & 0\cdots \\ 1 & 0 & \cdots & 0 & 0 & 0\cdots \end{pmatrix}$$

with the non-zero entry in the first row being in the $6^{\text{th}}$ column, in order to encode the charge $Z_C = 6$ for carbon.

The feature vector of an aggregated node represents charges of the heavy atom and its neighbouring H-atoms jointly.

As discussed in Definition 3.2, node feature matrices are translated as $\underline{X} = J^{\downarrow}X$. Applying $J^{\downarrow}$ to one-hot encoded atomic charges yields (normalized) bag-of-word embeddings on $\underline{G}$: Individual entries of feature vectors encode how much of the total charge of the super-node is contributed by individual atom-types. In the example of methane, the limit graph $\underline{G}$ consists of a single node with node-weight

$$\mu = 6 + 1 + 1 + 1 + 1 = 10.$$

The feature matrix

$$\underline{X} = J^{\downarrow}X$$

is a single row-vector given as

$$\underline{X} = \left( \frac{4}{10}, 0, \cdots, 0, \frac{6}{10}, 0, \cdots \right).$$

**Results:**

For convenience, we repeat here in Table 8 and Figure 12 the results corresponding to the use of resolution-limited data in the form of coarse-grained molecular graphs during inference, that were already presented in the main body of the paper.

Table 8: MAE on QM7 via coarsified molecular graphs.

| QM7 | MAE $[kcal/mol]$ |
|---|---|
| BernNet | $580.67_{\pm 99.27}$ |
| GCN | $124.53_{\pm 34.58}$ |
| ChebNet | $645.14_{\pm 34.59}$ |
| ARMA | $248.96_{\pm 15.56}$ |
| ResolvNet | $\mathbf{16.23}_{\pm 2.74}$ |

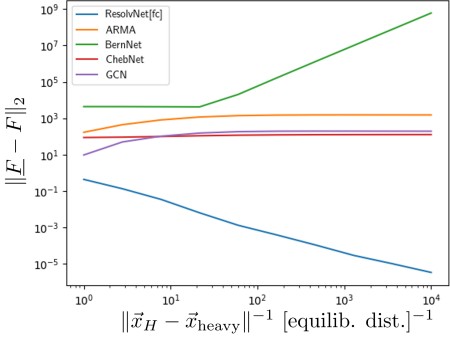

Figure 12: Feature-vector-difference for collapsed ($\underline{F}$) and deformed ($F$) graphs.

## G  COMPARISON OF RESOLVNET WITH OTHER POSSIBLE MULTI-SCALE PROPAGATION SCHEMES

Here we numerically compare our ResolvNet architecture with additional, conceptionally adjacent baselines beyond those already considered in Section 5.

### G.1  COMPARISON WITH (GNN + POOLING)

Aside from incorporating multiple-scales via the use of resolvents as in ResolvNet, one might also transition between different coarse-grained versions of an original graph using (learnable) pooling operations. Here we compare ResolvNet against a generic message passing network (i.e. GCN) with a generic learnable pooling operation (i.e. Self-Attention Graph Pooling (Lee et al., 2019)).

We consider two instantiations of the (GNN + Pooling) combination: In SAG, we first apply a pooling Layer, and then run GCN on the coarsified graph. SAG-Multi instead precisely follows the architecture described in Lee et al. (2019): It first generates graph representations first at the original scale via GCN, then pools, runs GCN on the coarsened graph and generates a second (coarser) Graph representation. This procedure is iterated again and all three representations are combined into a single representation.

We then rerun all experiments on QM7 data described in Section 5 again also for these two additional baselines: As can be inferred from Table 9, either way of combining GCN (as the prototypical message passing network) with pooling does not lead to a performance increase on the multi-scale data of the QM7-dataset.

Table 9: QM7-MAE

| QM7 | MAE [$kcal/mol$] |
| --- | --- |
| BernNet | $113.57_{\pm 62.90}$ |
| GCN | $61.32_{\pm 1.62}$ |
| ChebNet | $59.57_{\pm 1.58}$ |
| ARMA | $59.39_{\pm 1.79}$ |
| SAG | $64.69_{\pm 3.24}$ |
| SAG-Multi | $61.36_{\pm 4.47}$ |
| ResolvNet | $\mathbf{16.52}_{\pm 0.67}$ |

Table 10: QM7$_{\text{coarse}}$-MAE

| QM7 | MAE [$kcal/mol$] |
| --- | --- |
| BernNet | $580.67_{\pm 99.27}$ |
| GCN | $124.53_{\pm 34.58}$ |
| ChebNet | $645.14_{\pm 34.59}$ |
| ARMA | $248.96_{\pm 15.56}$ |
| SAG | $550.42_{\pm 24.43}$ |
| SAG-Multi | $246.95_{\pm 68.49}$ |
| ResolvNet | $\mathbf{16.23}_{\pm 2.74}$ |

What is more, it has a drastically negative effect when aiming to transfer models between graphs describing the same object (in this case a molecule) at different resolution-scales; as can be observed when comparing the MAE of GCN in Table 10 with either choice of SAG or SAG-Multi.

This effect is also observed when comparing the feature vectors SAG or SAG-Multi generate for a multi-scale graph $G$ and its collapsed version $\underline{G}$, as the scale imbalance in $G$ is increased:

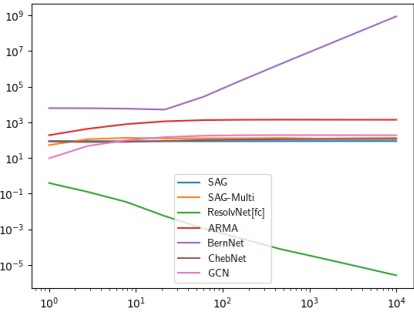

Figure 13: Feature-vector-difference for collapsed ($\underline{F}$) and deformed ($F$) graphs.

Rerunning the collapse experiment of Fig. 7, we observe from Fig. 13 that the difference between features generated for $G$ and $\underline{G}$ by SAG and SAG-Multi do not decay, as the larger scale is increased. We might thus conclude as before that these models – as opposed to ResolvNet – are scale- and resolution sensitive when generating graph level features.

### G.2 COMPARISONS WITH (GNN + MANUAL-SCALE-SEPARATION)

Apart from using learnable pooling operations to infer a collapsed version of an original graph, one might also manually determine the two graph structures $G_{\text{reg.}}$ and $G_{\text{high}}$ of Definition 2.1. One might then run graph neural networks on each graph $G_{\text{reg.}}$ and $G_{\text{high}}$ separately and subsequently combine the two individual representations into a single one, to represent the original graph $G$.

To compare our ResolvNet architecture against this approach, we reran the experiment corresponding to Table 2 with baselines now operating on two graph structures $G_{\text{reg.}}$ and $G_{\text{high}}$ for each molecule and results being subsequently combined into a single graph-level representation for each molecular graph $G$.

Given a molecular graph $G$, we determine the two graph structures $G_{\text{reg.}}$, $G_{\text{high}}$ as follows: Since weights scale as inverse distance ($w_{ij} \sim 1/\|\vec{x}_i - \vec{x}_j\|$) weights between nodes that are far away from each other are connected via weaker weights.

We thus determine a minimal distance $d_{\text{min}}$, and retain all edges of $G$ that connect nodes that are within distance $d_{\text{min}}$ of each other ($\|\vec{x}_i - \vec{x}_j\| \leqslant d_{\text{min}}$) in order to generate $G_{\text{reg.}}$. All other edges are taken to provide the geometry for $G_{\text{reg.}}$.

In our experiment, we set $d_{\text{min}}$ to be 1.5 times the minimal occurring bond length in any molecule of QM7. Results are collected in Table 12: For most baselines, this scale separation indeed improves performance significantly. However, ResolvNet is still performing better than the nearest competitor by roughly a factor of two.

Table 11: $G$-MAE

| QM7 | MAE $[kcal/mol]$ |
|---|---|
| BernNet | $113.57_{\pm 62.90}$ |
| GCN | $61.32_{\pm 1.62}$ |
| ChebNet | $59.57_{\pm 1.58}$ |
| ARMA | $59.39_{\pm 1.79}$ |
| ResolvNet | $\mathbf{16.52}_{\pm 0.67}$ |

Table 12: $G_{\text{reg.}} + G_{\text{high}}$-MAE

| QM7 | MAE $[kcal/mol]$ |
|---|---|
| BernNet | $85.24_{\pm 9.60}$ |
| GCN | $64.16_{\pm 1.89}$ |
| ChebNet | $47.11_{\pm 1.89}$ |
| ARMA | $30.88_{\pm 2.82}$ |
| $\longleftarrow \ \longleftarrow$ | $\longleftarrow \ \longleftarrow \ \longleftarrow$ |

## H ADDITIONAL EXPERIMENTS ON SYNTHETIC DATA

In order to showcase the multi-scale consistency of ResolvNet in a completely controlled environment, we conduct additional experiments on graphs generated via stochastic block models:

**Stochastic Block Models:** Stochastic block models (Holland et al., 1983) are generative models for random graphs that produce graphs containing strongly connected communities. In our experiments in this section, we consider a stochastic block model whose distributions is characterized by four parameters: The number of communities $c_{\text{number}}$ determine how many (strongly connected) communities are present in the graph that is to be generated. The community size $c_{\text{size}}$ determines the number of nodes belonging to each (strongly connected) community. The probability $p_{\text{inner}}$ determines the probability that two nodes within the same community are connected by an edge. The probability $p_{\text{inter}}$ determines the probabilities that two nodes in *different* communities are connected by an edge.

**Experimental Setup:** Since stochastic block models do not generate node-features, we equip each node with a randomly-generated unit-norm feature vector. Given such a graph $G$ drawn from a stochastic block model, we then compute a version $\underline{G}$ of this graph, where all communities are collapsed to single nodes as described in Definition 2.2. In analogy to the experimental setup for Table 3 and Figure 7 in Section 5, we then compare the feature vectors generated for $G$ and $\underline{G}$.

**Experiment I: Varying the Connectivity within the Communities:** As discussed in detail in Sections 2.1 and 2.2.2, we desire that ResolvNet assigns similar feature vectors to graphs with strongly connected communities and coarse-grained versions of this graph, where these communities are collapsed to aggregate node. The higher the connectivity within these communities, the more similar should the feature vector of the original graph $G$ and its coarsified version $\underline{G}$ be, as Theorem 3.3 established.

In order to verify this experimentally, we fix the parameters $c_{\text{number}}$, $c_{\text{size}}$ and $p_{\text{inter}}$ in our stochastic block model. We then vary the probability $p_{\text{inner}}$ that two nodes within the same community are connected by an edge from $p_{\text{inner}} = 0$ to $p_{\text{inner}} = 1$. This corresponds to varying the connectivity within the communities from very sparse (or in fact no connectivity) to full connectivity (i.e. the

community being a clique). In Figure 14 below, we then plot the difference of feature vectors generated by ResolvNet and baselines for for $G$ and $\underline{G}$ respectively. For each $p_{\text{inner}} \in [0, 1]$, results are averaged over 100 graphs randomly drawn from the same stochastic block model.

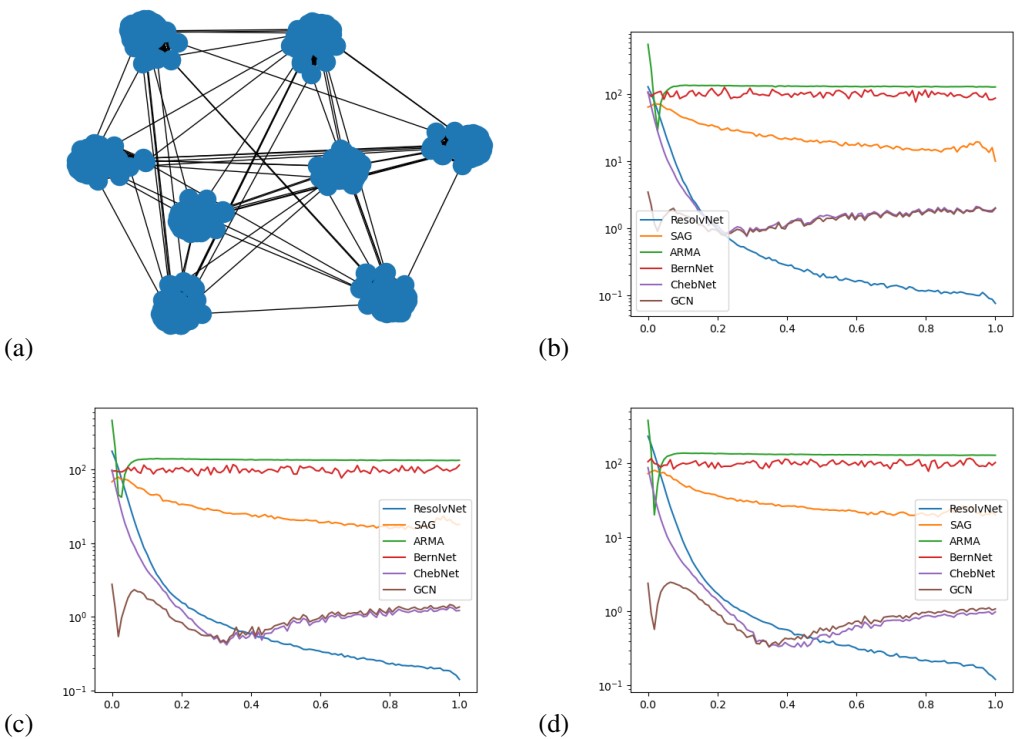

(a)

(b)

(c)

(d)

Figure 14: Varying the parameter $p_{\text{inner}} \in [0, 1]$ for fixed $c_{\text{size}} = 60$, $p_{\text{inter}} = 2/c_{\text{size}}^2$ and $c_{\text{number}} = 8, 12, 16$: (a) Example Graph for $c_{\text{number}} = 8$, (b) Results for $c_{\text{number}} = 8$, (c) Results for $c_{\text{number}} = 12$, (d) Results for $c_{\text{number}} = 16$.

We have chosen $p_{\text{inter}} = 2/c_{\text{size}}^2$ so that – on average – *clusters* are connected by two edges. The choice of two edges (as opposed to $1, 3, 4, 5, ...$) between clusters is not important; any arbitrary choice of $p_{\text{inter}}$ will ensure a decay behavior for ResolvNet as in Figure 14. A corresponding ablation study is provided below.

As can be inferred from Fig. 14, ResolvNet produces more and more similar feature-vectors for $G$ and its coarse-grained version, as the connectivity within the clusters is increased. This numerically verifies Theorem 4.2. For baselines, the difference of generated feature vectors between $G$ and $\underline{G}$ does not decay. This implies that these methods are not able to consistently integrate multiple scales and hence are not transferable between different graphs describe the same underlying object at different resolution scales.

**Experiment II: Varying the Parameter $p_{\text{inter}}$:**

In order to determine the sensitivity of the results of Fig. 14 to the connectivity outside the communities that are collapsed, we now fix the parameters $p_{\text{inner}}$, $c_{\text{number}}$ and $c_{\text{size}}$ and vary the parameters $p_{\text{inter}}$ from $p_{\text{inter}} = 0$ to $p_{\text{inter}} = 1$. As was to be expected, the transferability error is smallest when the connectivity outside the strongly connected communities is smallest (i.e. for $p_{\text{inter}} = 0$ in Fig. 15). As $p_{\text{inter}}$ is increased, the transferability error increases. It should however be noted that even if the connectivity inside and outside of the respective communities is the same (i.e. $p_{\text{inter}} = p_{\text{inner}} = 1$; corresponding to a fully connected graph $G$), it is still justified to collapse the previously assigned communities. This suggests that while the inequality of Theorem 3.3 is certainly valid and capturing the trend of better transferability for lower connectivity outside communities, it is – depending on the situation – not necessarily tight.

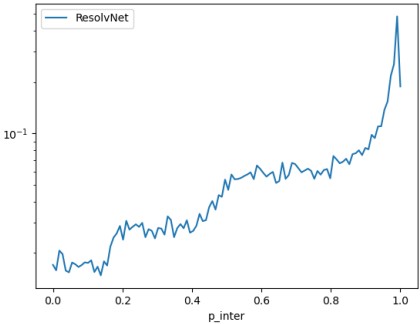

Figure 15: Results for $p_{\text{inner}} = 1$, $c_{\text{number}} = 8$, $c_{\text{size}} = 60$

**Experiment III: Varying the number of Communities**

Beyond simply varying the connectivity outside of communities by varying $p_{\text{inter}}$, we also investigate the influence of increasing the number of communities that are considered. For this experiment, we once again keep the average number of edges between communities fixed by setting $p_{\text{inter}} = 2/c_{\text{size}}^2$.

As the number of communities is increased, the transferability errors increases too. From a heuristic perspective, this is to be expected: As the number of communities is increased, but the average number of edges between communities is kepr constant, each community will acquire more and more outgoing edges. Hence the connectivity outside of these communities is increased. In light of Theorem 3.3 and Theorem 4.2, we thus also expect increasing transferability errors.

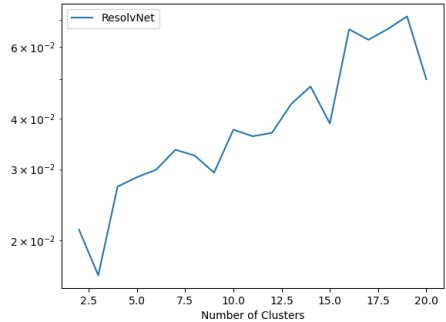

Figure 16: Results for $c_{\text{size}} = 60$, $p_{\text{inner}} = 1$ and $p_{\text{inter}} = 2/c_{\text{size}}^2$ (c.f. also Fig. 14 (b)).

## I   ANALYSIS OF COMPUTATIONAL OVERHEAD

Here we provide an analysis of the overhead of our ResolvNet method. As is evident from Tables 13, 14, 15 below, on most datasets our method is not the most memory intensive to train when compared to representative (spatial and spectral) baselines. For training times (total and per-epoch), we note that on most small to medium sized graphs, our model is not the slowest to train. On larger graphs it does take longer to train. Regarding complexity, the node update for our model is essentially $\mathcal{O}(N^2)$ (dense-dense matrix multiplication), while message passing baselines scale linearly in the number of edges.

Table 13: Maximal Memory Consumption [GB] while training a single model of depth 2 and width 32 for learning rate lr $= 0.1$, dropout $p = 0.5$, weight decay $\lambda = 10^{-4}$ and early stopping patience $t = 100$. All measurements performed on the same GPU via `torch.cuda.max_memory_allocated()`.

|          | MS_Acad. | Cora   | Pubmed | Citeseer | Cornell | Actor  | Squirrel | Texas  |
|----------|----------|--------|--------|----------|---------|--------|----------|--------|
| ResolvNet | 3.47     | 0.1266 | 2.9915 | 0.0996   | 0.0070  | 0.4936 | 0.2915   | 0.0175 |
| GAT       | 1.49     | 0.1559 | 0.6486 | 0.1105   | 0.0228  | 0.3666 | 2.1107   | 0.0219 |
| ChebNet   | 10.19    | 0.4741 | 0.4848 | 0.3389   | 0.0249  | 0.4830 | 6.3569   | 0.0241 |

Table 14: Training Time [s] for training a single model of depth 2 and width 32 for learning rate lr $= 0.1$, dropout $p = 0.5$, weight decay $\lambda = 10^{-4}$ and early stopping patience $t = 100$. All measurements performed on the same GPU.

|  | MS_Acad. | Cora | Pubmed | Citeseer | Cornell | Actor | Squirrel | Texas |
|---|---|---|---|---|---|---|---|---|
| ResolvNet | 474.409 | 3.671 | 34.140 | 1.387 | 1.745 | 9.623 | 4.874 | 0.875 |
| GAT | 34.388 | 2.194 | 5.741 | 0.891 | 2.123 | 1.610 | 23.060 | 1.375 |
| ChebNet | 87.567 | 6.818 | 3.221 | 2.833 | 2.713 | 1.488 | 14.383 | 4.511 |

Table 15: Average Training Time per Epoch [ms] for training a single model of depth 2 and width 32 for learning rate lr $= 0.1$, dropout $p = 0.5$, weight decay $\lambda = 10^{-4}$ and early stopping patience $t = 100$. All measurements performed on the same GPU.

|  | MS_Acad. | Cora | Pubmed | Citeseer | Cornell | Actor | Squirrel | Texas |
|---|---|---|---|---|---|---|---|---|
| ResolvNet | 1359.34 | 13.16 | 161.80 | 11.01 | 2.58 | 32.51 | 41.30 | 2.54 |
| GAT | 60.01 | 8.22 | 29.59 | 7.24 | 3.93 | 15.05 | 62.49 | 4.07 |
| ChebNet | 202.23 | 12.11 | 14.31 | 10.61 | 3.89 | 13.28 | 126.17 | 3.83 |

## J    SCALING RESOLVNET TO LARGE GRAPHS

**ResolvNet on small- to medium-sized Graphs:**    As discussed in Section 3.2, the convolutional filters learned by ResolvNet are given by

$$f_{z,\theta}(\Delta) := \sum_{k=a}^{K} \theta_i \left[ (\Delta - zId)^{-1} \right]^k .$$

During preprocessing, computational complexity is thus determined by the complexity of matrix inversion (i.e. $\mathcal{O}(N^3)$, with $N$ the number of nodes).

During training and inference, the complexity is determined by dense-dense matrix multiplication, and is hence given as $\mathcal{O}(N^2)$.

**ResolvNet on large Graphs:**    For large Graphs (i.e. $N \gg 1$), these operations generically are too costly to be implemented. Here we detail how we may instead achieve a much more economical $\mathcal{O}(E \cdot F \cdot C)$complexity; with $E$ the number of edges, $F$ the number of hidden dimensions and $C$ the size of the input-features:

Instead of actually performing the full inversion $(\Delta - zId)^{-1}$, we make use of the Neumann-approximation to the inversion of matrices of the form $(A + Id)$, which is given as

$$(A + Id)^{-1} = \sum_{m=0}^{\infty} A^m .$$

For us, this yields

$$(\Delta - zId)^{-1} = -\frac{1}{z} \cdot \left( -\frac{1}{z}\Delta + Id \right)^{-1} \approx -\frac{1}{z} \cdot \sum_{m=0}^{M} \left( -\frac{1}{z}\Delta \right)^m .$$

Similar considerations were already used to approximate the full PPNP-propagation scheme of (Gasteiger et al., 2019b), in order to scale it to large graphs.

More generally, the relation

$$[(\Delta-zId)^{-1}]^k = \sum_{m=0}^{\infty}\left((-1)^m(-z)^{-k-m}\binom{-k}{m}\right)\cdot\Delta^m \approx \sum_{m=0}^{M}\left((-1)^m(-z)^{-k-m}\binom{-k}{m}\right)\cdot\Delta^m; \tag{33}$$

as can be seen from computing a straightforward Taylor expansion (in $x$) of the function $g(x) = (x-z)^{-k}$ and comparing coefficients with (33).

Here $\binom{-k}{m}$ is an extension of the regular binomial coefficient to negative numbers as

$$\binom{-k}{m} = \frac{\Gamma(1-k)}{\Gamma(m+1)\Gamma(-k-n+1)},$$

with $\Gamma$ the standard Gamma function defined as

$$\Gamma(z) = \int_0^{\infty} t^{z-1}e^{-t}dt$$

for any $z$ with positive real-part.

On large graphs, we thus do not directly invert $(\Delta - zId)$, as this would be too costly. Instead we approximate $[(\Delta - zId)^{-1}]^k$ as in (33), which can be implemented as sparse-dense matrix multiplication (and addition), and hence has a significantly lower complexity of $\mathcal{O}(E \cdot F \cdot C)$, with $E$ the number of edges, $F$ the number of hidden dimensions and $C$ the size of the input-features. In applications, on would typically choose $M \approx 10$, as was e.g. done in (Gasteiger et al., 2019b).

**Empirical Complexity Evaluation on large Graphs:** In order to empirically investigate the complexity of ResolvNet (implemented via (33) for $K = 1$ and $M = 10$) on large graphs, we trained ResolvNet ChebNet (also with $K = 1$) and GAT in a two-layer deep configuration with $64$ hidden units in each layer for $100$ epochs on the OGBN-Arxiv dataset, which is comprised of $169,343$ Nodes as well as the OGBN-Products dataset, which contains more than 2M nodes (Hu et al., 2020). In Table 16 below, we report complexity in terms of maximal memory consumption (as measured via `torch.cuda.max_memory_allocated()`), training time per epoch and total number of performed multiply-accumulate operations (MACs), as these can be much more reliably computed than FLOPs.

Table 16: Empirical complexity on OGBN-Arxiv

| | trainable parameters | max-memory-allocated [GB] | Training-time per epoch [ms] | MACS |
|---|---|---|---|---|
| ResolvNet | 39848 | 0.68 | 634.71 | 4,595,291,648 |
| GAT | 351272 | 25.53 | 1089.96 | 3,468,144,640 |
| ChebNet | 15016 | 0.42 | 64.47 | 433,518,080 |

Table 17: Empirical complexity on OGBN-Product

| | trainable parameters | max-memory-allocated [GB] | Training-time per epoch [ms] | MACS |
|---|---|---|---|---|
| ResolvNet | 39848 | 19.23 | 37818.73 | 57,261,648,000 |
| GAT | 351272 | OOM | OOM | OOM |
| ChebNet | 15016 | 8.48 | 548.78 | 7,176,793,216 |

As can be inferred from Table 16, ResolvNet takes a lot less time to train on OGBN-Arxiv, when compared to GAT. It also needs considerably less GPU memory and multiply-accumulate operations compared to GAT. Compared to ChebNet (for which we have also set $K = 1$), our method needs to compute an order of magnitude more sparse-dense matrix multiplication operations (as we have set $M = 10$ and $K = 1$ in eq. (33)). This is reflected in the number of performed MACs, which for ResolvNet is roughly an order of magnitude larger than that for ChebNet.

On OGBN-Products, our method remains more complex than ChebNet, but in contrast to GAT remains trainable (c.f. Table 17).

## K  ABLATION STUDIES:

As dicussed in Section 3.2, the layer-update of our method is given as

$$X^{\ell+1} = \text{ReLu}\left(\sum_{k=a}^{K}(\Delta - \omega Id)^{-k} \cdot X^{\ell} \cdot W_k^{\ell+1} + B^{\ell+1}\right)$$

Here we provide ablation studies on the parameters $\omega$ and $K$:

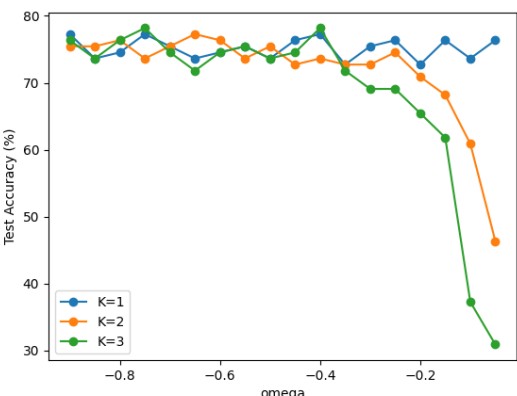

Figure 17: Performance dependence on hyperparameters $\omega$ and $K$: Line-plots

To investigate the sensitivity of ResolvNet's performance to the choice of hyperparameters $\omega$ and $K$, we fix a one layer deep ResolvNet of width $F = 128$ and perform node-classification on the Citeseer dataset while varying the $\omega$ and $K$. As can be inferred from Fig. 17, increasing the order $K$ of the utilized resolvent polynomials (2) beyond $K = 1$ does not aid classification accuracy. This is reminiscent of the behaviour of ChebNet (Defferrard et al., 2016), for which a similar fact holds (He et al., 2022).

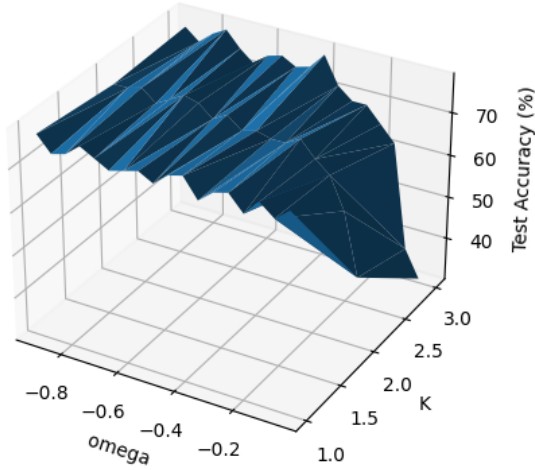

Figure 18: Performance dependence on hyperparameters $\omega$ and $K$: Surface-plot

Furthermore, we see that for fixed $K$, the model-performance is not overly sensitice to the choice of $\omega$, as long as $\omega$ is far away enough from zero. As $\omega \to 0$, the performance tends to worsen. As $\omega \to 0$, the matrix $(\Delta - \omega Id)$ converges to a singular matrix, so that the inversion operation becomes less and less stable.

