# OpenReview forum: "ResolvNet: A Graph Convolutional Network with multi-scale Consistency"
_ICLR.cc/2024/Conference — Submitted to ICLR 2024_

### Official Review · Reviewer_2iPY · 2023-10-30

**Soundness:** 2 fair
**Presentation:** 1 poor
**Contribution:** 2 fair
**Rating:** 3
**Confidence:** 4

**Summary:**

This paper first points out that the presence of strongly connected sub-graphs may severely restrict information flow in common GNN architectures. Then, it introduces the concept of multi-scale consistency, which can fit both the node-level and graph-level scenarios. In light of this, the authors introduce ResolvNet, a flexible graph neural network based on the mathematical concept of resolvents. Finally, it conducts some experiments to evaluate the proposed method, showing that the proposed method outperforms state-of-the-art baselines on several tasks across multiple datasets.

**Strengths:**

1.	It provides some theorical support for the proposed model.
2.	It tests on several widely-used datasets, and the proposed method can sometimes beat the existing methods.
3.	The authors provide their codes.

**Weaknesses:**

1.	SOTA baselines are largely ignored. On three famous datasets Cora, Citeseer and Pubmed, there are only two baselines are considered (in Table 1). Few baselines (like GCNII and GraphMAE2), which after 2022, are considered. As far as I know, GCNII (which is open source) can beat the proposed method on Cora and Pubmed. Moreover, even this, the proposed method cannot get the best performance in Table 3.
2.	The work is some kind of hard to follow. Although providing lots of theories will enhance the paper, the readability is also should be considered.
3.	Some grammatical errors, like 1) satisfied by poular graph -> “popular”; 2) severly restricts - > “severely”. 3) degree occuring -> “occurring”

**Questions:**

1.	Why the reported results cannot beat SOTA baselines (like GCNII) in Table 1?
2.	How many hyper-parameters are there in your method? If the proposed method contains too many hyper-parameters, it will be hard to reproduce.
3.	See the weakness in the “*Weaknesses” part.
4.	The work can be largely improved by enhancing its experiments and fixing gram errors.

---

> ### Author Response · Authors · 2023-11-21
> **ResolvNet performs better than GCNII. It also already performs best (not worst) in Table 3 since here “lower-is-better” (the metric being mean-absolute-error).**
>
> We would like to thank the reviewer for reviewing our paper and providing us with their feedback.
>
> As we understand it, the main points of criticism expressed by the reviewer are that 1) our method supposedly does not perform best in Table 3, and 2) the reviewer hypothesized that our ResolvNet method would not perform better than GCNII in Table 1.
>
> **Both points however  stem from misconceptions:**
>
> 1) **In Table 3** the metric is “mean-absolute-error”, so that **lower numbers correspond to better performance**. Our  method actually performs best in Table 3 by a large margin (i.e. by an error that is lower by factor of $10$ to $40$ compared to baselines) as discussed in the main-text of our submission.
> 2) Accuracies reported in the GCNII-paper [2] and Table 1 of our submission can not simply be compared, as the **experimental setups differ significantly**: Our paper uses the experimental setup and (class-balanced) random splits of [1] (i.e. “predict then propagate”). Instead, [2]  uses the experimental setup and splits of [3].
> **Evaluating ResolvNet and GCNII in a common evaluation setup clearly shows that ResolvNet performs better** (details provided below).
>
> Let us address these and all other points raised by the reviewer in detail individually below:

---

> ### Author Response · Authors · 2023-11-21
> **Experimental Results**
>
> * Why the reported results cannot beat SOTA baselines (like GCNII) in Table 1?
>
> The results in Table 1 were obtained following the experimental setup of [1] (i.e. “Predict then Propagate”) which evaluates on class-balanced  random splits. In contrast the GCNII-paper followed the experimental setup of  [2], so that reported numerical values for accuracies can not simply be compared.
>
> To allow for a comparison of GCNII with the other baselines of Table 1 as well as ResolvNet, we evaluated GCNII within the experimental Setup of Table 1 (i.e. following the setup of  [1]).
> We used the best-performing hyperparameters whenever they were reported in the original GCNII paper [3] and otherwise performed a full hyperparameter sweep.
>
> As is evident from Table 1 in our updated manuscript, our method actually performs better than GCNII on the vast majority of datasets.
>
> Regarding the performance in the homophilic regime the reviewer inquired about
> > (As far as I know, GCNII (which is open source) can beat the proposed method on Cora and Pubmed)
>
> we note the following result:
> |[\%]  | MS. Acad. | Cora  | Pubmed  | Citeseer  |
> |---|---|---|---|---|
> |  GCNII | 88.37 $\pm$ 0.16  |  84.41 $\pm$ 0.26 | 78.45 $\pm$ 0.31  | 73.12 $\pm$ 0.35  |
> |  ResolvNet| 92.73 $\pm$ 0.08 |  84.16 $\pm$ 0.26| 79.26 $\pm$ 0.36  | 75.03 $\pm$ 0.29 |
>
> On Cora, there is a marginal difference between results for ResolvNet and GCNII (84.16\% vs. 84.41 \%). ResolvNet clearly performs better on the other datasets.
>
>
>
>
>
> We would also like to thank the reviewer for bringing the recent and  interesting GraphMAE2 paper to our attention.
> This paper focusses on the self-supervised setting, the training procedure of the GraphMAE2 model involves pre-training and the backbones for its encoder and decoder might be constituted by any GNN-architecture (including ResolvNet).
> In contrast, Table 1 focusses on benchmarking the performance of different _convolutional_ - _layers_ against each other; all under the same (semi-)supervised training procedure and using the same loss.
> Nevertheless, we found both GraphMAE2 and its precursor GraphMAE to be interesting works and are happy to now cite them in the introduction to our paper.
>
> Finally, we would like to point out that – while ResolvNet generally performs better than baselines in the homophilic setting -- we are not concerned with designing another network that incrementally improves the state of the art on standard node-classification datasets.
>
> Instead we are concerned with introducing the concept of multi-scale consistency. We  explain its importance for the transferability of models between graphs describing the same object at different resolutions, as well as for the information-flow within multi-scale graphs. We then introduce ResovNet as a method to address the lack of multi-scale consistency of previous architectures.
>
>
>
> * On three famous datasets Cora, Citeseer and Pubmed, there are only two baselines are considered (in Table 1).
>
> We clearly compared against $10$ baselines on all datasets of Table 1 (including Cora, Citeseer and Pubmed) in our initial manuscript. We are  unsure how to interpret this comment; could the reviewer clarify?
>
> In our updated manuscript, we now additionally also compare against  the recent ChebNetII (He et al. (2022)) as an additional  baseline. Thus we now compare in total against $12$ (as opposed to the previous $10$) baselines in this table. Our conclusions  remain the same.
>
> * the proposed method cannot get the best performance in Table 3.
>
> As stated in the header of Table 3, this Table lists Mean Absolute Errors (MAEs). Here, lower numbers thus correspond to better performance. Thus our method is clearly the best performing one in Table 3.
>
> As already discussed in the corresponding section of the main-text, our method in fact not only performs best, but it does so by a large margin (MAEs are lower by factor of between 10 to 40 compared to baselines).
>
> While we had provided the evaluation metric in the header of Table 3,and marked our method via “\textbf” as the best-performing method in this Table, we have -- following this comment by the reviewer --  now added a “($\downarrow$)” symbol to the header of Table 3 for additional clarity.

---

> > ### Author Response · Authors · 2023-11-21
> > **Presentation, Questions and References**
> >
> > * [The paper] provides some theoretical support for the proposed model. / The work is some kind of hard to follow. Although providing lots of theories will enhance the paper, the readability is also should be considered.
> >
> > We agree that readability should be a primary concern for any publication.
> >
> > Our paper introduces multi-scale consistency as a completely novel concept with significant implications on the transferability of GNNs between different graphs and the handling of individual multi-scale graphs.
> > It subsequently proposes ResolvNet; a network that – as opposed to baselines – is able to consistently incorporate multiple scales within graphs.
> >
> > Since both the concept of multi-scale Consistency as well as the ideas on which ResolvNet builds are completely novel within the graph-learning-community, it is unfortunately impossible to avoid a thorough theoretical exposition of these concepts. We felt it crucial to provide watertight and exact discussions of all newly introduced ideas in order for the community to take up and build on our ideas and model.
> >
> > In our paper, we take great care to make our ideas and results as accessible as possible:
> > We provide a detailed conceptual introduction of the main novel concepts in the main body of our paper.
> >
> > 1) Proofs of results and mathematically fully rigorous discussions are deferred to the Appendix.
> > 2) We make use of multiple different techniques to convey our ideas (Text, Figures, Graphs, Theorems, Many Examples, etc.).
> > 3) We provide multiple examples for each newly introduced concept.
> > 4) We link to further and advanced discussions of concepts in the appendix.
> >
> > However, we would be more than happy to further improve our presentation! What can we change in order to improve readability further?
> >
> > * Some grammatical errors, like 1) satisfied by poular graph -> “popular”; 2) severly restricts - > “severely”. 3) degree occuring -> “occurring”
> >
> > We thank the reviewer for spotting these typos. We have corrected them in our updated manuscript.
> >
> >
> >  * How many hyper-parameters are there in your method? If the proposed method contains too many hyper-parameters, it will be hard to reproduce.
> >
> > For fixed  Type-0 and Type-I resolvent-filters of order $K$, there are exactly two hyperparameters:
> >
> > A normalizing factor for the Laplacian and the value of $z$.
> >
> > These are _global_ hyperparamters: They are set once for the entire architecture.
> >
> > For the experiments of Table 1, $K = 1$ is fixed and both $z$ and the normalization-factor are tuned over four possible values.
> >
> > For all other experiments, they are frozen and simply set to one.
> >
> > Full details on hyperparameters of all models (ours and baselines) are provided in “Appendix F: Additional Details on experiments”.
> >
> >
> >
> >
> >
> >
> >
> >
> >
> >
> >
> > References:
> >
> > [1]: Johannes Gasteiger, Aleksandar Bojchevski, and Stephan Günnemann. Predict then propagate: Graph neural networks meet personalized pagerank. In 7th International Conference on Learning Representations, ICLR 2019, New Orleans, LA, USA, May 6-9, 2019. OpenReview.net, 2019b. URL https://openreview.net/forum?id=H1gL-2A9Ym.
> >
> > [2]: Yang, Z., Cohen, W. W., and Salakhutdinov, R. Revisiting semi-supervised learning with graph embeddings. In ICML, 2016
> >
> > [3]: Simple and deep graph convolutional networks. In Proceedings of the 37th International Conference on Machine Learning, ICML 2020, 13-18 July 2020, Virtual Event, volume 119 of Proceedings of Machine Learning Research, pp. 1725–1735. PMLR, 2020. URL http://proceedings.mlr.press/v119/chen20v.html

---

### Official Review · Reviewer_ZGZ9 · 2023-10-31

**Soundness:** 3 good
**Presentation:** 3 good
**Contribution:** 3 good
**Rating:** 6
**Confidence:** 3

**Summary:**

This paper study multi-scale consistency (distinct graphs describing the same object at different resolutions should be assigned similar feature vectors) of node representation in graph neural network, which is indeed an important topic that is less well explored.

The authors show existing GNN method lack of multi-scale consistency, then they propose ResolvNet to solve this issue. Experiment shows improvement on GNN performance.

**Strengths:**

1. This paper study multi-scale consistency (distinct graphs describing the same object at different resolutions should be assigned similar feature vectors) of node representation in graph neural network, which is indeed an important topic that is less well explored.

2. This paper provide a very clear definition on multi-scale consistency in Definition 2.1, and explain in great details (using both figures, text, and examples) to help readers understand why it is important.

3. The proposed method capture the intuition of multi-scale consistency.

**Weaknesses:**

1. Experiment dataset is small. This is potentially because the proposed method has very high complexity due to matrix inverse (see feed-forward rule in paragraph **The ResolvNet Layer**. The authors need to conduct experiment on larger datasets (e.g., OGBN) and report complexity in terms of FLOP/Wall-clock time.

2. Part of the discription is not very clear, please refer to Questions.

**Questions:**

1. I understand the definition of $G_\text{high}$ and $G_\text{reg}$, but I am very clear how two split an original graph into this two graph. This is related to Definiton 2.1.

2. Please elaborate on "we would have a Lipschitz continuity relation that allows to bound the difference in generated feature vector in terms of a judiciously chosen distance". This is the sentense above Eq. 1. I don't understand why.

---

> ### Author Response · Authors · 2023-11-21
> **Performed complexity analysis and answered questions**
>
> We would like to sincerely thank the reviewer for the careful review of our paper.
>
>  We were especially happy to read that we consider
>
> > indeed an important topic,
>
> that
> > the paper provide a very clear definition on multi-scale  consistency
>
> and that our various explanations using different modalities (text, examples, figures)
> > help readers understand why [multi-scale consistency] is important.
>
> Let us address the raised points individually:
>
> * Experiment dataset is small. This is potentially because the proposed method has very high complexity due to matrix inverse (see feed-forward rule in paragraph The ResolvNet Layer. The authors need to conduct experiment on larger datasets (e.g., OGBN) and report complexity in terms of FLOP/Wall-clock time.
>
> Following this important point by the reviewer, we have included an additional section (Appendix J: Scaling ResolvNet to large Graphs) that discusses in detail how ResolvNet is scaled to large graphs.
>
> As detailed in Appendix J, the key ingredient here is a Neumann-Representation of the resolvent that avoids an explicit matrix-inversion.
>
> This replaces the hypothetical $\mathcal{O}(N^3)$ complexity during preprocessing and  $\mathcal{O}(N^2)$ complexity during training and inference that a naïve implementation of ResolvNet would constitute with a  $\mathcal{O}(EFN)$ complexity (i.e. the complexity of sparse-dense matrix multiplication). Here $N$ is the number of nodes,  $E$ is the number of edges and $F$ is the feature dimension.
>
> Beyond these theoretical considerations and in addition to the empirical complexity analysis already provided in Appendix G, we also followed the recommendation of the reviewer and evaluated complexity experimentally  (reporting  (reliably measurable) MACS (i.e. multiply-accumulate operations) in-lieu of FLOPS):
>
> We trained  ResolvNet, ChebNet (both with $K = 1$) and GAT in a two-layer deep configuration with $64$ hidden units in each layer for $100$ epochs on the OGBN-Arxiv and OGBN-Products datasets, which are comprised of $169,343$ Nodes and more than 2M nodes respectively.
>
> Results are collected in the tables below:
>
> Results are collected in the tables below:
>
> OGBN-Arxiv:
> |	    	 |  trainable parameters  |   max-memory-allocated [GB] | Training-time per epoch [ms]  | MACS|
> | ----------- | ----------- | ----------- | ----------- | ----------- |
> | ResolvNet  | 39848 	| 0.68  	| 634.71| 4,595,291,648 |
> |GAT  	| 351272 |  25.53   |  1089.96	| 3,468,144,640      |
> |ChebNet  |  15016	| 0.42	| 64.47 	| 433,518,080        |
>
>
> OGBN-Products:
>
> | |  trainable parameters |   max-memory-allocated [GB] | Training-time per epoch [ms]  | MACS |
> | ----------- | ----------- | ----------- | ----------- | ----------- |
> | ResolvNet  | 39848 | 19.23	|  37818.73 |   57,261,648,000  |
> | GAT  | 351272 | OOM | OOM | OOM |
> | ChebNet   | 15016	| 8.48 | 548.78	|   7,176,793,216 |
>
>
>
> As can be inferred from the tables above, ResolvNet takes significantly less time to train per epoch on OGBN-Arxiv, when compared to GAT. It also needs considerably less GPU memory and multiply-accumulate operations compared to GAT.
>
> Compared to ChebNet, our method needs to compute an order of magnitude more sparse-dense matrix multiplication operations (as detailed in Appendix J). This is reflected in the number of performed MACs, which for ResolvNet is roughly an order of magnitude larger than that for ChebNet.
>
> On OGBN-Products, our method remains more complex than ChebNet. In contrast to GAT however, ResolvNet remains trainable.

---

> > ### Author Response · Authors · 2023-11-21
> > **Adressing Questions and References**
> >
> > * I understand the definition of  $G_\text{high}$ and $G_\text{reg}$ , but I am very clear how two split an original graph into this two graph. This is related to Definiton 2.1.
> >
> > We are glad to elaborate more on this.
> >
> > First let us point out that one never needs to conduct such a decomposition in practice, when deploying ResolvNet: This decomposition is effectively automatically  administered by the utilized resolvent-filters.
> >
> > Examples of graphs where such a decomposition is straightforward to compute manually are e.g. graphs that have strongly connected cliques or weighted graphs whose weights are distributed along two different weight scales (as in the examples of Figure 1 and Figure 2).
> >
> > An approach to checking whether a general graph $G$ has a multi-scale structure is as follows:
> >
> >     1) Enumerate all nodes. Translate node identifiers into one-hot-encodings. Combine the one-hot-identifiers of nodes (row-wise) into a node-feature matrix of dimension $N \times N$.
> >     2) Fix $z < 0$. Compute the resolvent $R_z(\Delta)$.
> >     3) Compute the matrix product $Y = R_z(\Delta)\cdot X$
> >     4) Treat rows in $Y$ as updated node features
> >
> > If there are nodes in $G$ whose updated feature vectors are very similar, they have significantly exchanged information.
> >
> >  Due to the structure of resolvents (c.f. Theorem 3.3) this is the case if they belong to the same strongly connected subgraph (assuiming connectivity reaching outside of this subgraph is considerably less pronounced).
> >
> > After a single application of a resolvent (given the initialization in Step 1) above), nodes belonging to the same connected component of a strongly connected subgraph of $G$ are thus characterized by having  (approximately) the same feature vectors.
> >
> > This provides an avenue to actually conduct the decomposition “$G = G_{\text{reg.}} + G_{\text{high}}$”  (even though it is never needed in practice when deploying ResolvNet).
> >
> >
> >
> > * Please elaborate on "we would have a Lipschitz continuity relation that allows to bound the difference in generated feature vector in terms of a judiciously chosen distance". This is the sentense above Eq. 1. I don't understand why.
> >
> > We are very happy to do so!
> >
> > For this question we are in the graph-level setting. Here we desire, that a graph neural network $\Psi$ assigns similar feature vectors to graphs $G$ and $\hat{G}$ that describe the same underlying object at different resolution scales.
> >
> > For a judiciously chosen distance metric $d(\cdot,\cdot)$ on the space of graphs, we would have that two graphs $G$ and $\hat{G}$ that describe the same underlying object at different resolution scales are  close to each other (i.e. $d(G,\hat{G}) \ll 1$).
> > If we then have a Lipschitz continuity condition such as
> > $$ \|   \Psi(G) - \Psi(\hat{G}) \|  \lesssim $d(G,\hat{G}), $$
> > this automatically have that if $G$ and $\hat{G}$ are similar (e.g. if they describe the same object at different resolutions) that the two feature vectors generated  by $\Psi$ for $G$ and $\hat{G}$ are close to each other:
> >
> > $$ \|   \Psi(G) - \Psi(\hat G) \|  \lesssim d(G,\hat G)    \ll 1$$
> >
> > Similar ideas are e.g. also used when trying to understand how Euclidean convolutional networks can be made robust to non-rigid deformations in images (e.g. slightly deforming a hand-written digit). A very thorough analysis of this line of reasoning in the Euclidean setting is e.g. provided in the PhD Thesis [1] (c.f. e.g. equation (1) on page 1).
> >
> > References:
> >
> > [1]: Scattering Representations for Recognition, Joan Bruna, Ecole Polytechnique, Palaiseau, France, 2013, https://pastel.hal.science/file/index/docid/905109/filename/phdmain_final.pdf

---

### Official Review · Reviewer_MwCo · 2023-10-31

**Soundness:** 3 good
**Presentation:** 3 good
**Contribution:** 3 good
**Rating:** 8
**Confidence:** 2

**Summary:**

This paper points out a problem in graph neural networks where certain strongly connected parts, like cliques, can limit the spread of information in the graph. To solve this, the authors introduce the idea of multi-scale consistency. This means keeping a connected way of spreading information even if the connection density in the graph changes for the node level tasks. For the graph level tasks, it means graphs generated from the same ground truth, which are at different resolutions,  should be assigned similar feature vectors. The research shows that many popular GNN designs don't have this feature. To fix this, the authors of this work propose ResolvNet, a new Spectral-based GNN design based on a math concept called resolvents. By applying resolvent of the Laplacian, 	ResolvNet is able to have the same effect of projecting the dense connected components in the original graph to a coarsened graph, then efficiently propagating information and finally projecting the embedding back to the original graph node level. Authors have theoretically proved that the proposed method is able to consistently integrate multiple connectivity scales occurring within graphs. Also , extensive experiments have shown that ResolvNet has multi-scale consistency and does better than other baselines in many tasks using various datasets. It is also shown that the proposed method is more stable than the baselines under different resolution scales.

**Strengths:**

*Originality*: The paper identifies a novel issue in graph neural networks and introduces an effective framework, ResolvNet, to address it. This represents a significant and innovative contribution to the field.

*Quality*: The investigative experiments and primary results presented in the paper are persuasive. Supported by solid theoretical proofs, this work stands out as a high-quality piece of research.

*Clarity*: The paper is exceptionally well-organized. Its straightforward and lucid presentation of both the problem and the proposed solution allows readers to grasp the content quickly and comprehensively.

*Significance*: By highlighting a new issue and offering an effective framework to tackle it, this work holds substantial impact potential for the broader community.

**Weaknesses:**

*Insufficient Analysis*: The paper could benefit from more extensive ablation studies and parameter analyses. Understanding how variations in parameters like $\omega$ and $k$, as defined in the ResolvNet Layer, impact the final results would provide deeper insights.

*Complexity of Concepts*: The concept of "resolvents" is not a commonly understood mathematical idea. Providing more explanations, along with practical application cases, would greatly aid readers in grasping this concept and its significance in the proposed framework.

Minor issue:

*Notation Introduction*: The paper occasionally lacks a comprehensive introduction to certain notations. For instance, the notation $T$ in section 3.2 is introduced without adequate context or explanation.

**Questions:**

The datasets utilized in this study are primarily small to medium-sized. How would ResolvNet perform in terms of accuracy and computational time when applied to larger datasets?

How do learnable filters as polynomials in resolvents achieve similar effects of up-projection operator and down-projection operator. It may need more illustrations and explanations for this in Sec 3.2.

---

> ### Author Response · Authors · 2023-11-21
> **Added Ablation Study and corrected notational Typo**
>
> We would like to sincerely thank the reviewer for the careful evaluation of our paper and appreciation of our results. We were especially happy to read that
>
> > [our paper] represents a significant and innovative contribution to the field,
>
> > this work stands out as a high-quality piece of research
>
> and that
>
> >  The paper is exceptionally well-organized
>
> Let us address the raised points individually:
>
> * The paper could benefit from more extensive ablation studies and parameter analyses.
>
> Following this comment, we have conducted ablation studies on both $\omega$ and $K$. The results are collected in Appendix K in our updated manuscript.
>
> As a rough summary, we see that increasing $K$ beyond $K = 1$ does not significantly improve performance. This is reminiscent of similar results for e.g. Chebnet, whose polynomial filtering operation generically does not perform better if the polynomial dimension is increased.
>
> For $\omega$, we see that a too small value has  negative implications as far as performance is concerned. For sufficiently large $\omega$ (small) variations of $\omega$ do not affect performance. I.e. ResolvNet is stable under (sufficiently small) variations in $\omega$.
>
> * Complexity of Concepts: The concept of "resolvents" is not a commonly understood mathematical idea. Providing more explanations, along with practical application cases, would greatly aid readers in grasping this concept and its significance in the proposed framework.
>
> We will be very glad to do so! For a camera ready version, we will include an additional section in the Appendix, detailing the significance of resolvents in other fields such as Mathematics and Physics, as well as a detailed mathematical introduction, (following the pedagogical structure of  (Teschl (2014)).
>
> * Minor issue: Notation Introduction: The paper occasionally lacks a comprehensive introduction to certain notations. For instance, the notation $T$ in section 3.2 is introduced without adequate context or explanation.
>
> This is indeed a typo, which we thank the reviewer for spotting. We corrected it and now write the Laplacian as $\Delta$ instead of $T$ also in Section 3.2.

---

> ### Author Response · Authors · 2023-11-21
> **ResolvNet on large graphs**
>
> * The datasets utilized in this study are primarily small to medium-sized. How would ResolvNet perform in terms of accuracy and computational time when applied to larger datasets?
>
> This is indeed an important question. Following this raised point, we now include a new and additional section (Appendix J: Scaling ResolvNet to large Graphs) in the Appendix that discusses in detail how ResolvNet is scaled to large graphs.
>
> The key ingredient here is a Neumann-representation of the resolvent that avoids an explicit matrix-multiplication:
> $$ (T - zId)^{-1} \approx \frac{1}{z} \sum\limits_{k=1}^K (-T/z)^k $$
> This approximation becomes exact as $K$ is taken o infinity. A similar approximation was already used to scale the PPNP method of “PPNP Paper” to large graphs.
> This replaces the hypothetical $\mathcal{O}(N^3)$ complexity during preprocessing and  $\mathcal{O}(N^2)$ complexity during training and inference that a naïve implementation of ResolvNet would constitute with a  $\mathcal{O}(EFN)$ complexity (i.e. the complexity of sparse-dense matrix multiplication). Here $N$ is the number of nodes,  $E$ is the number of edges and $F$ is the feature dimension.
>
> Beyond these theoretical considerations and in addition to the empirical complexity analysis already provided in Appendix G, we also evaluated complexity on large graphs experimentally:
> We trained  ResolvNet, ChebNet (both with $K = 1$) and GAT in a two-layer deep configuration with $64$ hidden units in each layer for $100$ epochs on the OGBN-Arxiv and OGBN-Products datasets, which are comprised of $169,343$ Nodes and more than 2M nodes respectively.
>
> Results are collected in the tables below:
>
> OGBN-Arxiv:
> |	    	 |  trainable parameters  |   max-memory-allocated [GB] | Training-time per epoch [ms]  | MACS|
> | ----------- | ----------- | ----------- | ----------- | ----------- |
> | ResolvNet  | 39848 	| 0.68  	| 634.71| 4,595,291,648 |
> |GAT  	| 351272 |  25.53   |  1089.96	| 3,468,144,640      |
> |ChebNet  |  15016	| 0.42	| 64.47 	| 433,518,080        |
>
>
> OGBN-Products:
>
> | |  trainable parameters |   max-memory-allocated [GB] | Training-time per epoch [ms]  | MACS |
> | ----------- | ----------- | ----------- | ----------- | ----------- |
> | ResolvNet  | 39848 | 19.23	|  37818.73 |   57,261,648,000  |
> | GAT  | 351272 | OOM | OOM | OOM |
> | ChebNet   | 15016	| 8.48 | 548.78	|   7,176,793,216 |
>
>
>
> As can be inferred from the tables above, ResolvNet takes significantly less time to train per epoch on OGBN-Arxiv, when compared to GAT. It also needs considerably less GPU memory and multiply-accumulate operations compared to GAT.
>
> Compared to ChebNet, our method needs to compute an order of magnitude more sparse-dense matrix multiplication operations (as detailed in Appendix J). This is reflected in the number of performed MACs, which for ResolvNet is roughly an order of magnitude larger than that for ChebNet.
>
> On OGBN-Products, our method remains more complex than ChebNet. In contrast to GAT however, ResolvNet remains trainable.
>
> As for accuracy on large graphs, we again expect better performance on homophilic graphs as ResolvNet’s (necessary) inductive bias towards homophily is independent of graph-size. A full experimental investigation of this question is still ongoing and will be included in  a camera ready version of this submission.

---

> ### Author Response · Authors · 2023-11-21
> **The relation of projection operators and resolvents**
>
> * How do learnable filters as polynomials in resolvents achieve similar effects of up-projection operator and down-projection operator. It may need more illustrations and explanations for this in Sec 3.2.
>
> The key insight here is provided by Theorem 3.3. From this Theorem, we infer that for multi-scale graphs we have
> $$R_z(\Delta) \approx J^{\uparrow}R_z(\Delta_{\text{collapsed}}) J^{\downarrow}. $$
> Additionally, we have $ J^{\downarrow} \cdot J^{\uparrow} = Id_{G_{\text{collapsed}}}$, as detailed in Appendix B.
> Combining these two facts yields
> $$(R_z(\Delta))^k \approx J^{\uparrow}(R_z(\Delta_{\text{collapsed}}))^k J^{\downarrow}. $$
>
> Thus also polynomials in resolvents effectively first project down to $G_{\text{collapsed}}$, then propagate there and subsequently interpolate pack up to the original graph $G$.
>
> We provide a full discussion of this in Appendix B (c.f. especially Theorem B.4).
> Following this comment by the reviewer, we now highlight this even stronger in the main body of our submission.

---

### Official Review · Reviewer_nd7U · 2023-11-01

**Soundness:** 3 good
**Presentation:** 3 good
**Contribution:** 3 good
**Rating:** 5
**Confidence:** 3

**Summary:**

The paper considers graphs with two scales, one in which nodes are strongly connected into clique-like communities and a another scale in which the connections are weaker and uniform over the graph. A distinction is based between the two communities based on spectral analysis: the second eigenvalue of the first scale is much higher than all the eigenvalues of the second scale. The idea of resolvents is proposed to deal with such graphs and two types of filters, type-0 and type-1 are defined to propagate information in a GNN. The ideas are validated empirically. It is shown that the proposed method works well on graphs with high homophily.

**Strengths:**

The idea of separating a network into multiple scales is nice. The problem is well defined and motivated

The use of resolvents to design filters is novel. A theory is developed to justify the methods.

The experimental results show the usefulness of the method.

**Weaknesses:**

1. It would be good if the authors could demonstrate the performance of their methods on synthetically generated graphs, say using stochastic block models. That would allow all parameters to be controlled.

2. It is not clearly defined how the two kinds of filters are combined: does a node learn which filter to use?

3. There are some other obvious baselines with which the authors could compare their methods:
a. Apply pooling to learn the clusters (say using diffpool, gpool, eigenpooling among others) and then use a generic GNN on the coarsened graph.
b. Separate the two networks using Gaussian Mixture Models, have two different GNNs for the two scales, and combine the two representations for node/graph prediction.

4. The abstract is not clear, especially the sentence: At the graph level, multi-scale .." The last sentence also seems to make bolder claims than what the experiments show.

**Questions:**

Please look at the weaknesses.

---

> ### Author Response · Authors · 2023-11-21
> **Conducted additional synthetic experiments and compared with suggested baselines**
>
> We would like to sincerely thank the reviewer for the careful review of our paper and the insightful advice, which we have followed diligently.
> We were especially happy to read that our
> > Idea of separating a network into multiple scales is nice
>
> and that our
>
> > experimental results show the usefulness of the method
>
> Let us address each raised point individually:
>
> * It would be good if the authors could demonstrate the performance of their methods on synthetically generated graphs, say using stochastic block models. That would allow all parameters to be controlled.
>
> We think that this is a great idea as stochastic block models indeed allow to generate graphs with strongly connected communities.
>
> Following this comment by the reviewer, we have thus conducted  additional experiments and ablation studies on graphs drawn from stochastic block models. We provide precise descriptions of our experimental setup, as well as full experimental results in the newly created section “Appendix H: Additional Experiments on synthetic Data”.
>
> Here we provide a brief summary of the main result:
>
> We generate graphs using a custom stochastic block model: Apart from **cluster-size** and **number-of-clusters**, our model takes as input two probabilities:
>
> The probability $p_{\text{inner}}$ determines the **likelihood of two nodes within the same cluster being connected** by an edge.
>
> The probability $p_{\text{inter}}$ determines the **likelihood for two nodes in different clusters to be connected** by an edge.
>
> To numerically verify Theorem 4.2 and showcase  the intuition behind Figure 2, we first keep all parameters  except for $p_{\text{inner}}$ constant.
> We then vary $p_{\text{inner}}$ from $p_{\text{inner}} = 0$ (no edges within the clusters) to $p_{\text{inner}} = 1$ (fully connected clusters).
>
> In this setup, we compare the feature vectors that ResolvNet and baselines generate for such graphs with the feature vectors generated by these models for coarse-grained versions of the original graphs where clusters are collapsed to single nodes.
>
> **Figure 14 (b-d) in Appendix H* plots corresponding results: As the connectivity within the clusters increases towards full connectivity, the norm difference between features of the original graph and that of its collapsed version decreases for ResolvNet.
> This is what we desire, as discussed in Section 2 and Section 4: Very strongly connected sub-graphs are supposed to be treated in a similar way that single (collapsed) nodes are.
>
> As is also evident from **Figure 14 in Appendix H**, baselines (in contrast to ResolvNet) do not exhibit this behaviour. This implies that these baselines are not able to consistently integrate multiple scales and hence are not transferable between different graphs describe the same underlying object at different resolution scales.
>
> Beyond these results Appendix H contains additional investigations and experiments (c.f. e.g. **Figure 15** and **Figure 16** in **Appendix H**).

---

> ### Author Response · Authors · 2023-11-21
> **Experiments for additional baselines I: Pooling**
>
> * There are some other obvious baselines with which the authors could compare their methods: a. Apply pooling to learn the clusters (say using diffpool, gpool, eigenpooling among others) and then use a generic GNN on the coarsened graph.
>
> We had indeed not thought of this, as our main intention is to design a model that might be deployed at any scale, and does not need to coarse-grain given graphs to a predetermined scale, like pooling does.
>
> Nevertheless, we agree with the reviewer that this is an interesting comparison. We thus followed the reviewers recommendation and combined GCN as a representative baseline with Self-Attention Graph Pooling (SAG-pooling) [1]. We chose SAG-pooling as the pooling method since its implementation does not need data to be loaded in a dense format and hence is compatible with experimental setup.
>
> We considered two implementations:
>
> “SAG” simply pools first and then runs GCN on the coarsened graph.
>
> “SAG-Multi” instead generates graph representations first at the original scale via GCN, then pools, runs GCN on the coarsened graph and generates a second (coarser) Graph representation. This procedure is iterated again and all three representations are combined into a single representation.
>
>
>
>
> We then reran all experiments on QM$7$ data described in Section 5 again also for these two additional baselines.
>
>
>
>
>
>
> We here first provide an updated version of Table 2 (listing prediction errors for atomic energy prediction on original QM$7$) with these two baselines added:
>
> | QM$7$  | MAE [kcal/mol] |
> |---|---|
> |BernNet   |113.57 $\pm$  62.90  |
> | GCN  | 61.32 $\pm$ 1.62 |
> |  ChebNet | 59.57 $\pm$ 1.58  |
> |  ARMA |  59.39 $\pm$ 1.79|
> |  SAG |  64.69 $\pm$  3.24|
> |  SAG-Multi|  61.36 $\pm$ 4.47|
> |  ResolvNet |  16.52 $\pm$0.67|
>
>
>
> As can be inferred from this  Table , either way of combining GCN (as the prototypical message passing network) with pooling does not lead to a performance increase on the multi-scale data of the QM$7$-dataset.
>
>
>
> We also investigated the graph-level multi-scale stability of these models, updating Table 3 (which evaluated the models trained for Table 2 on coarse-grained/collapsed versions of the molecular graphs in QM$7$):
>
> |  QM$7$-coarse | MAE [kcal/mol] |
> |---|---|
> |BernNet   |580.67 $\pm$99.27 |
> | GCN  |124.53 $\pm$ 34.58 |
> |  ChebNet | 645.14 $\pm$ 34.59  |
> |  ARMA |   248.96 $\pm$ 15.56  |
> |  SAG |  550.42 $\pm$ 24.43|
> |  SAG-Multi|  246.95 $\pm$ 68.49|
> |  ResolvNet |  16.23 $\pm$ 2.74|
>
> From this table with the previous one, we may infer that including a learned pooling operations  has a drastically negative effect when aiming to transfer models between graphs describing the same object (in this case a molecule) at different resolution-scales:
>
> Additional discussions and an investigation of the behaviour of the (GNN + Pooling)-baselines in comparison with ResolvNet under continuously varying the scale-imbalance within graphs is discussed in Appendix G.1 (c.f. also Fig. 13).

---

> ### Author Response · Authors · 2023-11-21
> **Experiments for additional baselines II: Manual Scale Separation**
>
> *    b. Separate the two networks using Gaussian Mixture Models, have two different GNNs for the two scales, and combine the two representations for node/graph prediction.
>
> We agree that this is an interesting point for comparison. To implement the corresponding experiment, we considered the QM$7$ dataset.
>
> As described in Section 5, weights here essentially correspond to inverse distance between atoms. We hence separate the scales by fixing a cut-off distance: Nodes that are within the cut-off distance of each other are considered to be connected via an edge in $G_{\text{high}}$. Pairs of nodes that are further apart from each other provide an edge in $G_{\text{reg}}$.
>
>  The resulting prediction accuracies are provided below:
> | MAE[kcal/mol]  | Standard  |Mixture   |
> |---|---|---|
> |BernNet   |113.57 $\pm$  62.90  |  85.24 $\pm$9.60 |
> | GCN  | 61.32 $\pm$ 1.62 | 64.16 $\pm$ 1.89 |
> |  ChebNet | 59.57 $\pm$ 1.58  | 47.11 $\pm$ 1.89  |
> |  ARMA |  59.39 $\pm$ 1.79| 30.88 $\pm$ 2.82  |
> |  **ResolvNet** | **16.52  $\pm$ 0.67** | [**16.52  $\pm$ 0.67**] |
>
> As can be inferred from this table, the idea of the reviewer significantly improves accuracy for most baselines. However, mean absolute errors of baselines, even after the two-scale-mixture modification, is still at least roughly twice as large as that of Resolvnet (16.52 $\pm$ 0.67 kcal/mol).
>
> Additionally, in order to combine baselines with such a scale-separation procedure, we had to manually select graph decomposition into the respective scales, while ResolvNet automatically takes care of this.
>
> Full details on the considerations above are provided in Appendix G.2.

---

> ### Author Response · Authors · 2023-11-21
> **Adressing additional points and References**
>
> * It is not clearly defined how the two kinds of filters are combined: does a node learn which filter to use?
>
> In principle, any Type-0 filter can represent any fixed Type-I filter, so that the type of filter may indeed be learned.
> In practice, however it is beneficial to equip models with a stronger inductive bias towards the desired type of transferability (c.f. the discussion after Theorem 4.1). Thus in practice we treat the type-of-filter-choice as a binary hyperparameter. We have made this clearer in our updated manuscript now (c.f. the updated discussion after eq. (2)).
>
> * The abstract is not clear, especially the sentence: At the graph level, multi-scale .." The last sentence also seems to make bolder claims than what the experiments show.
>
> Following this comment, we  have amended the last sentence of the abstract and changed the “At the graph level, multi-scale .."- sentence  to “At the graph-level, a multi-scale consistent graph neural network assigns similar feature vectors to distinct graphs." describing the same object at different resolutions".
> We hope that the new formulation is clearer now; but would be happy to change the sentence again.
>
> References:
>
> [1]: Junhyun Lee, Inyeop Lee, and Jaewoo Kang. Self-attention graph pooling. In Kamalika Chaudhuri and Ruslan Salakhutdinov (eds.), Proceedings of the 36th International Conference on Machine Learning, ICML 2019, 9-15 June 2019, Long Beach,  California, USA, volume 97 of Proceedings of Machine Learning Research, pp. 3734–3743. PMLR, 2019. URL http://proceedings.mlr.press/v97/lee19c.html

---

> > ### Comment · Reviewer_nd7U · 2023-11-22
> > **Thanks for the response.**
> >
> > Thanks for conducting extensive experiments based on the feedback. I am not sure of your commentary on the performance of the mixture model. The gap between the two approaches is quite significant.

---

> ### Author Response · Authors · 2023-11-22
> **ResolvNet performs significantly better than mixture models and mixture models are not transferable between different resolutions**
>
> We thank the reviewer for the continued engagement and are very happy to extend our discussion of mixture models:
>
>
>
>
> | QM$7$-MAE [kcal/mol]  | |
> |---|---|
> |BernNet-mixture   |  85.24 $\pm$9.60 |
> | GCN-mixture  | 64.16 $\pm$ 1.89 |
> |  ChebNet-mixture |  47.11 $\pm$ 1.89  |
> |  ARMA-mixture |  30.88 $\pm$ 2.82  |
> | **ResolvNet** |  **16.52 $\pm$ 0.61** |
>
>
> Performing a manual scale-separation and running simultaneous networks on the different graph-structures improved the relative performance of baselines. This is most pronounced for ARMA, where the performance gap between the standard- and mixture-implementations is roughly a factor of two, which is quite significant; as the reviewer points out.
>
> However, **ResolvNet performs better** than this best performing micture-model (ARMA-mixture) **by an additional factor of two**; i.e. with a performance improvement of exactly the same significance.
>
> Thus ResolvNet is still able to incorporate multi-scale data a lot better than mixture-models.
>
>
> As an important additional point we note the following:
>
> **Considering mixture-models (as opposed to ResolvNet) does not lead to graph-level multi-scale consistency.**
>
> Mixture-models (as opposed to ResolvNet) are unable to consistently handle graphs describing the same object at different resolution scales:
>
> To showcase this, we have now investigated the graph-level multi-scale stability of mixture-models, by rerunning the experiment corresponding to Table 3 (which evaluated the models trained for Table 2 on coarse-grained/collapsed versions of the molecular graphs in QM$7$) for mixture-models:
>
>
>
>
> | QM$7_{\text{coarse}}$MAE[kcal/mol] | |
> |---|---|
> |BernNet-mixture    |300.02 $\pm$  18.13|
> | GCN-mixture   | 198.44 $\pm$ 35.55|
> |  ChebNet-mixture  |  821.10 $\pm$ 41.62  |
> |  ARMA-mixture  |  597.12 $\pm$ 35.54  |
> | **ResolvNet** |  **16.23 $\pm$ 2.74** |
>
> When confronted with such different resolution scales, the performance difference between ResolvNet and mixture-models becomes even larger:
>
>
> While ResolvNet performed better than mixute models by a factor of at least two on same-resolution multi-scale data, the performance gap now varies between a factor of $12$ to $50$ if multiple graph-resolutions are considered.
>
> Thus:
> 1) For single-resolution multi-scale data, ResolvNet performs better than mixture-models by a factor of two and more
> 2) In multi-resolution, multi-scale settings ResolvNet performs better than mixture-models by a factor of $12$ to $50$
>
> We hope our extended discussion was able to clarify our earlier commentary. We of course stand by, should further questions arise.

---

> > ### Comment · Reviewer_nd7U · 2023-11-23
> > **Thanks for the details**
> >
> > 1. What was the effect of manual separation on the node classification datasets?
> > 2. If I understand the authors' response, the performance of the method should be judged more on the regression task than the node classification task. Why then there is so much emphasis being placed on node classification in the experimental results? Are there any other tasks where the method does well?

---

> ### Author Response · Authors · 2023-11-23
> **ResolvNet is versatile and performs well in multiple settings**
>
> We sincerely thank the reviewer for the continued engagement with our work.
>
> * What was the effect of manual separation on the node classification datasets?
>
> One of the inherent drawbacks of the suggested manually-scale-separated mixture-model baseline is that any given graph needs to be manually separated into distinct graph structures (corresponding to the different scales) before such a baseline may be run.
>
> The node classification datasets (i.e. Citeseer, Cora, etc.) are not inherently multi-scale.
> It is thus not clear how such a manual separation into distinct scales as suggested  should proceed on these datasets.
>
> Should the reviewer have a separation method in mind, we would be happy to combine this separation-method with the already suggested mixture-models and conduct further experiments with this additional baseline in a camera ready version of our submission.
>
> * If I understand the authors' response, the performance of the method should be judged more on the regression task than the node classification task. Why then there is so much emphasis being placed on node classification in the experimental results? Are there any other tasks where the method does well?
>
> The focus of our paper is to introduce the concept of multi-scale consistency for graph neural networks. As we detail in the paper, this concept has two aspects:
>
> 1)	For a fixed multi-scale graph, multi-scale consistency refers to an unrestricted flow of information over the entire graph, even if connectivity varies within said graph
>
> 2)	When confronted with graphs describing the same object at different resolution scales, a multi-scale consistent network assigns similar feature vectors to graphs describing this same object at different resolution scales.
>
> We showcase the lack of multiscale consistency of baselines and how the proposed ResolvNet is able – in contrast to baselines –  to consistently incorporate varying  scales in different experiments:
>
> 1)	To showcase the first point 1) above, we e.g. conduct experiments on QM$7$ data. Here edge weights essentially correspond to inverse distances. Atoms that are far-away from each other are thus connected via edges with far lower weights. However, long-range communication (i.e. along the weight suppressed lower connectivity scale) within such molecular data is important for prediction tasks (c.f. the references provided in our paper). Since ResolvNet is able to better incorporate such varying connectivity scales than baselines (as established in extensive theoretical discussions, in our experiments and further cemented via additional experiments in the discussion phase), ResolvNet is able to better predict molecular properties than baselines.
>
> 2)	To showcase the second point 2) above, we e.g. confronted ResolvNet and baselines with coarse-grained descriptions of molecules during inference. ResolvNet’s prediction capabilities remained the same, while those of baselines severely declined (c.f. Table 3, or the discussion in an earlier comment).
>
> While these are the settings that our paper is concerned with and indeed the type of tasks ResolvNet was designed for, we find it important to also evaluate ResolvNet against other baselines on the standard banchmark of node classification. We make this clear by writing
>
> > To establish that the proposed ResolvNet architecture not only performs well in multi-scale settings, we conduct node classification experiments […]
>
> when discussing the corresponding node-classification results of Table 1.
>
> An inductive-bias towards homophily is necessary, to achieve multiscale consistency, as was discussed in the paper. We thus expect ResolvNet to perform well on homophilic graphs.
> This is indeed what our experiments show, with ResolvNet being by far the best performing method in this realm (c.f. Table 1).
>
> In total we thus established ResovNet’s superior performance in three different real-world settings:
>
> 1)	For node classification on homophilic graphs, ResolvNet was the best performing method (c.f. Table 1)
>
> 2)	On muli-scale data ResolvNet performed significantly better than baselines (c.f. e.g. Table 2 for experiments on corresponding real world data)
>
> 3)	In the multi-resolution-scale setting, ResolvNet also performed significantly better than baselines (c.f. e.g. Table 3 or Figure 7)
>
> We hope this discussion has clarified the versatility aspect of ResolvNet.
>
> We of course stand by should additional questions arise and would be happy to include even more experiments in a camera ready version of our paper.

---

### Author Response · Authors · 2023-11-23
**Response to all Reviewers**

As the discussion phase draws to a close, we would like to thank all reviewers again for the time and energy invested into reviewing our paper!

We were encouraged to see that  **for all categories of  soundness, presentation and contribution our paper almost exclusively received the high score of $3$** (i.e. by reviewers nd7U, MwCo, ZGZ9).

Since our paper is concerned with introducing the completely novel concept of multiscale-consistency we were happy to read that this issue was considered to be “**indeed an important topic**” (reviewer ZGZ9), that our paper “**provides a very clear definition of  multiscale consistency**”,  is “**exceptionally well organized**” and the “**straightforward and lucid** presentation […] allows readers to **grasp the content quickly and comprehensively.**”  (reviewers ZGZ9, MwCO).

We were especially delighted to read the sentiment that our “work holds **substantial impact potential** for the broader community.” and  “stands out as a **high-quality piece of research**” (reviewer MwCo).

Having introduced ResolvNet as a novel method that (as opposed to baselines) _is_ in fact multi-scale consistent, we were also happy to see that reviewers  formed the opinion that our **experimental results** “**show the usefulness of the method**” (reviewer nd7U) and “**are persuasive**” (reviewer MwCo).

Reviewer ZGZ9 was interested in an efficiency-analysis of ResolvNet on large graphs, and we hence included such an efficiency analysis (both from a theoretical perspective and via experiments) in our updated manuscript (c.f. Appendix J).

Reviewer MwCO asked for an ablation study on our hyperparameters $\omega$ and $K$, which we have dilligently included in our updated manuscript (c.f. Appendix K).

Reviewer nd7U asked us to conduct **additional experiments** on synthetic graphs and to compare with additional “conceptional” baselines. We thus conducted extensive additional experiments (c.f. e.g. Appendix H and Appendix G for corresponding results): Here we **cemented our existing findings** on real world data also on synthetic graphs and showed that ResolvNet also performs better than the suggested additional conceptional baselines.

Reviewer 2iPY was concerned that our method might be outperformed by GCNII on the task of node-classification. We hence included GCNII into our node-classification experiment: **ResolvNet performs in fact better** than GCNII, especially on the relevant homophilic datasets.
Additionally we cleared up the confusion surrounding **Table 3**: The corresponding metric is mean-absolute-error (i.e. MAE), so that our **ResolvNet** actually **performs best** (not worst) **by a large margin**.

Below, we have provided detailed responses to received comments and answers to raised questions in the individual responses.

We look forward to the second stage of the discussion period and stand by to address any further concerns, should they arise.

---

### Meta-Review · Area_Chair_akgb · 2023-12-07

**Metareview:**

This paper studies the problem that strongly connected components like cliques limit information propagation. As a solution, authors propose multi-scale consistency which encourages that the same graph at different “connection strength” scales should have similar representations. For node level tasks, it encourages consistent information propagation across “graph resolutions” with different connection strengths. Authors prove that known GNNs don’t have this property and propose a new model called ResolveNet to remedy this. ResolveNet improves the performance on several datasets. Overall paper identifies an interesting issue in GNNs, and proposes a novel solution for it with solid theoretical analysis and useful empirical results.

However, paper can improve clarity in presentation as pointed out by several reviewers. Additionally, the paper does not provide enough experiments on large scale datasets. Specifically, considerations provided during the review period lacks discussion of minibatching and actual performance on large datasets. Finally, it is possible that baseline method performances can be further improved. For example, authors mention that GCNII performance from past work cannot be directly compared since the settings are different. However, they directly use some of the hyper parameters from that paper instead of tuning them for the considered setting.

**Justification For Why Not Higher Score:**

Lack of clarity and enough experimentation

**Justification For Why Not Lower Score:**

N/A

---

### Decision · Program_Chairs · 2024-01-16

Reject